# INFORMATION SHAPES KOOPMAN REPRESENTATION

**Xiaoyuan Cheng**[1*]    **Wenxuan Yuan**[2*]    **Yiming Yang**[1]    **Yuanzhao Zhang**[3]
**Sibo Cheng**[4]    **Yi He**[1]    **Zhuo Sun**[5,6†]

[1]Dynamic Systems Lab, University College London
[2]Department of Earth Science and Engineering, Imperial College London
[3]Santa Fe Institute
[4]CEREA, ENPC and EDF R&D, Institut Polytechnique de Paris
[5]School of Statistics and Data Science, Shanghai University of Finance and Economics
[6]Institute of Big Data Research, Shanghai University of Finance and Economics
* *Xiaoyuan Cheng and Wenxuan Yuan contributed equally to this work.*

## ABSTRACT

The Koopman operator provides a powerful framework for modeling dynamical systems and has attracted growing interest from the machine learning community. However, its infinite-dimensional nature makes identifying suitable finite-dimensional subspaces challenging, especially for deep architectures. We argue that these difficulties come from suboptimal representation learning, where latent variables fail to balance expressivity and simplicity. This tension is closely related to the information bottleneck (IB) dilemma: constructing compressed representations that are both compact and predictive. Rethinking Koopman learning through this lens, we demonstrate that latent mutual information promotes simplicity, yet an overemphasis on simplicity may cause latent space to collapse onto a few dominant modes. In contrast, expressiveness is sustained by the von Neumann entropy, which prevents such collapse and encourages mode diversity. This insight leads us to propose an information-theoretic Lagrangian formulation that explicitly balances this tradeoff. Furthermore, we propose a new algorithm based on the Lagrangian formulation that encourages both simplicity and expressiveness, leading to a stable and interpretable Koopman representation. Beyond quantitative evaluations, we further visualize the learned manifolds under our representations, observing empirical results consistent with our theoretical predictions. Finally, we validate our approach across a diverse range of dynamical systems, demonstrating improved performance over existing Koopman learning methods. The implementation is publicly available at https://github.com/Wenxuan52/InformationKoopman.

## 1 INTRODUCTION

Modeling and predicting the behavior of nonlinear dynamical systems are fundamental problems in science and engineering (Brunton et al., 2020; Kovachki et al., 2023; Mezic, 2020). Classical approaches typically rely on nonlinear differential equations or black-box learning methods. In contrast, the Koopman operator framework offers a compelling alternative: it represents nonlinear evolution as a linear transformation in an appropriate function space (Koopman, 1931; Fritz, 1995).

**Motivation.** This linearization principle has recently attracted significant attention in the deep learning community, as it enables complex nonlinear dynamics to be modeled and predicted using linear representations. However, integrating this framework into deep architectures poses a fundamental challenge: the Koopman operator is inherently infinite-dimensional, necessitating the identification or learning of a suitable finite-dimensional subspace for practical implementation. Deep learning models, most notably variational autoencoders (VAEs), have been employed to approximate such

---

*Email Xiaoyuan Cheng and Wenxuan Yuan to ucesxc4@ucl.ac.uk and wenxuan.yuan@qq.com.
†Corresponding Author. Corresponding to Zhuo Sun: sunzhuo@mail.shufe.edu.cn.

subspaces in a purely data-driven manner (Otto & Rowley, 2019; Pan & Duraisamy, 2020; Liu et al., 2023). Yet in practice, the resulting representations often suffer from instability, mode collapse or fail to produce reliable dynamics. To address these challenges, some studies incorporate domain-specific priors—such as symmetry, conservation laws, dissipation, or ergodicity (Vaidya & Mehta, 2008; Weissenbacher et al., 2022; Azencot et al., 2020; Cheng et al., 2025)—into the Koopman representation. While effective in restricted settings, such approaches lack general principles for guiding Koopman representation. This calls for a more general and principled approach to constructing finite-dimensional representations, one that balances simplicity, in the form of latent linearity, with sufficient expressiveness (more literature review in Appendix C).

**Information Bottleneck View.** A natural way to achieve the tradeoff between simplicity and expressiveness is through the lens of the Information Bottleneck (IB). The classic IB framework formalizes the idea that a good representation should compress the input as much as possible while preserving information relevant to a downstream task (Tishby et al., 2000; Tishby & Zaslavsky, 2015). In the context of representation learning (see Table 1), this typically means finding a latent variable $z$ that minimizes Complexity$(x, z)$[1] from input $x$, while retaining expressiveness by improving Relevance$(z, y)$ (Vera et al., 2018). Instead of a static latent representation, the goal of Koopman representation is to predict the future state $x_n$ given the current state $x_{n-1}$ via a latent variable $z_{n-1}$. This gives rise to a dynamical information bottleneck formulation: we aim to learn a Koopman representation $z_{n-1}$ with maximal linear predictability of future state $x_n$, while remaining as compact as possible.

Table 1: Information-theoretic comparison between standard and Koopman representations. Here, $\beta$ controls the trade-off between simplicity and future-state expressiveness.

| | Latent Representation | | Koopman Representation | |
|---|---|---|---|---|
| **Goal** | Disentangled $z$ | | Predictive $z_{n-1}$ | |
| **Info. Flow** | $x \to z \to y$ | | $x_{n-1} \to z_{n-1} \xrightarrow{\text{Koopman operator}} z_n \to x_n$ | |
| **Lagrangian** | $\beta$ Complexity$(x, z)$ $-$ Relevance$(z, y)$ | | $\beta$ Complexity$(x_{n-1}, z_{n-1})$ $-$ Relevance$(z_{n-1}, x_n)$ | |

**Why Is Finding a Good Koopman Representation Challenging?** Learning Koopman representation imposes stricter constraints than conventional latent representation models (see Table 1). In VAE or $\beta-$VAE (Kingma et al., 2013; Burgess et al., 2018), the focus is on reconstructing the input $x$ or sampling from its distribution, which only requires the latent representation $z$ to contain enough information about $y$. However, in Koopman learning, the latent space needs to support linear forward from $z_n$ to $z_{n+1}$ in some finite-dimensional spaces. This constraint implies that the latent representation must not only capture information about the current state but also conform to a linear predictive structure (*structural consistency*) (Mardt et al., 2018; Kostic et al., 2023a;b; 2024), which imposes a stronger restriction. Prior work has shown that simply increasing the dimensions of the latent space does not necessarily improve performance (Li et al., 2020; Brunton et al., 2021), underscoring the importance of maintaining *temporal coherence*, i.e., ensuring that latent trajectories evolve consistently over time to prevent instability and error accumulation. Moreover, *predictive sufficiency* requires that the latent representation retains enough Koopman modes to faithfully reconstruct the system's future trajectories, such that multi-step prediction accuracy is preserved (Wang et al., 2022; 2025). Unlike standard VAEs and their variants, which emphasize flexible latent representations to support reconstruction, Koopman models demand dynamically consistent latent representations: small deviations can propagate and amplify over time. In summary, while conventional representation learning emphasizes disentanglement and reconstruction, Koopman representation learning requires three key properties: temporal coherence, predictive sufficiency, and structural consistency. The IB framework provides a meta view to navigate these trade-offs. It enables us to ask the central question:

*Is it possible to learn Koopman representations that are both structurally simple and expressive, under the guidance of information-theoretic principles?*

Motivated by this question, we approach the problem from a fresh IB perspective, leading to core contributions: **Theoretical Insight.** We develop an information-theoretic framework for Koopman representation, proving that mutual information controls error bounds while von Neumann entropy

---

[1]By Complexity$(x, z)$ we mean the mutual information $I(x; z)$ in the IB framework, quantifying how much of the input information $x$ is retained in $z$. Relevance$(z, y)$ denotes $I(z; y)$, the information $z$ carries about $y$.

determines the effective dimension. By disentangling the information content of Koopman representations, we reveal how temporal coherence, predictive sufficiency, and structural consistency are governed by latent information and how these components are intrinsically connected to the spectral properties of the Koopman operator. This yields a novel information-theoretic Lagrangian that extends the classical IB principle by explicitly incorporating dynamical constraints, thereby making the fundamental trade-off between simplicity and expressivity in Koopman representation mathematically explicit. **Principled Framework.** Building on our information-theoretic Lagrangian, we derive a tractable, architecture-agnostic loss function that translates our theory into a practical algorithm. Each term of the loss corresponds directly to one of the three desiderata—temporal coherence, predictive sufficiency, and structural consistency. This yields a general algorithm that is broadly applicable: it extends naturally from physical dynamical systems to high-dimensional visual inputs and graph-structured dynamics, and our empirical results validate the theoretical predictions.

## 2 PRELIMINARIES

**Notation.** Let $\mathcal{M} \subset \mathbb{R}^n$ be a finite-dimensional manifold equipped with a measure $\mu$. Consider a discrete-time nonlinear map $T : \mathcal{M} \rightarrow \mathcal{M}$, so that the state $x_t \in \mathcal{M}$ evolves according to $x_t = T(x_{t-1})$. We denote by $\mathcal{H} = L^2(\mathcal{M}, \mu)$, the Hilbert space of real-valued observables $\phi : \mathcal{M} \rightarrow \mathbb{R}$.

**Definition 2.1 (Koopman Operator (Koopman, 1931))** *The* Koopman operator $\mathcal{K} : \mathcal{H} \rightarrow \mathcal{H}$ *is a linear operator acting on observables as*

$$(\mathcal{K}\phi)(x) = \phi(T(x)), \quad for \ \phi \in \mathcal{H}, \ x \in \mathcal{M}. \tag{1}$$

Despite the appeal of lifting nonlinear dynamics into a linear forward via Koopman representation, practical approximations require projecting the infinite-dimensional function space $\mathcal{H}$ onto a finite-dimensional subspace. In the Koopman learning framework, this restriction manifests as learning a finite set of effective latent features $\{\phi_1, \phi_2, \ldots, \phi_d\}$ that map the state $x$ as a latent representation $z := \phi(x) \in Z \subsetneq \mathcal{H}$, where $Z$ is the latent space spanned by the selected latent features. The center of this paper is on discussion how to find a good representation $z$. To ground the principles of information theory, we introduce some essential technical definitions.

**Definition 2.2 (Mutual Information (MacKay, 2003))** *Given two random variables $x$ and $y$ with joint probability distribution $p(x, y)$ and marginal distributions $p(x)$ and $p(y)$, the* mutual information $I(x; y)$ *quantifies the amount of information shared between $x$ and $y$, and is defined as*

$$I(x; y) = \mathbb{E}[\log \frac{p(x, y)}{p(x)p(y)}] = \mathbb{E}[\log \frac{p(y|x)}{p(y)}].$$

**Definition 2.3 (Von Neumann Entropy (Witten, 2020))** *Let $\rho \in \mathbb{R}^{d \times d}$ be a symmetric, positive semidefinite matrix with trace $1$. The* von Neumann entropy *of $\rho$ is defined as*

$$S(\rho) = -\mathrm{tr}(\rho \log \rho).$$

*If $\{\lambda_i\}_{i=1}^d$ are the eigenvalues of $\rho$, then $S(\rho) = -\sum_{i=1}^d \lambda_i \log \lambda_i$. This value reflects latent effective dimensions: it is close to $0$ if $\rho$ is concentrated on a single direction, and close to $\log d$ if $\rho$ is spread uniformly. More connection with effective dimension is given in Appendix E.*

Intuitively, mutual information and the von Neumann entropy provide a principled way to measure the predictability and the intrinsic effective dimension of Koopman representation. Building on these preliminaries, we can quantify the preserved information under Koopman representation.

## 3 METHOD

Our approach proceeds as: (1) a probabilistic analysis in Koopman representation how information loss drives error accumulation; (2) an information-theoretic characterization linking lost information to Koopman spectral properties; (3) a general Lagrangian formulation to guide better representation.

### 3.1 INFORMATION FLOW IN KOOPMAN REPRESENTATION

**A probabilistic view of Koopman representation.** Firstly, we denote $x_{1:t}$ and $z_{1:t}$ as the states and their corresponding autoregressively generated latent variables from time step $1$ to $t$, respectively. According to the direct information flow in Table 1, the Koopman representation induces the following trajectory distribution given $x_0$:

$$p^{KR}(x_{1:t}|x_0) = \int p(z_0|x_0) \prod_{n=1}^{t} p(z_n|z_{n-1})p(x_n|z_n)dz_0 dz_1 \cdots dz_t. \tag{2}$$

Here, $p(z_0|x_0)$ acts as the encoder, mapping the initial state into a latent variable. The latent forward is modeled by a linear Gaussian transition, where $p(z_n|z_{n-1}) = \mathcal{N}(z_n|\mathcal{K}z_{n-1}, \Sigma)$ is a probabilistic representation of equation 1 with variance $\Sigma$. This directly reflects Definition 2.1, as the latent evolution is constrained to be linear. Finally, each state $x_n$ is reconstructed from its corresponding latent variable $z_n$ via a decoder $p(x_n|z_n)$, typically instantiated as a Gaussian. We now turn to the fundamental question of whether information is inevitably lost during latent propagation.

> **Proposition 1 (Information Loss in Latent Evolution)** *Let $x_{n-1} \to z_{n-1} \xrightarrow{\mathcal{K}} z_n \to x_n$ represent the information propagation in Koopman representation as shown in equation 2. Then, by the property of mutual information, the following holds:*
>
> $$I(x_{n-1}; x_n) \geq I(z_{n-1}; x_n) \geq I(z_{n-1}; z_n). \tag{3}$$

The detailed proof and its multi-step extension are provided in Appendix F.1. The first inequality reflects that the mapping $x_{n-1} \to z_{n-1}$ is a compressed representation, which may discard predictive information about $x_n$. The second inequality indicates that the latent forward propagation $z_{n-1} \to z_n$ is governed by Koopman operator, inherently limits the information that can be preserved in the latent space. As a result, $I(z_{n-1}; x_n)$ is larger than $I(z_{n-1}; z_n)$, since the future state $x_n$ generally carries more dependencies on $z_{n-1}$ than the latent evolution alone. In this sense, $I(z_{n-1}; z_n)$ sets the *information limit* of Koopman representation by the operator $\mathcal{K}$.

While Proposition 1 shows the degradation of information along latent propagation, it remains an abstract statement that is not directly tractable under the complex trajectory distributions in equation 2. To obtain a tractable measure, we turn to the Kullback–Leibler (KL) divergence as a natural way to quantify the discrepancy between true and Koopman-induced trajectories:

$$D_{\mathrm{KL}}\left(p(x_{1:t}|x_0) \,\|\, q^{KR}(x_{1:t}|x_0)\right) \leq D_{\mathrm{KL}}\left(p(x_{1:t}|x_0) \,\|\, p^{KR}(x_{1:t}|x_0)\right) + \mathcal{E}_{\mathrm{enc}} + \mathcal{E}_{\mathrm{tra}} + \mathcal{E}_{\mathrm{rec}} \tag{4}$$

Here, $p$ is the true distribution and $p^{KR}$ is the ideal Koopman model distribution in equation 2 without any approximations. $q^{KR}$ is the variational approximation, $\mathcal{E}_{\mathrm{enc}}$, $\mathcal{E}_{\mathrm{tra}}$ and $\mathcal{E}_{\mathrm{rec}}$ are approximation errors induced by the latent representation, Koopman operator and reconstruction (see details in Appendix F.2). This motivates the following result, which formalizes how the information gap translates into an autoregressive error bound for Koopman representations.

> **Proposition 2 (Autoregressive Error Bound of Koopman Representation)** *The distribution discrepancy between the true and Koopman-induced trajectories is bounded by the information gap as*
>
> $$\|p(x_{1:t} \mid x_0) - q^{KR}(x_{1:t} \mid x_0)\|_{TV} \leq \sqrt{\tfrac{1}{2}\left[D_{\mathrm{KL}}(p(x_{1:t} \mid x_0) \,\|\, p^{KR}(x_{1:t} \mid x_0)) + \mathcal{E}\right]}$$
>
> $$\leq \sqrt{\tfrac{1}{2}\sum_{n=1}^{t}\left(I(x_{n-1}; x_n) - I(z_{n-1}; z_n)\right) + \mathcal{E}}. \tag{5}$$
>
> *Here, $\|\cdot\|_{TV}$ is the total variation distance. The upper error bound is obtained as*
>
> $$\left\|\mathbb{E}_{q^{KR}}[x_{1:t} \mid x_0] - \mathbb{E}_p[x_{1:t} \mid x_0]\right\|_2 \leq \bar{C}\sqrt{2\sum_{n=1}^{t}\left(I(x_{n-1}; x_n) - I(z_{n-1}; z_n)\right) + \mathcal{E}}, \tag{6}$$
>
> *where $\bar{C}$ is a positive constant and $\mathcal{E}$ is related to the approximation error in equation 4.*

The proof is in Appendix F.3. The KL divergence between the true and Koopman-induced trajectory distributions reflects how much temporal coherence is lost during representation. Here, $I(x_{n-1}; x_n)$ quantifies the intrinsic dynamical coupling $T$ in the original system, while $I(z_{n-1}; z_n)$ characterizes the information of that coupling that exists under Koopman representation. Since $I(z_{n-1}; z_n)$ acts as the information limit (see Proposition 1), the gap $I(x_{n-1}; x_n) - I(z_{n-1}; z_n)$ measures the information that is lost when nonlinear dynamics are approximated by Koopman representation. Also, we link the upper/lower error bounds and distribution discrepancy in equations 6 (lower bound see equation 25). It reflects the prediction error is bounded by the step-wise information limit.

## 3.2 INFORMATION COMPONENTS IN KOOPMAN REPRESENTATION

The latent mutual information quantifies the magnitude of error, but does not uncover how this loss relates to Koopman spectral properties. To sharpen the insight from Propositions 1 and 2, we consider the aggregated quantity $I(z_t; x_t)$, which measures the total information available to the decoder $p(x_t|z_t)$. Our focus is on how much of this information can be stably propagated from past latent variables $z_{t-n}$.

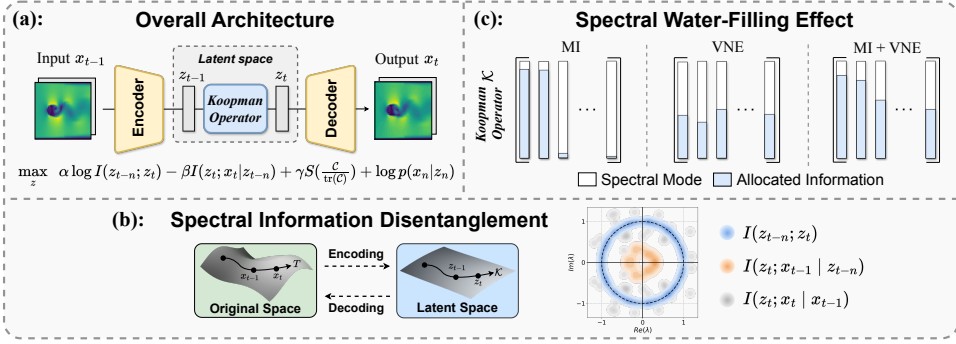

Figure 1: Information-theoretic Koopman framework. (a) Structure overview, (b) Information disentanglement with spectral interpretations, and (c) Water-filling effect of Mutual Information (MI) and von Neumann entropy (VNE) on spectral information allocation.

**Proposition 3 (Information Disentanglement and Spectral Property)** *The mutual information $I(z_t; x_t)$ can be disentangled into a summation of three distinct components, each with a spectral interpretation (see proof in Appendix F.4, see Figure 1(b)):*

| Component | Temporal-coherent | Fast-dissipating | Residual |
|---|---|---|---|
| Spectral property | $\lambda \approx 1$ | $\lambda < 1$ | no counterpart |
| Mutual info term | $I(z_{t-n}; z_t)$ ↑ | $I(z_t; x_{t-1} \mid z_{t-n})$ ↓ | $I(z_t; x_t \mid x_{t-1})$ ↓ |

*The decomposition shows that Koopman representations preserve temporal-coherent information associated with spectral modes of the Koopman operator whose eigenvalues lie near the unit circle, while information linked to dissipating modes ($|\lambda| < 1$) decays rapidly and noise-like components have no spectral support, hence compressible.*

*(1). Temporal-coherent information $I(z_{t-n}; z_t)$ (see closed form equation 29).* This term represents information that persists along the latent evolution $z_{t-n} \to \cdots \to z_t$. It corresponds to conserved or slowly dissipating information that remains stable during latent evolution. From a spectral perspective, *these are associated with Koopman modes whose eigenvalues are near to the complex unit circle (i.e., $|\lambda| \approx 1$)*, implying that the corresponding information is propagated almost losslessly across time and hence remains mutually informative between past and present latent variables.

*(2). Fast-dissipating information $I(z_t; x_{t-1}|z_{t-n})$ (see closed form equation 35).* This term reflects short-term dependencies that arise from the most recent state $x_{t-1}$, beyond what is already encoded in the past latent state $z_{t-n}$. Such information provides transient predictive power but quickly leaks out, since the autoregressive latent evolution $z_{t-n} \to \cdots \to z_t$ cannot continually access external inputs $x_{t-1}$. In contrast, *these contributions are associated with Koopman modes whose eigenvalues satisfy $|\lambda| < 1$, indicating exponential information dissipation with forward steps.* Consequently,

the mutual information they contribute vanishes rapidly as the time step $n$ increases, making those modes inherently cannot be captured by temporal-coherent information.

*(3). Residual information $I(z_t; x_t | x_{t-1})$ (see closed form equation 36).* This term measures unpredictable information in the current state that cannot be explained from the past state. It corresponds to information injected at the present step—such as noise or anomalies—that interferes with constructing a coherent latent state. Unlike temporal-coherent or fast-dissipating modes, *these residuals have no spectral counterpart in the Koopman operator: they are not tied to any eigenvalue structure.* From the IB perspective, such non-predictive component is compressible. Having the disentangled information, the next question is how latent mutual information shapes Koopman representations.

> **Proposition 4 (The Role of Latent Mutual Information)** *Maximizing the latent mutual information $I(z_{t-n}; z_t)$ allocates spectral weights to temporally coherent modes in the latent space, thereby enhancing the relevance of the Koopman representation. However, excessive emphasis on this objective can lead to mode collapse, where the representation concentrates on only a few dominant modes and loses effective dimension (see Figure 1(c)).*

In Koopman representation, the latent mutual information admits a closed form

$$I(z_{t-n}; z_t) = \frac{1}{2} \log \det(\mathrm{I} + M_n^{-\frac{1}{2}} (\mathcal{K}^n) \mathcal{C} (\mathcal{K}^n)^\top M_n^{-\frac{1}{2}}) \tag{7}$$

where $\det$ denotes the determinant, I is the identity matrix, $\mathcal{C} := Cov(z_{t-n})$ is the latent covariance matrix and $M_n := \sum_{i=0}^{n-1} \mathcal{K}^i \Sigma (\mathcal{K}^i)^\top$ is the $n-$step linear forward covariance (see detailed explanation and proof in Appendix F.5). We find that, from a Lagrangian perspective, maximizing $I(z_{t-n}; z_t)$ of Koopman representation under the finite variance constraint $\mathrm{tr}(\mathcal{C}) < \infty$ leads to a *water-filling allocation*: variance is distributed along the directions corresponding to the largest eigenvalues of $M_n^{-\frac{1}{2}} \mathcal{K}^n \mathcal{C} (\mathcal{K}^n)^\top M_n^{-\frac{1}{2}}$. These directions correspond to temporally coherent modes, which explains why higher latent mutual information enhances relevance. However, when the spectrum of this matrix is highly skewed, the water-filling solution degenerates into a low-rank allocation, *squeezing* information into only a few dominant directions. This effect reduces the effective dimension of the latent space $Z$, causing some spectral weights to vanish (cf. equation 43). To address the collapse induced by skewed spectral allocation, we next analyze how effective dimension can be preserved through entropy regularization.

> **Proposition 5 (Effective Dimension and Anti-Collapse)** *Low effective dimension (see Proposition 4) in Koopman representation indicates information collapse to few dominant modes and limits the model's ability to represent rich modes. Penalizing the von Neumann entropy $S(\frac{\mathcal{C}}{\mathrm{tr}(\mathcal{C})})$ encourages more expressive and spectrally diverse representations.*

Connecting to Proposition 4, Appendix F.6 contains a detailed proof via a water-filling and Lagrangian view. The normalized operator $\frac{\mathcal{C}}{\mathrm{tr}(\mathcal{C})}$ can be regarded as a density matrix in Hilbert space, and the effective dimension can be measured as $\exp(S)$ (see Definition E.2). When penalized with large the von Neumann entropy, the water-filling solution attains a non-zero allocation across all modes, preventing variance from collapsing entirely onto a few dominant directions (cf. equation 46). This ensures a positive distribution of spectral weights across all modes, thereby avoiding degenerate spectra and increasing the effective dimension of the latent space $Z$ (see Figure 1(c)).

## 3.3 INFORMATION-THEORETIC FORMULATION FOR PRACTICAL IMPLEMENTATION

The preceding analysis (Propositions 3, 4 and 5) reveals a fundamental trade-off in Koopman representation learning: maximizing latent mutual information enhances temporal coherence and predictive ability but risks mode collapse, whereas entropy regularization promotes spectral diversity for predictive sufficiency. Based on this principle, we formulate the following unified Lagrangian:

$$\max_z \quad \alpha \log I(z_{t-n}; z_t) - \beta I(z_t; x_t | z_{t-n}) + \gamma S\left(\frac{\mathcal{C}}{\mathrm{tr}(\mathcal{C})}\right) + \log p(x_t | z_t), \tag{8}$$

where $\alpha$, $\beta$ and $\gamma$ are Lagrangian multipliers. In equation 8, the first term in equation 7 preserves temporal-coherent information, the second term penalizes fast-dissipating or confounding components ($I(z_t; x_t | z_{t-n}) = I(z_t; x_{t-1} | z_{t-n}) + I(z_t; x_t | x_{t-1})$, see proof in equation 31), the third term

rewards larger von Neumann entropy of the normalized covariance to promote spectral diversity in the latent space $Z$. Lastly, $\log p(x_t|z_t)$ is the reconstruction terms from predicted latent variable $z_t$.

While the Lagrangian in equation 8 captures the desired information-theoretic trade-offs, it is not directly computable. To make it practical, we derive a tractable loss function for satisfying temporal coherence, predictive sufficiency and structural consistency

$$\max \sum_n \Big[ \underbrace{\alpha I(z_n; \mathcal{P}_n)}_{\text{Temporal coherence}} + \underbrace{\beta \mathbb{E}_{p_\theta(z_n|x_n)}[\log q_\psi(z_n|z_{n-1})]}_{\text{Structural consistency}} + \underbrace{\beta H_{p_\theta}(z_n|x_n)}_{\text{Encoder entropy}}$$
$$+ \underbrace{\log p_\omega(x_n|z_n)}_{\text{Reconstruction}} \Big] + \underbrace{\gamma S(\tfrac{\mathcal{C}}{\text{tr}(\mathcal{C})})}_{\text{Predictive sufficiency}} + \mathcal{L}_{\text{ELBO}}. \tag{9}$$

In VAE structure (shown in Figure 1(a)), each component of the loss plays a distinct role in balancing the information-theoretic objectives: (1) The mutual information $I(z_n; \mathcal{P}_n)$ captures temporal coherence by linking $z_n$ to its temporal neighborhood $\mathcal{P}_n = \{z_{n\pm i} \mid 1 \leq i \leq k\}$, which includes immediate past and future latent states; in practice, this can be computed either via the closed form in equation 7 for low-dimensional latents, or approximated by InfoNCE (Wu et al., 2020) for high-dimensional settings. (2) The term $-\mathbb{E}_{p_\theta(z_n|x_n)}[\log q_\psi(z_n|z_{n-1})] - H_{p_\theta}(z_n|x_n)$ serves as an equivalent representation of the conditional mutual information $I(z_t; x_t|z_{t-1})$, with linear Gaussian transition $q_\psi(z_n|z_{n-1}) = \mathcal{N}(z_n|\mathcal{K}_\psi z_{n-1}, \Sigma_\psi)$ and entropy of encoder $H_{p_\theta}(z_n|x_n)$ (see Appendix G.1). Minimizing this KL not only encourages the latent representation to capture information from the state $x_n$, but also compresses fast-dissipating and residual components, ensuring that the representation remains expressive yet simple. Here, $\mathbb{E}_{p_\theta(z_n|x_n)}[\log q_\psi(z_n|z_{n-1})]$ enforces structural consistency in latent space. (3) The term $\log p_\omega(x_n|z_n)$ is the decoder loss from predicted latent variable $z_n$. (4) von Neumann entropy term $S\left(\frac{\mathcal{C}}{\text{tr}(\mathcal{C})}\right)$ is computed from the normalized co-variance matrix $\mathcal{C} = \frac{1}{B}\sum_{i=1}^B (z_i - \bar{z})(z_i - \bar{z})^\top$ of the latent codes within a minibatch of size $B$. This promotes spectral diversity and guards against mode collapse, ensuring that the learned Koopman representation retains predictive sufficiency. (5) $\mathcal{L}_{\text{ELBO}}$ is the Evidence Lower Bound (ELBO) for training stability and reconstruction (see more analysis and implementation details in Appendix G.1). For AE structure, $\mathbb{E}_{p_\theta(z_n|x_n)}[\log q_\psi(z_n|z_{n-1})]$ degenerates into a $L^2$ loss enforcing the structural consistency $\|z_{n+1} - \mathcal{K}_\psi z_n\|^2$, and ELBO becomes AE reconstruction term (see G.2).

## 4 EXPERIMENTS

**Tasks.** We evaluate our approach across three types of dynamical data: (1) **Physical simulations**, including Lorenz 63, Kármán vortex street, Dam flow, and weather forecasting task (ERA5), which test the ability to capture nonlinear, stochastic and high-dimensional physical dynamics; (2) **Visual-input control**, including image-based Planar, Pendulum, Cartpole, and 3-Link manipulator, which evaluate the ability to extract latent dynamics from high-dimensional visual inputs while controllable in latent spaces; and (3) **Graph-structured dynamics prediction**, including Rope and Soft Robotics, which tests generalized abilities of latent representation on dynamics with graph structures (see experimental details in Appendix G.3).

**Metrics.** We assess performance on both **forecasting** and **control**. For forecasting, we report (i) normalized root mean square error (NRMSE) for short- and long-term predictions (for physical simulation and graphs-structured dynamics), (ii) physical consistency metrics based on spectral distribution errors based on $1000-$step sequences (SDEs), (iii) distributions of state measured by the Kullback–Leibler divergence (KLD), and (iv) structural similarity index (SSIM) for physical simulations. (v) the quality of low-dimensional manifold construction from high-dimensional visual inputs. For control, we measure the success rate of latent-space control of visual inputs following the setting in (Levine et al., 2020).

**Baseline Algorithms.** We compare against competitive baselines for each type of task. For **physical simulations**, we include VAE (Burgess et al., 2018), Koopman Autoencoder (KAE) (Pan et al., 2023), Koopman Kernel Regression (KKR) (Bevanda et al., 2023), and a SOTA Koopman variant for chaos - Poincaré Flow Neural Network (PFNN) (Cheng et al., 2025). For **visual-input control**,

we consider VAE-based representation learning methods, including Embed to Control (E2C) (Banijamali et al., 2019), as well as Prediction, Consistency and Curvature (PCC) (Levine et al., 2020), together with KAE. For **graph-structured dynamics**, we compare with Compositional Koopman Operator (CKO) (Li et al., 2020), the current SOTA method for graph-structured dynamics.

Table 2: Performance comparison of five algorithms on physical simulation tasks. PFNN is designed for chaotic dynamics and is thus not evaluated on Dam Flow. Here, $N$-NRMSE and $N$-SSIM denote errors at $N$ prediction steps, values in parentheses indicate variance, and SDE is the spectral distribution error. Best results are highlighted in bold with green, second best are shaded in blue.

| Task | Metric | VAE | KAE | KKR | PFNN | Ours |
|---|---|---|---|---|---|---|
| Lorenz 63 ($n = 3$) | 5-NRMSE | 0.005 (0.002) | 0.006 (0.003) | 0.004 (0.002) | 0.005 (0.003) | **0.003 (0.002)** |
| | 20-NRMSE | 0.011 (0.007) | 0.014 (0.009) | 0.009 (0.008) | 0.011 (0.007) | **0.007 (0.004)** |
| | 50-NRMSE | 0.019 (0.011) | 0.023 (0.013) | 0.017 (0.009) | 0.017 (0.007) | **0.013 (0.008)** |
| | KLD | 1.047 | 0.464 | 0.342 | 0.293 | **0.285** |
| Kármán Vortex ($n = 64 \times 64 \times 2$) | 5-NRMSE | 0.127 (0.005) | 0.149 (0.011) | 0.114 (0.065) | 0.075 (0.007) | **0.068 (0.006)** |
| | 20-NRMSE | 0.134 (0.003) | 0.195 (0.015) | 0.157 (0.057) | 0.125 (0.012) | **0.114 (0.015)** |
| | 50-NRMSE | 0.211 (0.018) | 0.233 (0.027) | 0.209 (0.028) | **0.137 (0.015)** | 0.138 (0.018) |
| | 5-SSIM | 0.743 (0.100) | 0.719 (0.030) | 0.868 (0.087) | 0.920 (0.030) | **0.936 (0.025)** |
| | 20-SSIM | 0.720 (0.079) | 0.586 (0.039) | 0.732 (0.086) | 0.800 (0.050) | **0.823 (0.047)** |
| | 50-SSIM | 0.539 (0.045) | 0.571 (0.037) | 0.581 (0.061) | **0.710 (0.030)** | 0.688 (0.020) |
| | SDE | 0.538 | 0.620 | 0.799 | 0.278 | **0.256** |
| Dam Flow ($n = 64 \times 64 \times 2$) | 5-NRMSE | 0.030 (0.001) | 0.037 (0.000) | 0.019 (0.003) | – | **0.018 (0.001)** |
| | 20-NRMSE | 0.033 (0.000) | 0.042 (0.000) | 0.028 (0.002) | – | **0.024 (0.001)** |
| | 50-NRMSE | 0.034 (0.000) | 0.046 (0.001) | 0.031 (0.002) | – | **0.026 (0.003)** |
| | 5-SSIM | 0.522 (0.021) | 0.419 (0.031) | 0.720 (0.034) | – | **0.760 (0.012)** |
| | 20-SSIM | 0.443 (0.007) | 0.282 (0.024) | 0.584 (0.025) | – | **0.627 (0.010)** |
| | 50-SSIM | 0.404 (0.005) | 0.176 (0.008) | 0.502 (0.010) | – | **0.577 (0.006)** |
| | SDE | 0.563 | 0.488 | 0.373 | – | **0.244** |
| ERA5 Weather (channel avg) | 5-NRMSE | – | 0.055 | 0.058 | 0.049 | **0.028** |
| | 10-NRMSE | – | 0.063 | 0.068 | 0.060 | **0.035** |
| | 50-NRMSE | – | 0.118 | 0.074 | 0.079 | **0.068** |
| | 5-SSIM | – | 0.666 | 0.664 | 0.697 | **0.867** |
| | 10-SSIM | – | 0.619 | 0.606 | 0.635 | **0.808** |
| | 50-SSIM | – | 0.481 | 0.707 | 0.695 | **0.781** |

**Result Analysis.** Our analysis is organized around the contributions established in propositions(Section 3), and we structure the discussion by addressing the following key questions. *(1) Does the latent mutual information determine the predictive limit of the Koopman representation? (Proposition 2)* – Yes. This is verified by the quantitative results of physical simulations in Table 2. Consistent with proposition, the prediction error under Koopman representation inevitably accumulates and is bounded by the latent mutual information. By regularizing with latent mutual information, both short- and long-term predictions are improved. Notably, PFNN (Cheng et al., 2025) is a SOTA model specifically designed with domain-specific knowledge, while our method, grounded in general information theory, achieves comparable performance on chaotic tasks (Lorenz 63 and Kármán vortex). Compared with other Koopman-based methods, our approach yields substantial improvements in both physical consistency and predictive accuracy.

*(2) How is the preserved information—particularly that associated with Koopman eigenmodes near the unit circle—shaped by latent mutual information and von Neumann entropy in constructing a dynamics-relevant manifold? (Proposition 4 and 5)* The preserved information manifests in Koopman modes with eigenvalues lying close to the unit circle, capturing the recurrent structure of the Kármán vortex limit cycle, as shown in Figure 2 (left). However, KAE suffers from some eigenvalues collapse toward zero, reducing the effective latent dimension. This collapse explains the drift observed in its autoregressive prediction. In contrast, our model captures the limit-cycle structure and produces stable autoregressive trajectories, consistently revolving around the true orbit (Figure 2, right). Baselines such as KKR and PFNN also capture limit-cycle structure (via one-step reconstruction) but gradually deviate from the correct trajectory over long horizons. By incorporating latent mutual information, we ensure that temporal-coherent information is retained, while von Neumann entropy prevents eigenvalue degeneration and preserves sufficient modes. Consequently, the information behind those modes can be preserved over long horizons, which directly translates into improved long-term prediction accuracy and statistical consistency, as also reported in Table 2.

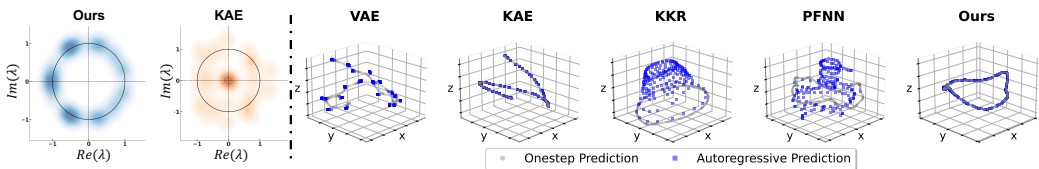

Figure 2: Eigenvalue comparison and manifold visualization of the Kármán vortex street. **Left:** Eigenvalue distributions of Koopman operators. **Right:** t-SNE visualization of learned latent manifolds for five methods on the Kármán vortex. The underlying dynamics is abstracted as a limit cycle.

*(3) How does explicit information-theoretic regularization sufficiently capture essential dynamics, compared with VAEs and Koopman autoencoders? (Proposition 4 and 5)* As shown in the reconstructed manifolds of Figure 3, our method produces a latent manifold that aligns most closely with the ground truth. For E2C, which is directly built on a VAE architecture, the latent geometry is heavily distorted (the loss of coherence). The manifold learned by KAE collapses into a nearly one-dimensional structure, reflecting the lack of effective dimensions in its latent space. PCC, a modified VAE-based method designed to improve manifold construction, demonstrates partial improvement but still exhibits a gap compared with our approach. By preserving both effective dimensionality and temporal coherence, our Koopman representation achieves the best average control performance in both noiseless and noisy environments (Table 8 and 9 in Appendix G.5.2).

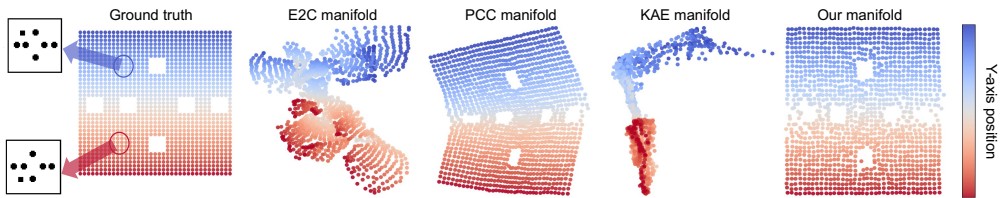

Figure 3: Latent manifolds of Planar visualized using locally linear embedding. The first subfigure shows the ground truth, while the second to fifth depict manifolds learned by different algorithms.

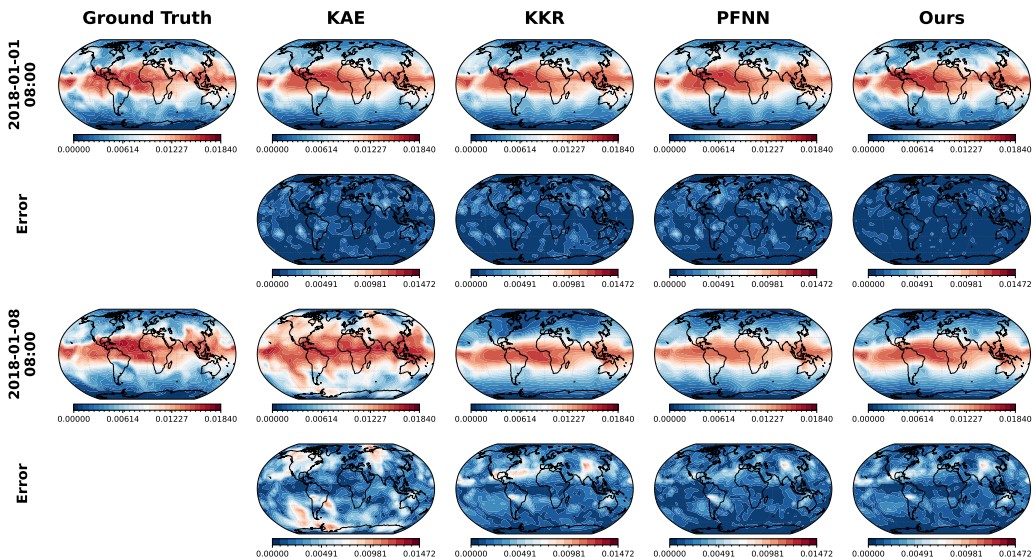

Figure 4: Comparison of continuous predictions for the global humidity starting from $2018 - 01 - 01 - 00 : 00$ to $2018 - 01 - 08 - 08 : 00$. Error maps in the lower panels demonstrate that, compared with other models, showing with more stable and accurate results of our model (see more demonstration in Appendix G.5.1).

*(4) How robust are the findings under noise, extended prediction horizons, and large-scale settings? (Proposition 1 and 2)* Our method remains robust under both noisy observations and extended prediction horizons. As shown in Table 2 and Figure 4, it maintains stable performance in long-term rollouts and physical statistics in large scale weather forecasting. Moreover, our approach supports control under noisy environments, achieving competitive performance. These quantitative results are consistent with our probabilistic propositions.

*(5) To what extent can our Lagrangian formulation be generalized to diverse architectures and adapted to support downstream tasks? (Proposition 1-5)* Our formulation demonstrates broad applicability: it consistently improves performance across physical simulations (see Table 2), visual perception tasks for manifold construction and control (see Figure 3, Tables 8 and 9 in Appendix G.5.2), and graph-structured dynamics prediction (see Figure 5). These gains indicate that the proposed Lagrangian principle is architecture-agnostic and can be readily incorporated into different settings to enhance both predictive accuracy and task effectiveness (more results are referred to Appendix G.5).

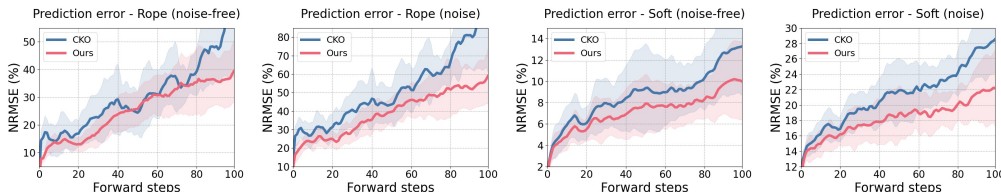

Figure 5: Comparison of prediction over 100 rollout steps. The left two figures show results for the Rope environment ($n \in [40, 56]$) with and without noise; the right two subfigures are the results for the Soft environment ($n \in [160, 224]$).

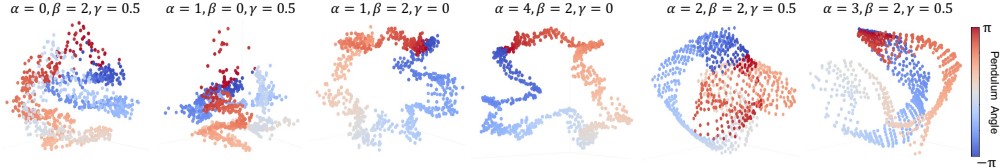

Figure 6: Ablation study on the pendulum task. Latent manifolds are learned from high-dimensional pendulum images, where the ground-truth phase space is isomorphic to $\mathcal{S}^1 \times \mathbb{R}$. Color represents the pendulum angle. Each subplot corresponds to removing or adjusting one regularization term: latent mutual information ($\alpha$), KL divergence ($\beta$), and von Neumann entropy ($\gamma$).

**Ablation Studies.** We analyze the effect of varying each Lagrangian multiplier to understand its role in shaping Koopman representation. In the pendulum task, the ground-truth phase space is $\mathcal{S}^1 \times \mathbb{R}$, consisting of a periodic angle and an angular velocity. The ablation study in Figure 6 illustrates how each regularization term contributes to recovering this manifold from high-dimensional visual inputs. Without mutual information regularization ($\alpha = 0$), temporal coherence is lost and the latent space degenerates into scattered points without geometric structure. Without structural consistency ($\beta = 0$), the latent manifold collapses, highlighting its role in enforcing the dynamics of Koopman representation. Removing the von Neumann entropy term ($\gamma = 0$) retains the circular $\mathcal{S}^1$ component but suppresses the $\mathbb{R}$ dimension, indicating the necessity of preserving effective dimensions. Increasing mutual information alone concentrates the representation on the $\mathcal{S}^1$ component (reflecting Proposition 4), while regularizing with von Neumann entropy yields a manifold that closely approximates the full $\mathcal{S}^1 \times \mathbb{R}$ structure. These observations align with the theoretical roles of the three penalties: temporal coherence, structural consistency and predictive sufficiency.

## 5 CONCLUSION

We presented a new perspective on Koopman representation by formulating it through an information-theoretic lens, leading to a general Lagrangian formulation that balances simplicity and expressiveness. Our analysis reveals the relationship between Koopman spectral properties and information in deep architectures. The proposed algorithm based on the Lagrangian formulation consistently improves the performance in a wide range of dynamical system tasks.

## ACKNOWLEDGMENTS

Zhuo Sun is supported by Fundamental Research Funds for the Central Universities 2025110590 of Shanghai University of Finance and Economics.

## ETHICS STATEMENT AND REPRODUCIBILITY STATEMENT

This work raises no specific ethical concerns beyond standard practices in machine learning research. All methods, datasets, and hyperparameters are described in detail, and core code is released in supplementary materials.

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

## THE USE OF LARGE LANGUAGE MODELS (LLMS)

During the preparation of this manuscript, the authors used ChatGPT to polish the writing (e.g., improving grammar, readability, and clarity). The content, technical contributions, and conclusions of the paper were developed entirely by the authors, who take full responsibility for all ideas and results presented.

## A NOTATION

Table 3: Notations in the Main Text

| Notations | Meaning |
|---|---|
| $d$ | latent dimension |
| $\det$ | determinant of matrix |
| $n$ | state dimension |
| $p$ | probability distribution |
| $p^{KR}$ | probability distribution of the Koopman-induced trajectory |
| $q^{KR}$ | variational approximation of the Koopman-induced trajectory |
| $t$ | time step |
| $\mathrm{tr}$ | trace of matrix |
| $x$ | state of dynamical systems |
| $z$ | latent variable of dynamical systems |
| $\mathcal{C}$ | latent covariance matrix |
| $\mathcal{H}$ | Hilbert space |
| $I$ | mutual information |
| $\mathrm{I}$ | identity matrix |
| $\mathcal{K}$ | Koopman operator |
| $D_{\mathrm{KL}}$ | KL divergence |
| $L^2(\mathcal{M}, \mu)$ | Lebesgue space equipped with inner product |
| $M_n$ | $n-$step linear forward covariance |
| $\mathcal{M}$ | finite-dimensional manifold |
| $\mathcal{N}$ | Gaussian distribution |
| $S$ | von Neumann entropy |
| $T$ | discrete-time nonlinear map of dynamics |
| $Z$ | latent space spanned by $\{\phi_1, \ldots, \phi_d\}$ |
| $\alpha, \beta, \gamma$ | Lagrangian multipliers |
| $\lambda$ | eigenvalues |
| $\rho$ | density matrix |
| $\phi$ | observable/feature |
| $\psi, \theta, \omega$ | parameters for neural networks in Koopman representation |

## B   A GENTLE INTRODUCTION TO APPENDIX

The appendix is organized to complement the main paper with precise definitions, detailed theoretical analysis, and additional experimental material. Given the information density of the appendix, we provide here a short roadmap to guide the reader:

- **Appendix C More Literature review.** We provide the classic to modern Koopman learning methods.
- **Appendix D Limitations and Future Directions.** We state the limitations of framework and propose some future directions for Koopman representation.
- **Appendix E. Technical definitions.** We collect the technical background, notations, and formal definitions used throughout the paper for ease of reference.
- **Appendix F. Theoretical analysis.** This section develops our theory step by step, where each proposition answers a natural "next question" in the following chain:

  **Q1.** *Will information be lost?* (Proposition 1)

  **Q2.** *If so, how much is lost?* (Proposition 2)

  **Q3.** *What kind of information is lost?* (Proposition 3)

  **Q4.** *How can we optimize information retention?* (Proposition 4)

  **Q5.** *How can we avoid negative side effects such as mode collapse?* (Proposition 5)

  In this way, the proofs form a coherent progression: each proposition is the answer to the next natural question raised by the previous one.

- **Appendix G. Experimental setup and additional results.** We provide implementation details, dataset descriptions, and supplementary results to support and validate the propositions made in the main text.

This structure ensures that readers can navigate the appendix according to their interests: consult Appendix A for notation, Appendix B for the full theoretical journey.

## C  MORE LITERATURE REVIEW RELATED TO KOOPMAN REPRESENTATION

The Koopman operator was originally introduced by Koopman and von Neumann as a linear embedding of Hamiltonian dynamical systems (Koopman, 1931; Koopman & Neumann, 1932). However, its infinite-dimensional nature makes it difficult to identify suitable handcrafted basis functions using conventional methods (Brunton et al., 2021). To address this, kernel techniques from functional analysis have been employed as bases for learning the Koopman operator (Das & Giannakis, 2020; Das et al., 2021; Kostic et al., 2022; Bevanda et al., 2025). Owing to the well-posed properties of kernel functions in reproducing kernel Hilbert spaces (e.g., linearity, existence, and convergence guarantees), the Koopman operator can be directly approximated via (extended) dynamic mode decomposition (DMD or EDMD) (Williams et al., 2015; Takeishi et al., 2017; Arbabi & Mezic, 2017; Xu et al., 2025a). Despite these theoretical advantages, fixed kernel functions are often too restrictive to capture a general function space (Berlinet & Thomas-Agnan, 2011; Alpay, 2012).

In contrast, deep learning frameworks provide a more flexible alternative: leveraging the universal approximation property of neural networks (Baker & Patil, 1998; Kidger & Lyons, 2020), they allow learning a general Koopman representation without relying on predefined kernels. Following this principle, (variational) autoencoder (AE/VAE) architectures have been widely adopted to extract features spanning the Koopman subspace (Liu et al., 2023; Wu et al., 2025; Xu et al., 2025b;c). The resulting latent representations are flexible and support downstream tasks such as prediction and control (Li et al., 2020; Mauroy et al., 2020; Korda & Mezić, 2020; Weissenbacher et al., 2022). However, these representations are typically learned in a purely self-supervised manner, lacking explicit grounding in dynamical systems theory. To improve their reliability, recent studies incorporate domain-specific priors—such as symmetry, conservation laws, dissipation, or ergodicity—into the Koopman representation (Vaidya & Mehta, 2008; Weissenbacher et al., 2022; Azencot et al., 2020; Cheng et al., 2025). Within the VAE setting, recent studies (Federici et al., 2023) have started to link Markovian dynamics and information theory, demonstrating that time-lagged tricks can exploit mutual information to obtain better latent representations. While existing approaches are effective for specific dynamical systems, a formal theoretical foundation for guiding the learning of Koopman representations remains insufficient. In this work, we investigate how general information-theoretic principles can be employed to fill this gap.

## D  LIMITATIONS AND FUTURE DIRECTIONS

A current limitation of our framework is that it does not address the sample complexity or non-asymptotic convergence of the Koopman representation; future work could explore more rigorous theoretical analyses in this direction. In addition, recent studies have highlighted connections between kernel methods (Kostic et al., 2022; 2023a; 2024) and information theory (Bach, 2022), suggesting an interesting avenue for extending conventional kernel techniques in Koopman theory through an information-theoretic perspective.

## E    KEY TECHNICAL DEFINITIONS AND RELATED PROPERTIES

**Definition E.1 (Density Matrix (Bach, 2022))** *A density matrix $\rho \in \mathbb{R}^{d \times d}$ is a real symmetric matrix satisfying:*

- *$\rho$ is positive semi-definite: $\rho \succeq 0$*

- *The trace of $\rho$ is 1: $\mathrm{tr}(\rho) = 1$*

*Such a matrix can be interpreted as a probability-weighted combination of orthonormal directions in $\mathbb{R}^d$. It admits a spectral decomposition:*

$$\rho = \sum_{i=1}^{d} p_i v_i v_i^{\top}, \quad \text{where } p_i \geq 0, \sum_{i=1}^{d} p_i = 1, \text{ and } v_i \in \mathbb{R}^d \text{ with } \|v_i\| = 1.$$

**Definition E.2 (Effective Dimension (Roy & Vetterli, 2007))** *Given a density matrix $\rho$ on a Hilbert space, the* effective dimension *is defined as*

$$d_{\mathrm{eff}}(\rho) := \exp\big(S(\rho)\big),$$

*where $S(\rho) = -\mathrm{Tr}(\rho \log \rho)$ is the von Neumann entropy of $\rho$.*

The *effective dimension* measures how many directions in a representation space are substantially used. Given a symmetric, positive semi-definite matrix $\rho$ with unit trace, its von Neumann entropy

$$S(\rho) = -\mathrm{tr}(\rho \log \rho)$$

quantifies the spectral diversity of $\rho$. The effective dimension is then defined as

$$d_{\mathrm{eff}}(\rho) = \exp\big(S(\rho)\big),$$

so that $d_{\mathrm{eff}}(\rho)$ can be interpreted as the number of dimensions effectively occupied by the latent variable. In particular, $d_{\mathrm{eff}}(\rho) = 1$ when $\rho$ is concentrated on a single direction (pure state in quantum mechanics), while $d_{\mathrm{eff}}(\rho) = d$ when $\rho$ is maximally mixed and spreads uniformly over all $d$ directions. A higher effective feature dimension is often required to ensure predictive sufficiency.

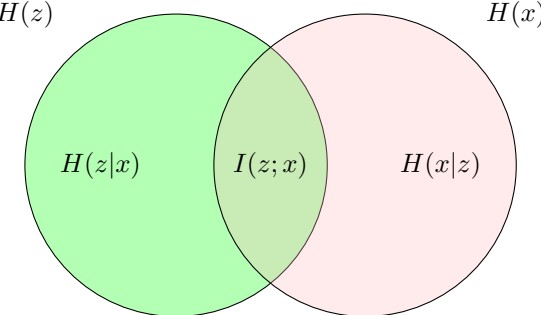

Figure 7: A Venn diagram illustrating entropy, conditional entropy, and mutual information between the true state $x$ and latent variable $z$. $H(x)$ and $H(z)$ denote the Shannon entropy (total information) of $x$ and $z$, respectively. Their symmetric overlap, $I(z; x)$, represents the mutual information that quantifies how much information about the true dynamics is preserved in the Koopman representation. The non-overlapping regions, $H(x|z)$ and $H(z|x)$, correspond to the residual uncertainty not captured by $I(z; x)$.

**Definition E.3 (Entropy, Mutual Information and Conditional Mutual Information)** *Let $x, y, z$ be random variables. Beyond the Definition 2.2 in the main text, we give a standard definition of (conditional) mutual information based on Shannon entropy (shown in Figure 7, (Csiszár et al., 2004)).*

- ***D1 Entropy** of a random variable $x$ is defined as*

$$H(x) := -\int p(x) \log p(x) dx.$$

- **D2 Mutual Information** *is defined as*

$$I(z;x) := H(z) + H(x) - H(z,x),$$

*equivalently,*

$$I(z;x) = H(x) - H(x|z) = H(z) - H(z|x).$$

*It can also be expressed in terms of the Kullback–Leibler (KL) divergence:*

$$I(x;y) = D_{\mathrm{KL}}\big(p(x,y) \,\big\|\, p(x)p(y)\big),$$

*where*

$$D_{\mathrm{KL}}(p(x)\|q(x)) := \int p(x) \log \frac{p(x)}{q(x)} dx.$$

- **D3. Conditional Mutual Information** *is defined as*

$$I(x;y \mid z) := H(x \mid z) + H(y \mid z) - H(x,y \mid z),$$

*equivalently,*

$$I(x;y \mid z) = H(x \mid z) - H(x \mid y,z).$$

*In terms of KL divergence,*

$$I(x;y \mid z) = \mathbb{E}_z\Big[D_{\mathrm{KL}}\big(p(x,y \mid z) \,\big\|\, p(x \mid z)\,p(y \mid z)\big)\Big]$$

$$= \mathbb{E}_z\Big[D_{\mathrm{KL}}\big(p(x \mid y,z) \,\big\|\, p(x \mid z)\big)\Big].$$

# F    THEORETICAL FRAMEWORK

To ground our theoretical analysis, we first formalize the autoregressive structure of Koopman representations illustrated in Figure 8. The original dynamics evolve as a nonlinear transformation

$$x_{t-n} \to x_{t-n+1} \to \cdots \to x_t.$$

In parallel, states are *encoded* into latent variables $z_{t-n}$, which propagate linearly under the Koopman operator $\mathcal{K}$ and are subsequently *decoded* back to approximate the original states. This two-layer structure makes clear where information may dissipate: (i) during encoding from $x$ to $z$, (ii) along the linear latent evolution governed by $\mathcal{K}$, and (iii) during reconstruction from $z$ to $x$. Analyzing this flow of information is therefore essential for understanding the fundamental limits of Koopman representations, and the proofs of the following propositions will be developed around this structure.

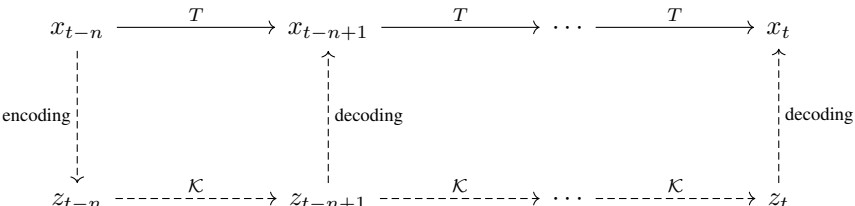

Figure 8: The autoregressive structure of Koopman representation. **Top row (solid arrows):** the original states $x_{t-n}$ evolve under the nonlinear map $T$. **Bottom row (dashed arrows):** the states are first *encoded* into latent variables $z_{t-n}$, which then evolve linearly under the Koopman operator $\mathcal{K}$. The latent variables are subsequently *decoded* back to approximate the original states. Thus, the latent evolution under Koopman representation captures essential structure but do not directly contain the full state information. Note: the dashed arrows represent the approximated Koopman representation in latent space, whereas the solid arrows denote the true underlying dynamics.

**Fact F.1** *As shown in Figure 8, we set that the latent variable $z_t$ is obtained via a probabilistic encoder that depends only on the current true state $x_t$, i.e., $p(z_t \mid x_t)$. Consequently, the information content of $z_t$ cannot exceed that of $x_t$, so $H(z_t) \leq H(x_t)$ (due to the data processing inequality (Stone, 2024)). Moreover, under this setting, $z_t$ is conditionally independent of any other variable in the dynamical system given $x_t$, i.e.,*

$$I(z_t; \square \mid x_t) = 0, \quad \text{or equivalently} \quad \square \perp\!\!\!\perp z_t \mid x_t,$$

*where $\square$ denotes any variable in the dynamical system.*

## F.1    PROOF OF PROPOSITION 1

Beyond establishing Proposition F.1, it is crucial to highlight the phenomenon of *information dissipation* in the autoregressive Koopman representation (see Figure 8) and to clarify how the Koopman operator $\mathcal{K}$ connects to the *information limit*. By a fundamental principle of information theory, a compressed representation can never increase the available information. While this observation yields a one-step inequality 3, our analysis extends it to the multi-step analysis, where the cumulative effect of encoding, autoregressive latent evolution via $\mathcal{K}$, and decoding can be rigorously tracked. This extension makes explicit how information gradually dissipates at each stage of the Koopman representation, ultimately leading to the error accumulation.

**Lemma F.2 (Chain Rule of Information)** *For variables $x, y, z$, the mutual information with their joint variable satisfies*

$$I(x; y) = I(x; (y, z)) - I(x; z \mid y),$$

*where $(y, z)$ is treated as the joint random variable with distribution $p(y, z)$. According to the non-negativity of conditional mutual information, it is obvious that*

$$I(x; (y, z)) \geq I(x; y).$$

**Proof 1** *We prove this proposition as two steps.*

**Step 1.** *According to the autoregressive structure of the Koopman representation, the first inequality 3 follows directly from the data processing inequality (Stone, 2024). Since the latent variable $z_{t-1}$ is a compressed representation of $x_{t-1}$, the mutual information between successive states cannot exceed that induced by the original dynamics $T$. Formally,*

$$I(x_{t-1}; x_t) \geq I(z_{t-1}; x_t). \tag{10}$$

*This can be derived based on the Fact F.1, it can be factorized as*

$$
\begin{aligned}
I(x_{t-1}; x_t) &= I((x_{t-1}, z_{t-1}); x_t) - I(z_{t-1}; x_t|x_{t-1}) \quad &\textit{(Lemma F.2)} \\
&= I((x_{t-1}, z_{t-1}); x_t) - \underline{I(z_{t-1}; x_t|x_{t-1})} \quad &\textit{(Fact F.1)} \\
&= I(z_{t-1}; x_t) + I(x_{t-1}; x_t|z_{t-1}) \quad &\textit{(Lemma F.2)} \\
&\geq I(z_{t-1}; x_t). \quad &\textit{(non-negativity of mutual information)}
\end{aligned}
\tag{11}
$$

*Thus, the first inequality is derived.*

**Step 2.** *In terms of seconding inequality, we derive it as follow*

$$
\begin{aligned}
I(z_{t-1}; x_t) &= I(z_{t-1}; (x_t, z_t)) - I(z_{t-1}; z_t|x_t) \quad &\textit{(Lemma F.2)} \\
&= I(z_{t-1}; (x_t, z_t)) - \underline{I(z_{t-1}; z_t|x_t)} \quad &\textit{(Fact F.1)} \\
&= I(z_{t-1}; z_t) + I(z_{t-1}; x_t|z_t) \quad &\textit{(Lemma F.2)} \\
&\geq I(z_{t-1}; z_t). \quad &\textit{(non-negativity of mutual information)}
\end{aligned}
\tag{12}
$$

*Here, the proof for the proposition ends.*

Beyond the proposition, we would like to disentangle the what information is lost during the Koopman representation. Combining equations 11 and 12, we obtain

$$
\begin{aligned}
I(x_{t-1}; x_t) &= I(z_{t-1}; x_t) + I(x_{t-1}; x_t|z_{t-1}) \\
&= I(z_{t-1}; z_t) + I(z_{t-1}; x_t|z_t) + I(x_{t-1}; x_t|z_{t-1}),
\end{aligned}
\tag{13}
$$

then,

$$
I(x_{t-1}; x_t) - I(z_{t-1}; z_t) = \underbrace{I(z_{t-1}; x_t|z_t) + I(x_{t-1}; x_t|z_{t-1})}_{\text{step-wise information gap in Koopman representation}}.
\tag{14}
$$

Furthermore, we extend from one-step mutual information to multi-step ones (with $n \geq 1$), such that

$$
\begin{aligned}
I(x_{t-n}; x_t) &= I((x_{t-n}, z_{t-n}); x_t) - I(z_{t-n}; x_t|x_{t-n}) \quad &\text{(Lemma F.2)} \\
&= I((x_{t-n}, z_{t-n}); x_t) \quad &\text{(Fact F.1)} \\
&= I(z_{t-n}; x_t) + I(x_{t-n}; x_t|z_{t-n}) \quad &\text{(Lemma F.2)} \\
&= I(z_{t-n}; z_t) + I(z_{t-n}; x_t|z_t) + I(x_{t-n}; x_t|z_{t-n}). \quad &\text{(repeating previous procedure)}
\end{aligned}
\tag{15}
$$

Also, $I(x_{t-n}; x_t|z_{t-n})$ can be disentangled as follows:

**Step 1 — Introduce $z_{t-n+1}$**  Apply Lemma F.2:

$$I(x_{t-n}; x_t \mid z_{t-n}) = I(x_{t-n}; z_{t-n+1} \mid z_{t-n}) + I(x_{t-n}; x_t \mid z_{t-n}, z_{t-n+1}).$$

Here, the graphical structure ensures that:

$$I(x_{t-n}; z_{t-n+1} \mid x_t, z_{t-n}) = 0,$$

so there is no correction term.

**Step 2 — Introduce $z_{t-n+2}$.** Expand the second term via Lemma F.2:

$$
\begin{aligned}
I(x_{t-n}; x_t \mid z_{t-n}, z_{t-n+1}) = &\, I(x_{t-n}; z_{t-n+2} \mid z_{t-n}, z_{t-n+1}) \\
&+ I(x_{t-n}; x_t \mid z_{t-n}, z_{t-n+1}, z_{t-n+2}).
\end{aligned}
$$

Again, the graphical structure implies:

$$I(x_{t-n}; z_{t-n+2} \mid x_t, z_{t-n}, z_{t-n+1}) = 0.$$

**Step 3 — Repeat recursively.** For each $i = t - n + 1,\, t - n + 2,\, \ldots,\, t$:

$$I(x_{t-n}; x_t \mid z_{t-n:i-1}) = I(x_{t-n}; z_i \mid z_{t-n:i-1}) + I(x_{t-n}; x_t \mid z_{t-n:i}),$$

where

$$z_{t-n:i} := (z_{t-n}, z_{t-n+1}, \ldots, z_i).$$

Equivalently, in compact notation:

$$I(x_{t-n}; x_t | z_{t-n}) = \sum_{i=t-n+1}^{t} I(x_{t-n}; z_i | z_{t-n:i-1}) + I(x_{t-n}; x_t | z_{t-n:t}). \tag{16}$$

Thus, by combining equations 15 and 16, the multi-step lost information becomes

$$
\begin{aligned}
& I(x_{t-n}; x_t) - I(z_{t-n}; z_t) \\
&= \sum_{i=t-n+1}^{t} I(x_{t-n}; z_i | z_{t-n:i-1}) + I(x_{t-n}; x_t | z_{t-n:t}) + I(z_{t-n}; x_t | z_t).
\end{aligned}
\tag{17}
$$

Here, we interpret the physical meaning of the three parts in Koopman representation. Each term $I(x_{t-n}; z_i | z_{t-n:i-1})$ measures how much new information about past state $x_{t-n}$ revealed by the latent variable the latent variable $z_i$, given all previous latent states. Physically, this corresponds to the fast-dissipating modes that decay quickly; as more latent steps are added, this residual information diminishes rapidly to zero.

The second term $I(x_{t-n}; x_t | z_{t-n:t})$ measures the residual dependency between the past and the current state that cannot be fully represented the latent sequence $\{z_{t-n:t}\}$ due to the compressed representation.

The quantity $I(z_{t-n}; x_t | z_t)$ measures the information about the state $x_t$ that remains in the past latent variable $z_{t-n}$ but is not preserved in the current latent $z_t$. A positive value therefore indicates information loss during latent evolution. This phenomenon arises from Koopman modes with eigenvalues $|\lambda| < 1$, whose contributions decay over time and thus dissipate predictive information in the Koopman representation.

## F.2 Derivation of Variational Distribution

The derivation of discrepancy between true and Koopman-induced trajectories is listed as follow:

$$
\begin{aligned}
& D_{\mathrm{KL}}\left(p(x_{1:t}|x_0) \,\|\, q^{KR}(x_{1:t}|x_0)\right) \\
=& \mathbb{E}\Big[\log \frac{p(x_{1:t}|x_0)p^{KR}(x_{1:t}|x_0)}{p^{KR}(x_{1:t}|x_0)q^{KR}(x_{1:t}|x_0)}\Big] \\
=& \mathbb{E}\Big[\log \frac{p(x_{1:t}|x_0)}{p^{KR}(x_{1:t}|x_0)}\Big] + \mathbb{E}\Big[\log \frac{p^{KR}(x_{1:t}|x_0)}{q^{KR}(x_{1:t}|x_0)}\Big] \\
\leq& \mathbb{E}\Big[\log \frac{p(x_{1:t}|x_0)}{p^{KR}(x_{1:t}|x_0)}\Big] + \mathbb{E}\Big[\log \frac{p^{KR}(z_{0:t}, x_{1:t}|x_0)}{q^{KR}(z_{0:t}, x_{1:t}|x_0)}\Big] \\
=& \mathbb{E}\Big[\log \frac{p(x_{1:t}|x_0)}{p^{KR}(x_{1:t}|x_0)}\Big] + \mathbb{E}\Big[\log \frac{p(z_0|x_0)\prod_{n=1}^{t} p(z_n|z_{n-1})p(x_n|z_n)}{q^{KR}(z_0|x_0)\prod_{n=1}^{t} q^{KR}(z_n|z_{n-1})q^{KR}(x_n|z_n)}\Big] \\
=& \mathbb{E}\Big[\log \frac{p(x_{1:t}|x_0)}{p^{KR}(x_{1:t}|x_0)}\Big] + \mathbb{E}\Big[\log \frac{p(z_0|x_0)}{q^{KR}(z_0|x_0)}\Big] \\
& \qquad + \sum_{n=1}^{t} \mathbb{E}\Big[\log \frac{p(z_n|z_{n-1})}{q^{KR}(z_n|z_{n-1})}\Big] + \mathbb{E}\Big[\log \frac{p(x_n|z_n)}{q^{KR}(x_n|z_n)}\Big] \\
=& \underbrace{D_{\mathrm{KL}}\left(p(x_{1:t}|x_0)\,\|\,p^{KR}(x_{1:t}|x_0)\right)}_{\text{discrepancy between true and Koopman-induced distributions}} + \underbrace{D_{\mathrm{KL}}\left(p(z_0|x_0)\,\|\,q^{KR}(z_0|x_0)\right)}_{\text{latent representation error, } \mathcal{E}_{\text{enc}}} \\
& + \sum_{n=1}^{t} \underbrace{D_{\mathrm{KL}}\left(p(z_n|z_{n-1})\,\|\,q^{KR}(z_n|z_{n-1})\right)}_{\text{Koopman operator error, } \mathcal{E}_{\text{tra}}} + \underbrace{D_{\mathrm{KL}}\left(p(x_n|z_n)\,\|\,q^{KR}(x_n|z_n)\right)}_{\text{reconstruction error, } \mathcal{E}_{\text{rec}}}.
\end{aligned}
\tag{18}
$$

Here, $q^{KR}$ denotes the variational approximation. For notational convenience, we denote the last three terms in equation 18 by $\mathcal{E}_{\text{enc}}$, $\mathcal{E}_{\text{tra}}$, and $\mathcal{E}_{\text{rec}}$, respectively. Also, the logic from third line to fourth line holds since the inequality follows from the fact that marginalization cannot increase KL divergence.

## F.3 Proof of Proposition 2

Before proving Proposition 2, we first introduce a technical lemma.

**Lemma F.3 (Pinsker's Inequality (Yeung, 2008))** *For any two probability distributions $p$ and $q$ over the same space $X$, the total variation distance*

$$
\|p - q\|_{TV} := \sup_X |p(X) - q(X)|
$$

*is equal to one half of their $L^1$ distance:*

$$
\|p - q\|_{TV} = \tfrac{1}{2}\int |p(x) - q(x)|\, dx.
$$

*Moreover, it is bounded by the Kullback–Leibler divergence:*

$$
\|p - q\|_{TV} = \tfrac{1}{2}\int |p(x) - q(x)|\, dx \leq \sqrt{\tfrac{1}{2}D_{\mathrm{KL}}(p\,\|\,q)}.
$$

**Proof 2** *The proof proceeds in four steps: (1) establish the connection between KL divergence and total variation distance, (2) relate KL divergence to latent mutual information, (3) derive the upper error bound via information-theoretic limits, and (4) show that the lower bound decays exponentially with increasing latent mutual information.*

**Step 1.** *By applying Pinsker's inequality (Lemma F.3) and equation 18, we can directly bound the distributional discrepancy as*

$$\|p(x_{1:t}|x_0) - q^{KR}(x_{1:t}|x_0)\|_{TV} \leq \sqrt{\frac{1}{2}\left[D_{\mathrm{KL}}\left(p(x_{1:t}|x_0) \,\|\, p^{KR}(x_{1:t}|x_0)\right) + \mathcal{E}_{enc} + \mathcal{E}_{tra} + \mathcal{E}_{rec}\right]}. \tag{19}$$

**Step 2.** *The connection between mutual information and KL divergence is given as follows:*

$$
\begin{aligned}
&I(x_{t-1}; x_t) - I(z_{t-1}; x_t) \\
=&\mathbb{E}[\log \frac{p(x_t|x_{t-1})}{p(x_t)}] - \mathbb{E}[\log \frac{p(z_t|z_{t-1})}{p(z_t)}] \quad \text{(Definition 2.2)} \\
=&\mathbb{E}[\log \frac{p(x_t|x_{t-1})}{p(x_t)}] + \mathbb{E}[\log \frac{p(z_t)}{p(z_t|z_{t-1})}] \\
=&\mathbb{E}[\log \frac{p(x_t|x_{t-1})p(z_t)}{p(x_t)p(z_t|z_{t-1})}] \\
=&\mathbb{E}[\log \frac{p(x_t|x_{t-1})}{p(z_t|z_{t-1})} \cdot \frac{\frac{p(x_t,z_t)}{p(x_t|z_t)}}{\frac{p(x_t,z_t)}{p(z_t|x_t)}}] \quad \text{(Bayes' rule)} \\
=&\mathbb{E}[\log \frac{p(x_t|x_{t-1})}{p(z_t|z_{t-1})} \cdot \frac{p(z_t|x_t)}{p(x_t|z_t)}].
\end{aligned}
\tag{20}
$$

*By recursively using the result in equation 20, we can summation the results as*

$$
\begin{aligned}
&\sum_{n=1}^{t} \left( I(x_{n-1}; x_n) - I(z_{n-1}; z_n) \right) \\
=&\sum_{n=1}^{t} \mathbb{E}[\log \frac{p(x_n|x_{n-1})}{p(z_n|z_{n-1})} \cdot \frac{p(z_n|x_n)}{p(x_n|z_n)}] \\
=&\mathbb{E}[\log \frac{p(z_0|x_0) \prod_{n=1}^{t} p(x_n|x_{n-1})}{p(z_0|x_0) \prod_{n=1}^{t} p(z_n|z_{n-1})} \cdot \frac{\prod_{n=1}^{t} p(z_n|x_n)}{\prod_{n=1}^{t} p(x_n|z_n)}].
\end{aligned}
\tag{21}
$$

*Here, $p(x_n|x_{n-1})$ and $p(z_n|z_{n-1})$ are governed by the original nonlinear dynamics $T$ and Koopman operator $\mathcal{N}(z_t|\mathcal{K}z_{t-1}, \Sigma)$, respectively. Based on this fact, we can further to develop equation 21 as*

$$
\begin{aligned}
&\mathbb{E}[\log \frac{p(z_0|x_0) \prod_{n=1}^{t} p(x_n|x_{n-1})}{p(z_0|x_0) \prod_{n=1}^{t} p(z_n|z_{n-1})} \cdot \frac{\prod_{n=1}^{t} p(z_n|x_n)}{\prod_{n=1}^{t} p(x_n|z_n)}] \\
=&\mathbb{E}[\log \frac{p(z_{0:t}, x_{1:t})}{p^{KR}(z_{0:t}, x_{1:t})}] \\
\geq&\mathbb{E}[\log \frac{p(x_{1:t})}{p^{KR}(x_{1:t})}] \\
=&D_{\mathrm{KL}}\left(p(x_{1:t}|x_0) \,\|\, p^{KR}(x_{1:t}|x_0)\right).
\end{aligned}
\tag{22}
$$

*Plugging equation 22 into equation 19 in Step 1, we have*

$$
\begin{aligned}
&\left\|p(x_{1:t}|x_0) - q^{KR}(x_{1:t}|x_0)\right\|_{TV} \\
\leq&\sqrt{\frac{1}{2}\left[D_{\mathrm{KL}}\left(p(x_{1:t}|x_0) \,\|\, p^{KR}(x_{1:t}|x_0)\right) + \mathcal{E}_{enc} + \mathcal{E}_{tra} + \mathcal{E}_{rec}\right]} \\
\leq&\sqrt{\frac{1}{2}\sum_{n=1}^{t}(I(x_{n-1}; x_n) - I(z_{n-1}; z_n)) + \mathcal{E}_{enc} + \mathcal{E}_{tra} + \mathcal{E}_{rec}}.
\end{aligned}
\tag{23}
$$

**Step 3.** *Based on the distributional discrepancy in equation 23, we have the following inequality*

$$\left\|\mathbb{E}_{q^{KR}}[x_{1:t} \mid x_0] - \mathbb{E}_p[x_{1:t} \mid x_0]\right\|_2$$

$$= \left\|\int x_{1:t} \, dq^{KR}(x_{1:t} \mid x_0) - \int x_{1:t} \, dp(x_{1:t} \mid x_0)\right\|_2 \qquad \textit{(Lebesgue measure)}$$

$$\leq \left\|x_{1:t}\right\|_\infty \underbrace{\int \left|q^{KR}(x_{1:t} \mid x_0) - p(x_{1:t} \mid x_0)\right| dx_{1:t}}_{L^1 \ distance} \qquad \textit{(triangle inequality)}$$

$$\leq 2\bar{C} \left\|p(x_{1:t} \mid x_0) - q^{KR}(x_{1:t} \mid x_0)\right\|_{TV} \qquad \textit{(Lemma F.3)}$$

$$\leq \bar{C} \sqrt{2 D_{\mathrm{KL}}(p(x_{1:t} \mid x_0) \,\|\, q^{KR}(x_{1:t} \mid x_0))} \qquad \textit{(Pinsker's inequality)}$$

$$\leq \bar{C} \sqrt{2 \sum_{n=1}^{t} \big(I(x_{n-1}; x_n) - I(z_{n-1}; z_n)\big) + \mathcal{E}_{enc} + \mathcal{E}_{tra} + \mathcal{E}_{rec}}, \qquad \textit{(via equation 19)}.$$

*where the state $x$ lies in a compact space $\mathcal{M}$ with a complete metric, ensuring $\|x_{1:t}\|_\infty \leq \bar{C} < \infty$. The proof ends.*

**Step 4.** *The classical rate-distortion theorem (Cover, 1999) states that $x_{1:t} \in \mathbb{R}^{n \times t}$, under $L^2$ error distortion via the ideal Koopman model $p^{KR}$, the minimal achievable distortion $D$ is bounded by*

$$D \geq \frac{nt}{2\pi e} \exp(\frac{2}{nt} H(x_{1:t})) \cdot \exp(-\frac{2}{nt} \sum_{n=1}^{t} I(z_{n-1}; z_n)). \qquad (24)$$

*Since the entropy $H(x_{1:t})$ can be totally measured by the mutual information $\sum_{n=1}^{t} I(x_{n-1}; x_n)$ of original dynamics $T$. Then the accumulative mean-squared error after $t$ steps given $x_0$ is bounded below as*

$$\mathbb{E}_p\big[\|x_{1:t} - \mathbb{E}_{q^{KR}}[x_{1:t}|x_0]\| \,|\, x_0\big] \geq \underline{C} \exp(-\frac{2}{nt}(\sum_{n=1}^{t} I(z_{n-1}; z_n) + \mathcal{E}_{enc} + \mathcal{E}_{tra} + \mathcal{E}_{rec})), \quad (25)$$

*where the constant $\underline{C} = \frac{nt}{2\pi e} \exp(\frac{2}{nt} \sum_{n=1}^{t} I(x_{n-1}; x_n))$ absorbs the marginal entropy of the trajectory.*

**Remark F.4** *A special case of Proposition 2 is ergodic system, conditioning on $x_0$ and then taking the long-time average*

$$\lim_{t \to \infty} \frac{1}{t} D_{KL}\big(p(x_{1:t} \mid x_0) \,\|\, q^{KR}(x_{1:t} \mid x_0)\big) \leq \lim_{t \to \infty} \frac{1}{t} \sum_{n=1}^{t} \big(I(x_{n-1}, x_n) - I(z_{n-1}, z_n)\big),$$

*which follows from Proposition 2. The left-hand side is the* relative entropy rate*, and the inequality shows that the dynamic discrepancy between $p$ and $q^{KR}$ can be controlled by the per-step information difference.*

### F.4 PROOF OF PROPOSITION 3

Beyond establishing Proposition 3, it is even more important to clarify the connection between spectral theory and the information components of the Koopman representation. Before proceeding to the detailed proof, we first derive the closed-form expression of latent mutual information under the Koopman representation. This will allow us to interpret the spectral properties of the Koopman operator from an information-theoretic perspective.

For one-step forward under Koopman representation, we have

$$z_t = \mathcal{K} z_{t-1} + \epsilon_{t-1}, \quad \epsilon_{t-1} \sim \mathcal{N}(0, \Sigma).$$

For multi-step forward, it can be recursively derived as

$$z_t = \mathcal{K}(\mathcal{K} z_{t-2} + \epsilon_{t-2}) + \epsilon_{t-1},$$

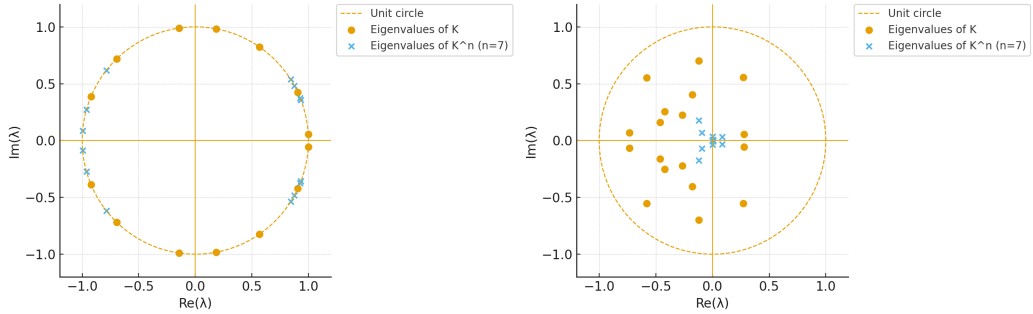

Figure 9: Spectral behavior of the Koopman operator under different regimes. **Left:** Eigenvalues of $\mathcal{K}$ (orange dots) lie on the complex unit circle ($|\lambda| = 1$), and those of $\mathcal{K}^n$ with $n = 7$ (blue crosses) remain on the unit circle, indicating temporal coherence and preservation of information. **Right:** Eigenvalues of $\mathcal{K}$ lie strictly inside the complex unit circle ($|\lambda| < 1$), and the spectrum of $\mathcal{K}^n$ contracts toward the origin as $n$ increases, reflecting fast mixing and information dissipation.

then,

$$z_t = \mathcal{K}^n z_{t-n} + \sum_{i=0}^{n-1} \mathcal{K}^i \epsilon_{t-i}.$$

Here, $\epsilon_{t-i} \sim \mathcal{N}(0, \Sigma)$ is a time-independent Gaussian distribution for all $i$. Therefore, $z_t$ follows the distribution as $\mathcal{N}(\mathcal{K}^n z_{t-n}, \sum_{i=0}^{n-1} \mathcal{K}^i \Sigma (\mathcal{K}^i)^\top)$. For convenience, we denote the covariance matrix as

$$M_n := \sum_{i=0}^{n-1} \mathcal{K}^i \Sigma (\mathcal{K}^i)^\top, \qquad \text{with } z_t \sim \mathcal{N}(\mathcal{K}^n z_{t-n}, M_n) \tag{26}$$

Without loss of generality, we can set $z_{t-n} \sim \mathcal{N}(0, \mathcal{C})$ and the covariance matrix is denoted as $\mathcal{C} := Cov(z_{t-n})$. Given the latent variable $z_{t-n}$, the conditional entropy $H(z_t | z_{t-n})$ is calculated as

$$H(z_t | z_{t-n}) = \frac{1}{2} \log[(2\pi e)^d \det M_n]. \tag{27}$$

On the other hand, the entropy $H(z_t)$ is calculated as

$$H(z_t) = \frac{1}{2} \log[(2\pi e)^d \det \left(\mathcal{K}^n \mathcal{C}(\mathcal{K}^n)^\top + M_n\right)]. \tag{28}$$

**Proof 3** *The proof of this proposition proceeds in three steps: (1) we first interpret latent mutual information in relation to the spectral properties of the Koopman representation; (2) we disentangle the mutual information $I(z_t; x_t)$ to clarify the role of each component and its associated spectral behavior; and (3) we derive the closed-form expression of mutual information under the Koopman representation, which highlights how the Koopman operator governs the information flow.*

***Step 1. Spectral Properties in Latent Mutual Information.*** *Based on the Definition E.3, mutual information $I(z_{t-n}, z_t)$ is calculated via equations 26, 27 and 28 as*

$$\begin{aligned}
I(z_{t-n}, z_t) &= H(z_t) - H(z_t | z_{t-n}) \\
&= \frac{1}{2} \log[(2\pi e)^d \det \left(\mathcal{K}^n \mathcal{C}(\mathcal{K}^n)^\top + M_n\right)] - \frac{1}{2} \log[(2\pi e)^d \det M_n] \\
&= \frac{1}{2} \log \frac{\cancel{(2\pi e)^d} \det \left(\mathcal{K}^n \mathcal{C}(\mathcal{K}^n)^\top + M_n\right)}{\cancel{(2\pi e)^d} \det M_n} \\
&= \frac{1}{2} \log \det(\mathrm{I} + M_n^{-\frac{1}{2}} (\mathcal{K}^n) \mathcal{C}(\mathcal{K}^n)^\top M_n^{-\frac{1}{2}}).
\end{aligned} \tag{29}$$

*We now examine how the behavior of the Koopman representation depends on its spectral properties, by analyzing the cases where the eigenvalues of $\mathcal{K}$ satisfy $|\lambda| > 1$, $|\lambda| \approx 1$, and $|\lambda| < 1$.*

$|\lambda| > 1$: *The Koopman representation is explosive: $\|\mathcal{K}^n\|$ grows exponentially, so the term $\mathcal{K}^n \mathcal{C}(\mathcal{K}^n)^\top$ dominates. As a result, the mutual information $I(z_{t-n}; z_t)$ diverges with $n$, reflecting amplification of initial uncertainty.*

$|\lambda| \approx 1$: *The Koopman representation is temporally coherent: $\mathcal{K}^n \mathcal{C}(\mathcal{K}^n)^\top$ remains bounded, and when noise is small the mutual information is approximately conserved at a constant level. This corresponds to the temporal-coherent information component.*

$|\lambda| < 1$: *The Koopman represent is fast mixing: $\mathcal{K}^n \to 0$ as $n \to \infty$, so the additional term vanishes relative to $M_n$. Thus the mutual information $I(z_{t-n}; z_t) \to 0$, indicating that information from the remote past information is asymptotically lost due to contraction and noise accumulation.*

*An illustration is given in Figure 9, where the left panel shows the $|\lambda| = 1$ case with eigenvalues lying on the unit circle, and the right panel shows the $|\lambda| < 1$ case with eigenvalues contracting toward the origin as $n$ increases.*

**Step 2. Disentanglement of mutual information** $I(z_t; x_t)$.

*According to the chain rule of mutual information, we have*

$$H(x) = H(x \mid z) + I(z; x), \tag{30}$$

*where $H(x \mid z)$ measures the irreducible uncertainty of $x$ given the latent variable (i.e., the information lost in the latent space), and $I(z; x)$ quantifies the total amount of information about $x$ preserved in the latent representation.*

*In this sense, $I(z; x)$ can be regarded as the* maximum information that the decoder can retain *about the data through the latent variables. To better understand the role of structural consistency, we next* decompose $I(z; x)$ *into components, in order to examine how much of this retained information is attributable to the latent forward via Koopman representation.*

$$
\begin{aligned}
&I(z_t; x_t) \\
=& I(z_t, (z_{t-n}, x_t)) - \underline{I(z_{t-n}; z_t | x_t)} \quad \text{(Fact F.1)} \\
=& I(z_{t-n}; z_t) + I(z_t; x_t | z_{t-n}) \\
=& I(z_{t-n}; z_t) + I(z_t; (x_{t-1}, x_t) | z_{t-n}) - \underline{I(z_t; x_{t-1} | z_{t-n}, x_t)} \quad \text{(Fact F.1)} \\
=& I(z_{t-n}; z_t) + I(z_t; x_{t-1} | z_{t-n}) + I(z_t; x_t | z_{t-n}, x_{t-1}) \\
=& I(z_{t-n}; z_t) + I(z_t; x_{t-1} | z_{t-n}) + I(z_t; (z_{t-n}, x_t) | x_{t-1}) - \underline{I(z_{t-n}; z_t | x_{t-1}, x_t)} \\
=& I(z_{t-n}; z_t) + I(z_t; x_{t-1} | z_{t-n}) + I(z_t; x_t | x_{t-1}) - \underline{I(z_t; z_{t-n} | x_{t-1}, x_t)} \\
=& I(z_{t-n}; z_t) + I(z_t; x_{t-1} | z_{t-n}) + I(z_t; x_t | x_{t-1}).
\end{aligned}
\tag{31}
$$

**Step 3.** *According to Step 1, the latent mutual information has been thoroughly explained; we now turn to the second and third terms in equation 31.*

*For linear conditional Gaussian (Lubbe, 1997), the mutual information for*

$$I(a, b | c) = \frac{1}{2} \log \frac{\det \Sigma_{a|c} \det \Sigma_{b|c}}{\det \Sigma_{a,b|c}} \tag{32}$$

*where $\Sigma_{a|c}$ and $\Sigma_{b|c}$ denote the conditional covariance matrices of $a$ and $b$ given $c$, respectively, and $\Sigma_{a,b|c}$ denotes the joint conditional covariance matrix of $(a, b)$ given $c$, i.e.,*

$$\Sigma_{a,b|c} = \begin{bmatrix} \Sigma_{a|c} & \Sigma_{ab|c} \\ \Sigma_{ba|c} & \Sigma_{b|c} \end{bmatrix},$$

*with $\Sigma_{ab|c}$ and $\Sigma_{ba|c}$ being the conditional cross-covariances.*

*The closed-form expression for the mutual information $I(z_t; x_{t-1} \mid z_{t-n})$ is derived as follows. Conditioned on $z_{t-n}$, we have the linear–Gaussian relations*

$$z_{t-1} \mid z_{t-n} \sim \mathcal{N}(\mathcal{K}^{n-1} z_{t-n}, \ M_{n-1}),$$
$$z_t \mid z_{t-n} \sim \mathcal{N}(\mathcal{K}^n z_{t-n}, \ M_n), \tag{33}$$

*where $M_k = \sum_{i=0}^{k-1} \mathcal{K}^i \Sigma (\mathcal{K}^i)^\top$. Assume $x_{t-1} = \mathcal{D} z_{t-1} + \epsilon_{t-1}$ with $\epsilon_{t-1} \sim \mathcal{N}(0, R)$, independent of the process noise. Then*

$$\Sigma_{z_t, x_{t-1} \mid z_{t-n}} = \begin{bmatrix} M_n & \mathcal{K} M_{n-1} \mathcal{D}^\top \\ \mathcal{D} M_{n-1} \mathcal{K}^\top & \mathcal{D} M_{n-1} \mathcal{D}^\top + R \end{bmatrix}. \tag{34}$$

*Plugging equation 33–equation 34 into the Gaussian closed-form identity yields*

$$I(z_t; x_{t-1} \mid z_{t-n}) = \tfrac{1}{2} \log \frac{\det(M_n) \det(\mathcal{D} M_{n-1} \mathcal{D}^\top + R)}{\det\left( \begin{bmatrix} M_n & \mathcal{K} M_{n-1} \mathcal{D}^\top \\ \mathcal{D} M_{n-1} \mathcal{K}^\top & \mathcal{D} M_{n-1} \mathcal{D}^\top + R \end{bmatrix} \right)}. \tag{35}$$

*To analyze how $I(z_t; x_{t-1} \mid z_{t-n})$ scales with $n$, it suffices to study the growth of $M_n = \sum_{i=0}^{n-1} \mathcal{K}^i \Sigma (\mathcal{K}^i)^\top$. If the spectral radius $\rho(\mathcal{K}) < 1$, then $M_n$ converges to the unique solution $M_\infty$ of the discrete Lyapunov equation $M_\infty = \Sigma + \mathcal{K} M_\infty \mathcal{K}^\top$., hence $I(z_t; x_{t-1} \mid z_{t-n})$ remains bounded ("compressible"). Conversely, if $\rho(\mathcal{K}) > 1$, then $M_n$ diverges and the information grows unbounded along the unstable directions. When $\rho(\mathcal{K}) \approx 1$, the growth is slow and reflects long-term temporal coherence (then it contradicts with equation 29, and can be captured by the latent mutual information). Therefore, this compressible information corresponds to the spectral radius $\rho(\mathcal{K}) < 1$.*

*As for the residual term $I(z_t; x_t \mid x_{t-1})$, it can be expanded as*

$$I(z_t; x_t \mid x_{t-1}) = H(x_t \mid x_{t-1}) - H(x_t \mid z_t, x_{t-1})$$
$$= H(x_t \mid x_{t-1}) - \tfrac{1}{2} \log \det((2\pi e)^d R), \tag{36}$$

*where the second equality follows from the observation model $x_t = \mathcal{D} z_t + \epsilon_t$ with $\epsilon_t \sim \mathcal{N}(0, R)$, which implies $H(x_t \mid z_t, x_{t-1}) = \tfrac{1}{2} \log \det((2\pi e)^d R)$. Therefore, this residual mutual information depends only on the noise covariance $R$ and original dynamics $T$, but it has no spectral counterpart in the Koopman operator.*

**Summary.** *The mutual information $I(z_t; x_t)$ naturally decomposes into three parts: (i) temporal-coherent information $I(z_{t-n}; z_t)$, which captures temporal coherence when eigenvalues of Koopman operator $\lambda \approx 1$; (ii) fast-dissipating information $I(z_t; x_{t-1} \mid z_{t-n})$, which remains bounded only in the stable regime $\rho(\mathcal{K}) < 1$; and (iii) residual information $I(z_t; x_t \mid x_{t-1})$, which reflects observation noise and has no spectral counterpart. These three components and their spectral interpretations are summarized in Table 4.*

Table 4: Spectral interpretation of information components in Koopman representation

| Information | Spectral property | Temporal behavior | Information meaning |
|---|---|---|---|
| Temporal-coherent | $\lambda \approx 1$ | Long-lived, persistent | Predictable, low entropy |
| Fast-dissipating | $\lambda < 1$ | Rapidly decaying, short-lived | Transient, compressible under IB |
| Residual / Confounding | – (no spectral counterpart) | Unpredictable, injected at present step | Noise, anomalies, non-predictive leftovers |

### F.5 PROOF OF PROPOSITION 4

To establish this proposition, we analyze the problem from a Lagrangian perspective. Specifically, we investigate how the Koopman representation behaves when the latent mutual information $I(z_{t-n}; z_t)$ is maximized under a finite variance constraint. This perspective reveals a water-filling allocation principle that governs how variance is distributed across spectral modes, thereby clarifying the connection between latent mutual information and the latent variable $z$.

**Proof 4** *Consider latent variable under Koopman representation showing equation 26*

$$z_t = \mathcal{K}^n z_{t-n} + \varepsilon, \quad \varepsilon \sim \mathcal{N}(0, M_n), \quad \mathcal{C} := \mathbb{E}[z_{t-n} z_{t-n}^\top], \tag{37}$$

*where denotes the covariance matrix of $z_{t-n}$. The matrix $\mathcal{C}$ characterizes the spectral distribution of the latent variable, and our goal is to investigate how maximizing latent mutual information influences Koopman representation.*

*According to the previous proof in equation 29, we have*

$$I(z_{t-n}; z_t) = \frac{1}{2} \log \det(\mathrm{I} + M_n^{-\frac{1}{2}} (\mathcal{K}^n) \mathcal{C} (\mathcal{K}^n)^\top M_n^{-\frac{1}{2}}).$$

*Denote the singular value decomposition $M_n^{-\frac{1}{2}} (\mathcal{K})^n = U diag(\sqrt{g_i}) V^\top$ with $g_i \geq 0$. Then*

$$I(z_{t-n}; z_t) = \frac{1}{2} \log \det \left( \mathrm{I} + U diag(\sqrt{g_i}) V^T \mathcal{C} V diag(\sqrt{g_i}) U^\top \right). \tag{38}$$

*As $\mathrm{tr}(\mathcal{C})$ measures the total second moment, we impose $\mathrm{tr}(\mathcal{C}) \leq C$ for some finite constant $C$. This assumption ensures that the Koopman representation has a bounded total variance, preventing degenerate solutions where the variance grows without bound.*

*Under the constraint $\mathrm{tr}(\mathcal{C}) \leq C$, maximizing the latent mutual information under Koopman representation becomes an optimization problem as*

$$\max_{\mathcal{C}} \quad \frac{1}{2} \log \det \left( \mathrm{I} + U diag(\sqrt{g_i}) V^T \mathcal{C} V diag(\sqrt{g_i}) U^\top \right)$$
$$s.t. \ \mathrm{tr}(\mathcal{C}) \leq C. \tag{39}$$

*In equation 38, the matrices $U$ and $V$ are orthogonal, and $\mathcal{C}$ is a symmetric positive semidefinite matrix with eigenvalues $\{p_1, \ldots, p_d\}$ with $p_i \geq 0$ for all $i$. We interpret these eigenvalues as spectral weights of the Koopman observables/features, indicating how variance is allocated across the observable/feature directions $\{\phi_1, \ldots, \phi_d\}$ defined in equation 1. Then, optimization problem in equation 39 becomes a water-filling problem as*

$$\max_{p_i, \sum_{i=1}^d p_i \leq C} \frac{1}{2} \sum_{i=1}^d \log(1 + g_i p_i). \tag{40}$$

*The Lagrangian formulation becomes*

$$\mathcal{L} = \frac{1}{2} \sum_{i=1}^d \log(1 + g_i p_i) - \mu(\sum_{i=1}^d p_i - C) - \sum_{i=1}^d \eta_i p_i. \tag{41}$$

*According to Karush–Kuhn–Tucker (KKT) condition (Boyd & Vandenberghe, 2004), we obtain*

$$\frac{\partial \mathcal{L}}{\partial p_i} = \frac{g_i}{2(1 + g_i p_i)} - \mu - \eta_i = 0, \ \eta_i = 0 \quad \Rightarrow \quad p_i = \frac{1}{2\mu} - \frac{1}{g_i}. \tag{42}$$

*Based on the non-negativity of the eigenvalues $p_i \geq 0$ for all $i$, the optimal allocation is*

$$p_i = \max \left\{ 0, \ \frac{1}{2\mu} - \frac{1}{g_i} \right\}, \tag{43}$$

*where $\mu$ is the Lagrange multiplier determined by the variance budget constraint. This solution characterizes the spectral weights of the Koopman representation along each observable/feature direction, and reveals two key phenomena:*

- ***Concentration on temporally coherent modes.*** *Since $g_i$ depends on the Koopman eigenvalues through $\mathcal{K}^n$, larger $p_i$ in equation 43 are assigned to eigen-directions with $|\lambda| \approx 1$, corresponding to temporal-coherent modes.*

- ***Mode collapse.*** *Because $\sum_{i=1}^d p_i \leq C$, variance is preferentially allocated to directions with larger gain $g_i$, while less informative directions receive zero weight. This leads to a low-rank allocation (low effective dimension) where only a subset of modes are retained.*

*In summary, this proof demonstrates that maximizing latent mutual information is equivalent to a water-filling allocation of spectral weights, which naturally explains why emphasizing this objective can lead to mode collapse in the Koopman representation.*

## F.6 PROOF OF PROPOSITION 5

Connecting to Proposition 4, we continue to prove Proposition 5 via Lagrangian formulation. Without entropy regularization, the solution degenerates to low-rank (mode collapse); with entropy regularization, the solution assigns non-zero weights to all directions, improving effective dimension.

**Proof 5** *According to Definition 2.3, we can normalize $\mathcal{C}$ into a density matrix $\frac{\mathcal{C}}{\mathrm{tr}(\mathcal{C})}$, then*

$$S(\tfrac{\mathcal{C}}{\mathrm{tr}(\mathcal{C})}) = -\sum_{i=1}^d \frac{p_i}{\mathrm{tr}(\mathcal{C})} \log \frac{p_i}{\mathrm{tr}(\mathcal{C})}$$

*with $p_i$ is the eigenvalue of $\mathcal{C}$.*

*Under a given regularization coefficient $\gamma$, this normalization allows us to improve the effecitve dimension. Based on equation 41, the modified Lagrangian formulation under the regularized Von Neumann entropy becomes*

$$\mathcal{L} = \frac{1}{2}\sum_{i=1}^d \log(1 + g_i p_i) + \gamma(-\sum_{i=1}^d \frac{p_i}{\mathrm{tr}(\mathcal{C})} \log \frac{p_i}{\mathrm{tr}(\mathcal{C})}) - \mu(\sum_{i=1}^d p_i - C) - \sum_{i=1}^d \eta_i p_i. \quad (44)$$

*Based on the KKT condition, we have*

$$\frac{\partial \mathcal{L}}{\partial p_i} = \frac{g_i}{2(1 + g_i p_i)} - \mu - \eta_i - \frac{\gamma}{\mathrm{tr}(\mathcal{C})}(\log \frac{p_i}{\mathrm{tr}(\mathcal{C})} + 1) = 0, \quad (45)$$

*where $\eta_i = 0$ under the KKT condition. The solution of equation 45 is*

$$\frac{g_i}{2(1 + g_i p_i)} - \mu = \frac{\gamma}{\mathrm{tr}(\mathcal{C})}(\log \frac{p_i}{\mathrm{tr}(\mathcal{C})} + 1).$$

*Since a closed-form solution is not directly available, we proceed with further algebraic transformation.*

*By reorganization,*

$$\log \frac{p_i}{\mathrm{tr}(\mathcal{C})} = \frac{\mathrm{tr}(\mathcal{C})}{\gamma}\left(\frac{g_i}{2(1 + g_i p_i)} - \mu\right) - 1.$$

*Exponential both sides:*

$$\frac{p_i}{\mathrm{tr}(\mathcal{C})} = \exp\left(\frac{\mathrm{tr}(\mathcal{C})}{\gamma}\left(\frac{g_i}{2(1 + g_i p_i)} - \mu\right) - 1\right)$$

*Then,*

$$p_i = \text{tr}(\mathcal{C}) \exp\left( \frac{\text{tr}(\mathcal{C})}{\gamma} \left( \frac{g_i}{2(1 + g_i p_i)} - \mu \right) - 1 \right)$$

*We can transform the above form as,*

$$p_i \underbrace{\exp\left( -\frac{\text{tr}(\mathcal{C})}{\gamma} \frac{g_i}{2(1 + g_i p_i)} \right)}_{>0} = \underbrace{\text{tr}(\mathcal{C}) \exp\left( -1 - \frac{\text{tr}(\mathcal{C})}{\gamma} \mu \right)}_{=C_1 > 0}. \tag{46}$$

*Here $C_1 = \text{tr}(\mathcal{C}) \exp\left( -1 - \frac{\text{tr}(\mathcal{C})}{\gamma} \mu \right)$ is a positive constant Introducing $y = 1 + g_i p_i$, we can write equation 46 via algebraic transform:*

$$(y - 1) \exp\left( -\frac{\text{tr}(\mathcal{C}) g_i}{2\gamma y} \right) = g_i C_1. \tag{47}$$

*The above equation is related to the form $x \exp(x) + rx = constant$, we can solve it via the generalized Lambert W function (also known as r-Lambert W [a], see (Veberič, 2012))*

$$y = \frac{\frac{\text{tr}(\mathcal{C}) g_i}{2\gamma}}{W_{1/(g_i C_1)}\left( \frac{\frac{\text{tr}(\mathcal{C}) g_i}{2\gamma}}{g_i C_1} \right)}. \tag{48}$$

*Here, $W_{1/(g_i C_1)}(\cdot)$ is the r-Lambert W function. Since $p_i = \frac{y-1}{g_i}$, the closed form of $p_i$ becomes*

$$p_i = \frac{\frac{\text{tr}(\mathcal{C})}{2\gamma}}{W_{1/(g_i C_1)}\left( \frac{\frac{\text{tr}(\mathcal{C}) g_i}{2\gamma}}{g_i C_1} \right)} - \frac{1}{g_i} > 0. \tag{49}$$

*Then, the solution of equation 44 under the regularized von Neumann entropy assigns non-zero spectral weight to all observable/feature directions $\{\phi_1, \ldots, \phi_d\}$, since $p_i > 0$ holds according to equation 46. Consequently, the effective dimension is provably improved.*

---

[a] $W_r$ denotes the generalized Lambert W function, defined as the solution of $x \exp(x) + rx = constant$.

# G PRACTICAL DETAILS, IMPLEMENTATION AND EXPERIMENTAL DETAILS

## G.1 IMPLEMENTATION DETAILS FOR VAE

---

**Algorithm 1** Information-Theoretic Koopman Representation (VAE, probabilistic)

---

**Require:** Dataset $\mathcal{D} = \{x_n\}_{n=0}^T$; network parameters $(\alpha, \beta, \gamma)$; learning rate $\eta$; number of epochs $K$; batch size $B$; neighbor window $k$; temperature $\tau$.

1: Initialize encoder $p_\theta(z|x)$, decoder $p_\omega(x|z)$, and latent dynamics network $q_\psi(z_n|z_{n-1})$.
2: **for** epoch $= 1$ to $K$ **do**
3:     **for** each minibatch $\{x_1, \ldots, x_B\}$ from $\mathcal{D}$ **do**
4:         Sample latents $z_i \sim p_\theta(z|x_i)$.
5:         **Temporal coherence (InfoNCE):** For each $z_n$, treat its *temporal* neighbors $\mathcal{P}_n = \{z_{n\pm i} \mid 1 \leq i \leq k\}$ as positive samples, compute

$$I(z_n; \mathcal{P}_n) \approx \frac{1}{|\mathcal{P}_n|} \sum_{p \in \mathcal{P}_n} \log \frac{\exp(z_n^\top z_p/\tau)}{\sum_{j=1}^B \exp(z_n^\top z_j/\tau)}.$$

6:         **Structural consistency:** compute latent likelihood

$$\mathbb{E}_{p_\theta(z_n|x_n)}[\log q_\psi(z_n|z_{n-1})]$$

7:         **Predictive sufficiency:** compute covariance $\mathcal{C} = \frac{1}{B}\sum_i (z_i - \bar{z})(z_i - \bar{z})^\top$, where $\bar{z} = \frac{1}{B}\sum_i z_i$; normalize $P = \mathcal{C}/\mathrm{tr}(\mathcal{C})$, then compute

$$S(P) = -\sum_j \lambda_j \log \lambda_j, \quad \text{where } \lambda_j \text{ denotes } j\text{th eigenvalue of } P.$$

8:         **Standard Evidence Lower Bound (ELBO) term (for training stability and reconstruction):**

$$\mathcal{L}_{\text{ELBO}} = \log p_\omega(x_{n-1}|z_{n-1}) - D_{\text{KL}}(p_\theta(z_{n-1}|x_{n-1}) \,\|\, \mathcal{N}(0, I)).$$

9:         **Total Loss:**

$$\mathcal{L} = -\Big[\alpha I(z_n; \mathcal{P}_n) + \mathbb{E}_{p_\theta(z_n|x_n)}[\log q_\psi(z_n|z_{n-1})] + H_{p_\theta}(z_n|x_n)$$
$$+ \log p_\omega(x_n|z_n) + \gamma S(P) + \mathcal{L}_{\text{ELBO}}\Big].$$

10:         Update $\theta, \omega, \psi$ using Adam step $\eta$
11:     **end for**
12: **end for**

---

**Connection to Structural Consistency.** By Definition E.3, when given $z_{n-1}$ the conditional mutual information is

$$I(z_n; x_n \mid z_{n-1}) = H(z_n \mid z_{n-1}) - H(z_n \mid x_n, z_{n-1}).$$

Since the encoder is independent of $z_{n-1}$ in our setting (according to Figure 8), this simplifies to

$$I(z_n; x_n \mid z_{n-1}) = H(z_n \mid z_{n-1}) - H(z_n \mid x_n). \tag{50}$$

Consider the encoder distribution

$$p_\theta(z_n \mid x_n),$$

which maps observations to latent variables, and the Koopman prior

$$q_\psi(z_n \mid z_{n-1}) = \mathcal{N}(z_n \mid \mathcal{K}_\psi z_{n-1}, \Sigma_\psi),$$

which models latent evolution as a linear Gaussian transition governed by the Koopman operator $\mathcal{K}_\psi$.

Then the conditional mutual information can be equivalently written as the following form according to Definition E.3 or equation 50:

$$I(z_n; x_n \mid z_{n-1}) = \mathbb{E}_{p_\theta(z_n|x_n)} \left[ \log \frac{p_\theta(z_n \mid x_n)}{q_\psi(z_n \mid z_{n-1})} \right]. \tag{51}$$

Expanding the term in equation 51, we

$$I(z_n; x_n \mid z_{n-1}) = \mathbb{E}_{p_\theta(z_n|x_n)}[-\log q_\psi(z_n \mid z_{n-1})] - H_{p_\theta}(z_n \mid x_n).$$

has two effects:

1. **Alignment with Koopman dynamics.** The expectation term $\mathbb{E}_{p_\theta(z_n|x_n)}[-\log q_\psi(z_n|z_{n-1})]$ requires samples drawn from the encoder to lie in regions of high likelihood under the Koopman prior. Since the prior is parameterized as a linear Gaussian transition, minimizing the KL forces the encoder outputs to be predictable under a linear structure.

2. **Entropy regularization.** The entropy term $H_{p_\theta}(z_n|x_n)$ encourages that the encoder not to be deterministic.

Together, these effects ensure that the latent variables produced by the encoder not only encode information about the current state but also evolve consistently with the linear Gaussian dynamics imposed by the Koopman operator. Formally,

$$p_\theta(z_n \mid x_n) \approx q_\psi(z_n \mid z_{n-1}) \quad \implies \quad z_n \text{ evolves approximately linearly under } \mathcal{K}_\psi,$$

which enforces *structural consistency* in the latent space.

## G.2 Implementation Details for AE

---

**Algorithm 2** Information-Theoretic Koopman Representation (AE, deterministic)

---

**Require:** Dataset $\mathcal{D} = \{x_n\}_{n=0}^{T}$; hyperparameters $(\alpha, \beta, \gamma)$; learning rate $\eta$; number of epochs $K$; batch size $B$; neighbor window $k$; temperature $\tau$.

1: Initialize deterministic encoder $z_n = f_\theta(x_n)$, decoder $\hat{x}_n = g_\omega(z_n)$, and Koopman operator $\mathcal{K}_\psi$.

2: **for** epoch $= 1$ to $K$ **do**

3:     **for** each minibatch $\{x_1, \ldots, x_B\}$ from $\mathcal{D}$ **do**

4:         Encode latents $z_i = f_\theta(x_i)$ for $i = 1, \ldots, B$.

5:         **Temporal coherence (InfoNCE):**

$$I(z_n; \mathcal{P}_n) \approx \tfrac{1}{|\mathcal{P}_n|} \sum_{p \in \mathcal{P}_n} \log \frac{\exp(z_n^\top z_p / \tau)}{\sum_{j=1}^{B} \exp(z_n^\top z_j / \tau)}.$$

6:         **Structural consistency (deterministic):**

$$\mathcal{L}_{\text{Koop}} = \|z_{n+1} - \mathcal{K}_\psi z_n\|^2.$$

7:         **Predictive sufficiency:** compute $S(P)$ from normalized covariance $P = \frac{\mathcal{C}}{\text{tr}(\mathcal{C})}$, $\mathcal{C} = \frac{1}{B} \sum (z_i - \bar{z})(z_i - \bar{z})^\top$.

8:         **Reconstruction:**

$$\mathcal{L}_{\text{rec}} = \|x_n - g_\omega(z_n)\|^2.$$

9:         **Total Loss:**

$$\mathcal{L} = \mathcal{L}_{\text{rec}} - \alpha I(z_n; \mathcal{P}_n) + \beta \mathcal{L}_{\text{Koop}} - \gamma S(P).$$

10:         Update $\theta, \omega, \psi$ with Adam step $\eta$.

11:     **end for**

12: **end for**

---

## G.3 Experiment Settings and Additional Results

### G.3.1 Physical Simulation

**Lorenz.** The Lorenz dataset is generated from the classical Lorenz system of ordinary differential equations (ODEs), which model simplified atmospheric convection. The governing equations are:

$$\begin{aligned}
\dot{x} &= \sigma(y - x), \\
\dot{y} &= x(\rho - z) - y, \\
\dot{z} &= xy - \beta z,
\end{aligned} \tag{52}$$

where $\sigma = 10$, $\rho = 28$, and $\beta = 8/3$ are the standard chaotic parameters. The system is integrated using a fixed time step $\Delta t = 0.1\,\text{s}$ with a fourth-order Runge–Kutta method. The resulting trajectories exhibit chaotic behavior and are commonly used as benchmarks for nonlinear dynamical system identification.

**Kármán Vortex.** The Kármán vortex street dataset is generated from the two-dimensional incompressible Navier–Stokes equations, which describe the velocity field $(u, v)$ and pressure $p$ of a viscous fluid:

$$\begin{aligned}
\frac{\partial u}{\partial t} + u\frac{\partial u}{\partial x} + v\frac{\partial u}{\partial y} &= -\frac{\partial p}{\partial x} + \frac{1}{Re}\left(\frac{\partial^2 u}{\partial x^2} + \frac{\partial^2 u}{\partial y^2}\right), \\
\frac{\partial v}{\partial t} + u\frac{\partial v}{\partial x} + v\frac{\partial v}{\partial y} &= -\frac{\partial p}{\partial y} + \frac{1}{Re}\left(\frac{\partial^2 v}{\partial x^2} + \frac{\partial^2 v}{\partial y^2}\right), \\
\frac{\partial u}{\partial x} + \frac{\partial v}{\partial y} &= 0,
\end{aligned} \tag{53}$$

where $Re = UL/\nu$ is the Reynolds number, defined with characteristic velocity $U$, length $L$, and kinematic viscosity $\nu$. The training dataset covers flows with $Re \in [40, 1000]$, while the test dataset focuses on $Re = 1000$. The flow is simulated around a cylinder, producing the characteristic alternating vortex shedding pattern. The domain is discretized on a $64 \times 64$ grid, with time step $\Delta t = 0.001\,\text{s}$, and both $u$ and $v$ velocity components are recorded at each grid point (data from (Yining et al., 2023)).

**Dam Flow.** The dam flow dataset is also generated from the two-dimensional incompressible Navier–Stokes equations, using the same formulation as in the Kármán vortex case. The training dataset spans $Re \in [40, 1000]$ and the test dataset uses $Re = 1000$. The flow is initialized in a rectangular channel with a fixed dam obstacle, where an imposed inlet velocity drives the fluid past the dam-like structure, generating a simple wake pattern downstream. The domain is discretized on a $64 \times 64$ spatial grid, with temporal resolution $\Delta t = 0.1\,\text{s}$ (data from (Yining et al., 2023)).

The ERA5 dataset is a global atmospheric reanalysis produced by the European Centre for Medium-Range Weather Forecasts (ECMWF), providing a physically consistent estimate of the large-scale circulation from 1940 to the present (Hersbach et al., 2020). The physic state consists of five channels: 500,hPa geopotential, 850,hPa temperature, 700,hPa specific humidity, and 850,hPa wind components in the zonal and meridional directions. We train all baselines from 1979-01-01 to 2016-01-01 and test after 2018-01-01 (data from (Rasp et al., 2024)). Illustrations are provided in Figure 10.

### G.3.2 Visual Inputs

**Planar System** In this task the main goal is to navigate an agent in a surrounded area on a 2D plane (Breivik & Fossen, 2008), whose goal is to navigate from a corner to the opposite one, while avoiding the six obstacles in this area. The system is observed through a set of $40 \times 40$ pixel images taken from the top view, which specifies the agent's location in the area. Actions are two-dimensional and specify the $x - y$ direction of the agent's movement, and given these actions the next positional state of the agent is generated by a deterministic underlying (unobservable) state evolution function. **Start State**: one of three corners (excluding bottom-right). **Goal State**: bottom-right corner. **Agent's Objective**: agent is within Euclidean distance of 2 from the goal state.

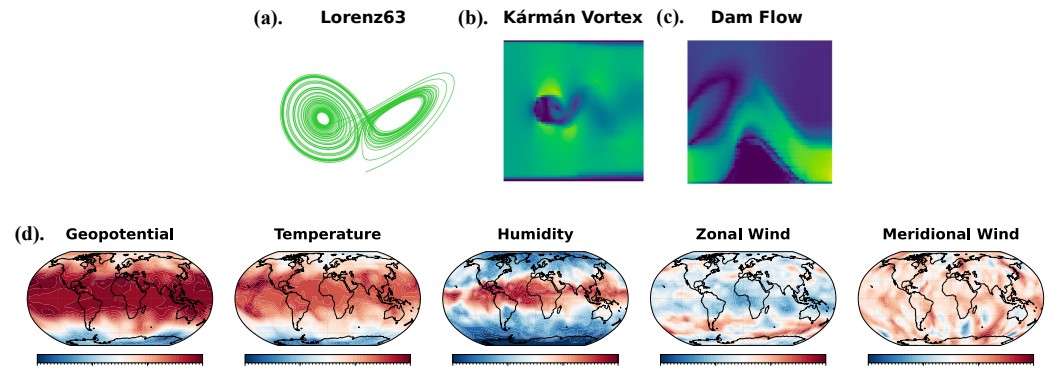

Figure 10: Examples of physical simulation: (a).Lorenz 63, (b).Kármán Vortex, (c).Dam Flow, (d).ERA5

**Inverted Pendulum — SwingUp & Balance**   This is the classic problem of controlling an inverted pendulum (Furuta et al., 1991) from $48 \times 48$ pixel images. The goal of this task is to swing up an under-actuated pendulum from the downward resting position (pendulum hanging down) to the top position and to balance it. The underlying state $s_t$ of the system has two dimensions: angle and angular velocity, which is unobservable. The control (action) is 1-dimensional, which is the torque applied to the joint of the pendulum. To keep the Markovian property in the observation (image) space, similar to the setting in E2C, each observation $x_t$ contains two images generated from consecutive time-frames (from current time and previous time). This is because each image only shows the position of the pendulum and does not contain any information about the velocity. **Start State**: Pole is resting down (SwingUp), or randomly sampled in $\pm\pi/6$ (Balance). **Agent's Objective**: pole's angle is within $\pm\pi/6$ from an upright position.

**CartPole**   This is the visual version of the classic task of controlling a cart-pole system (Geva & Sitte, 1993). The goal in this task is to balance a pole on a moving cart, while the cart avoids hitting the left and right boundaries. The control (action) is 1-dimensional, which is the force applied to the cart. The underlying state of the system $s_t$ is 4-dimensional, which indicates the angle and angular velocity of the pole, as well as the position and velocity of the cart. Similar to the inverted pendulum, in order to maintain the Markovian property the observation $x_t$ is a stack of two $80 \times 80$ pixel images generated from consecutive time-frames. **Start State**: Pole is randomly sampled in $\pm\pi/6$. **Agent's Objective**: pole's angle is within $\pm\pi/10$ from an upright position.

**3-link Manipulator — SwingUp & Balance**   The goal in this task is to move a 3-link manipulator from the initial position (which is the downward resting position) to a final position (which is the top position) and balance it. In the 1-link case, this experiment is reduced to inverted pendulum. In the 2-link case the setup is similar to that of acrobot , except that we have torques applied to all intermediate joints, and in the 3-link case the setup is similar to that of the 3-link planar robot arm domain that was used in the E2C paper, except that the robotic arms are modeled by simple rectangular rods (instead of real images of robot arms), and our task success criterion requires both the swing-up (manipulate to final position) and balance. The underlying (unobservable) state $s_t$ of the system is 6-dimensional, which indicates the relative angular velocities and relative angles of the 3 links. **Start State**: Pole is resting down. **Agent's Objective**: pole's angle is within $\pm\pi/6$ from an upright position.

The control algorithm is linear quadratic control in the latent space and the corresponding control horizon follows the setting in (Levine et al., 2020).

### G.3.3   GRAPH-STRUCTURED DYNAMICS FOR SIMULATION

In the numerical experiments, we adopt the graph environments introduced in (Li et al., 2020), where interactions among objects are modeled differently according to their connection types and physical

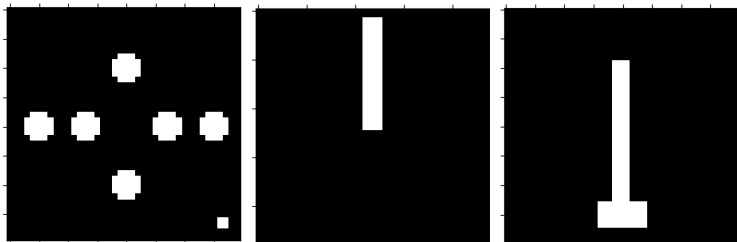

Figure 11: The examples of visual inputs: Planar (left), Pendulum (middle), Cartpole (right).

properties. These environments are designed to capture diverse interaction dynamics through a Koopman representation (see illustrative examples in Figure 12), as detailed below:

In the Rope environment, the top mass is fixed in height and is treated differently from the other masses, resulting in two distinct types of self-interactions: one for the top mass and one for non-top masses. Additionally, there are eight types of interactions between different objects. Each mass is represented by four dimensions, encoding its state and velocity. Objects in a relation can be either the top mass or a non-top mass, yielding four possible combinations. Interactions may occur between adjacent masses or between masses that are two hops apart. In total, this gives $4 \times 2 = 8$ types of interactions between different objects. Training is performed on environments with 5–9 objects, while testing uses 10–14 objects. The overall dimensionality ranges from 40 to 56.

In the Soft environments, quadrilaterals are categorized into four types: rigid, soft, actuated, and fixed, each with its own form of self-interaction. For interactions between objects, an edge is defined between two quadrilaterals only if they are connected at a point or along an edge. Connections from different directions are treated as distinct relations, with eight possible directions: up, down, left, right, up-left, down-left, up-right, and down-right. Relation types also encode the category of the receiving object, resulting in a total of $(8 + 1) \times 4 = 36$ possible relation types between objects. Training is conducted on environments with 5–9 quadrilaterals, while testing uses 10–14 quadrilaterals. Each quadrilateral is represented by a 16-dimensional vector, giving a total dimensionality ranging from 160 to 224.

In noisy environment, the additive noise is zero-mean Gaussian with standard deviation equal to 10% of the standard deviation of the observation data.

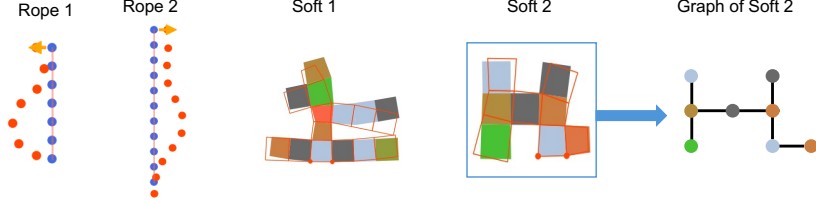

Figure 12: Examples of ropes and soft robots. Left: Blue nodes denote the initial states of Rope 1 and Rope 2, while orange nodes show their states after 40 time steps. Right: Interconnected quadrilaterals indicate the initial states, and the boxes represent the states of the soft robots after 40 time steps. The second soft robot (Soft 2) can be abstracted as a graph structure shown on the right.

### G.4 IMPLEMENTATION ALGORITHM OF THREE TASKS

Table 5: Model structures across experimental environments. Here, $\mathcal{K}z_t + Ba_t$ denotes a controlled latent transition with linear control input $a_t$ (Visual Inputs case). $\mathcal{K}(A)$ denotes an adjacency-conditioned Koopman operator, corresponding to a shared Koopman composition modulated by the adjacency matrix $A$ (i.e., $\mathcal{K}(A) := A \otimes \mathcal{K}$ in graph environments; see Li et al. (2020, Page 4) for details).

| Environment | Structure | Key Features |
|---|---|---|
| Physical Simulation | AE (G.2) | $z_{t+1} = \mathcal{K}z_t$; reconstruction; Koopman linear forward; InfoNCE; von Neumann entropy |
| Visual Inputs (Control Tasks) | VAE (G.1) | $z_{t+1} = \mathcal{K}z_t + Ba_t + \epsilon$ (Linear Gaussian); VAE ELBO; reconstruction; InfoNCE; von Neumann entropy |
| Graph-structured Dynamics | AE (G.2) | $z_{t+1} = \mathcal{K}(A)z_t$ (adjacency-conditioned); reconstruction; InfoNCE; von Neumann entropy |

## G.5 MORE EXPERIMENTAL RESULTS

### G.5.1 PHYSICAL SIMULATIONS

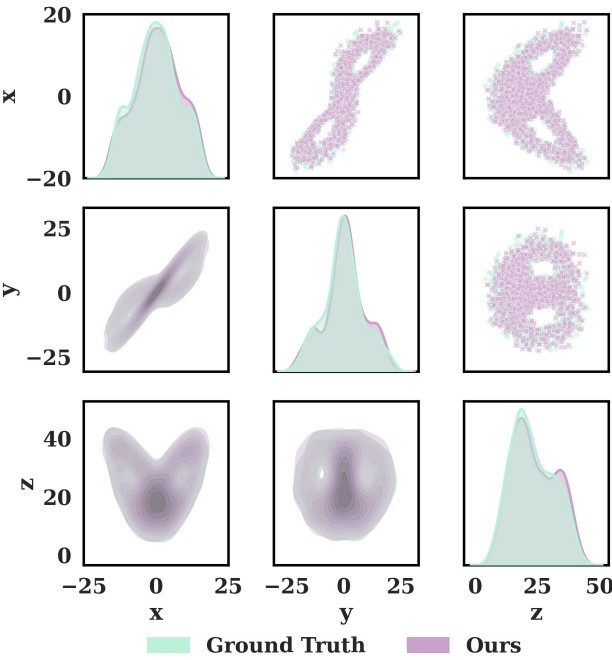

Figure 13: Comparison of sampled spatial distributions based on $100000-$step data for Lorenz 63. **Green** denotes the ground-truth distribution from the physical solver, and **purple** denotes samples generated by our method. Across both marginal and joint projections, the two distributions exhibit close agreement, demonstrating that our empirical results capture the underlying dynamics.

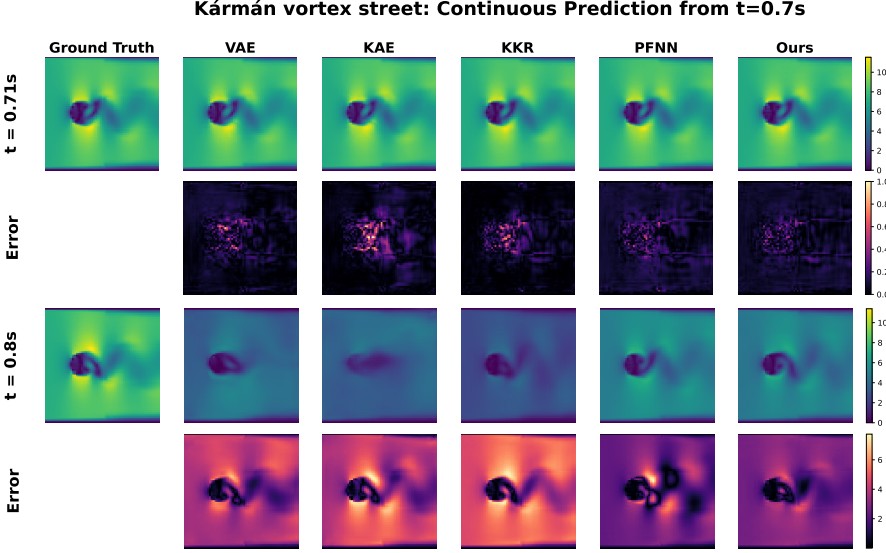

Figure 14: Comparison of continuous predictions for the Kármán vortex street starting from $t = 0.7s$. Ground truth are contrasted with predictions from VAE, KAE, KKR, PFNN, and our method. Error maps in the lower panels demonstrate that, compared with other models, our method achieves the closest agreement and effectively prevents collapse in the predicted fields.

Figure 15: Comparison of continuous predictions for the dam flow starting from $t = 5.0s$. Ground truth are contrasted with predictions from VAE, KAE, KKR, and our method. Error maps in the lower panels demonstrate that, compared with other models, our method more effectively prevents collapse in the predicted fields for dam flow.

Table 6: Per-channel performance comparison on ERA5 weather forecasting. $N$-NRMSE and $N$-SSIM denote errors at $N$ prediction steps; values in parentheses indicate the standard deviation across test samples. Even under the highly stochastic and high-dimensional ERA5 weather dynamics, our approach outperforms all baselines across both short-term and long-term prediction horizons.

| Channel | Metric | KAE | KKR | PFNN | Ours |
|---|---|---|---|---|---|
| Geopotential | 5-NRMSE | 0.058 (0.020) | 0.061 (0.027) | 0.046 (0.012) | **0.023 (0.005)** |
| | 10-NRMSE | 0.068 (0.018) | 0.074 (0.026) | 0.062 (0.018) | **0.032 (0.010)** |
| | 50-NRMSE | 0.157 (0.071) | 0.082 (0.025) | 0.082 (0.016) | **0.075 (0.017)** |
| | 5-SSIM | 0.860 (0.054) | 0.852 (0.064) | 0.882 (0.036) | **0.964 (0.011)** |
| | 10-SSIM | 0.836 (0.051) | 0.820 (0.065) | 0.848 (0.052) | **0.943 (0.028)** |
| | 50-SSIM | 0.665 (0.195) | 0.790 (0.049) | 0.765 (0.055) | **0.806 (0.039)** |
| Temperature | 5-NRMSE | 0.049 (0.019) | 0.052 (0.032) | 0.040 (0.009) | **0.022 (0.003)** |
| | 10-NRMSE | 0.056 (0.017) | 0.064 (0.028) | 0.051 (0.015) | **0.026 (0.006)** |
| | 50-NRMSE | 0.114 (0.054) | 0.074 (0.030) | 0.067 (0.018) | **0.063 (0.019)** |
| | 5-SSIM | 0.866 (0.046) | 0.862 (0.058) | 0.888 (0.026) | **0.956 (0.008)** |
| | 10-SSIM | 0.844 (0.044) | 0.829 (0.063) | 0.859 (0.042) | **0.942 (0.019)** |
| | 50-SSIM | 0.671 (0.216) | 0.802 (0.059) | 0.803 (0.051) | **0.825 (0.042)** |
| Humidity | 5-NRMSE | 0.064 (0.029) | 0.069 (0.043) | 0.055 (0.011) | **0.031 (0.003)** |
| | 10-NRMSE | 0.077 (0.027) | 0.085 (0.041) | 0.070 (0.020) | **0.038 (0.006)** |
| | 50-NRMSE | 0.165 (0.086) | 0.093 (0.040) | 0.088 (0.021) | **0.084 (0.028)** |
| | 5-SSIM | 0.859 (0.056) | 0.856 (0.076) | 0.880 (0.026) | **0.954 (0.008)** |
| | 10-SSIM | 0.818 (0.064) | 0.805 (0.095) | 0.835 (0.053) | **0.933 (0.021)** |
| | 50-SSIM | 0.663 (0.220) | 0.781 (0.087) | 0.776 (0.058) | **0.799 (0.057)** |
| Wind $u$ direction | 5-NRMSE | 0.055 (0.009) | 0.056 (0.011) | 0.053 (0.006) | **0.032 (0.006)** |
| | 10-NRMSE | 0.059 (0.008) | 0.061 (0.009) | 0.060 (0.008) | **0.041 (0.010)** |
| | 50-NRMSE | 0.096 (0.030) | 0.063 (0.006) | 0.070 (0.007) | **0.062 (0.005)** |
| | 5-SSIM | 0.505 (0.122) | 0.502 (0.141) | 0.537 (0.084) | **0.814 (0.051)** |
| | 10-SSIM | 0.433 (0.115) | 0.415 (0.134) | 0.424 (0.120) | **0.721 (0.107)** |
| | 50-SSIM | 0.300 (0.251) | 0.361 (0.085) | 0.267 (0.100) | **0.382 (0.068)** |
| Wind $v$ direction | 5-NRMSE | 0.051 (0.008) | 0.051 (0.009) | 0.050 (0.006) | **0.031 (0.005)** |
| | 10-NRMSE | 0.054 (0.007) | 0.054 (0.008) | 0.055 (0.007) | **0.040 (0.010)** |
| | 50-NRMSE | 0.060 (0.006) | **0.057 (0.005)** | 0.087 (0.025) | **0.057 (0.005)** |
| | 5-SSIM | 0.240 (0.159) | 0.247 (0.183) | 0.300 (0.091) | **0.649 (0.098)** |
| | 10-SSIM | 0.163 (0.133) | 0.163 (0.150) | 0.208 (0.101) | **0.499 (0.163)** |
| | 50-SSIM | 0.105 (0.077) | 0.093 (0.082) | **0.165 (0.193)** | 0.094 (0.074) |

Table 7: Training time statistics for different models and tasks. Epoch times are reported as mean ± std (in seconds). For *Ours*, InfoNCE and entropy (von Neumann entropy) rows correspond to the total per-epoch computation. Notably, the overhead introduced by InfoNCE and von Neumann entropy is marginal, accounting for only a small percentage of the total training time.

| Task | Metric | VAE | KAE | KKR | PFNN | Ours |
|---|---|---|---|---|---|---|
| Kármán vortex | Epoch time (s) | 172.94 ± 4.11 | 195.26 ± 3.47 | 186.47 ± 1.86 | 182.52 ± 2.53 | 201.23 ± 1.07 |
| | InfoNCE time (s) | – | – | – | – | 13.78 ± 0.93 |
| | Entropy time (s) | – | – | – | – | 0.97 ± 0.13 |
| Dam Flow | Epoch time (s) | 16.21 ± 0.29 | 16.95 ± 0.33 | 17.48 ± 0.29 | – | 18.92 ± 0.36 |
| | InfoNCE time (s) | – | – | – | – | 0.76 ± 0.04 |
| | Entropy time (s) | – | – | – | – | 0.56 ± 0.08 |
| ERA5 | Epoch time (s) | – | 240.20 ± 0.52 | 224.95 ± 0.54 | 242.33 ± 0.71 | 253.24 ± 2.05 |
| | InfoNCE time (s) | – | – | – | – | 15.09 ± 0.80 |
| | Entropy time (s) | – | – | – | – | 7.43 ± 0.64 |

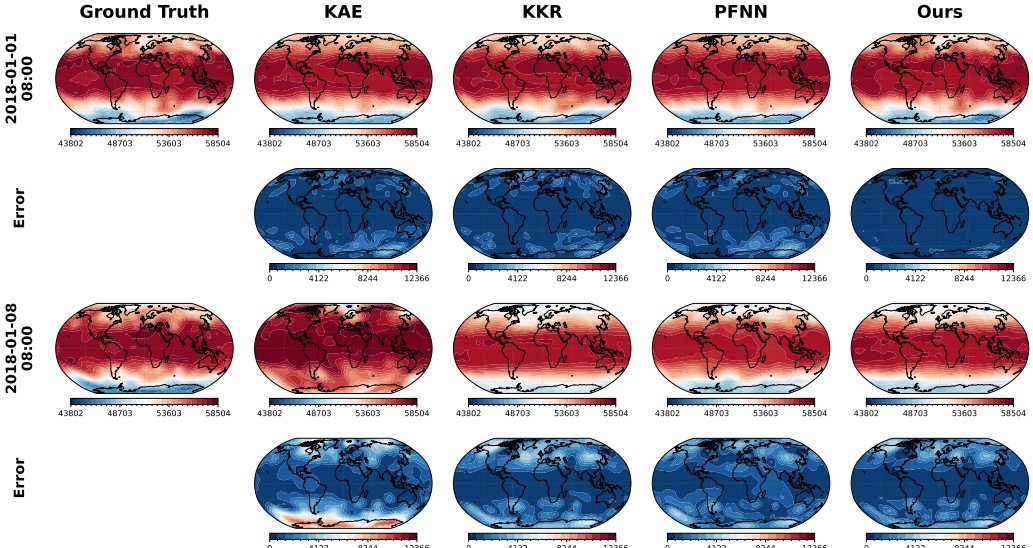

Figure 16: Comparison of continuous predictions for the global geopotential starting from $2018 - 01 - 01 - 00 : 00$ to $2018 - 01 - 08 - 08 : 00$. Ground truth are contrasted with predictions from KAE, KKR, PFNN, and our method. Error maps in the lower panels demonstrate that, compared with other models, showing with more stable and accurate results of our model.

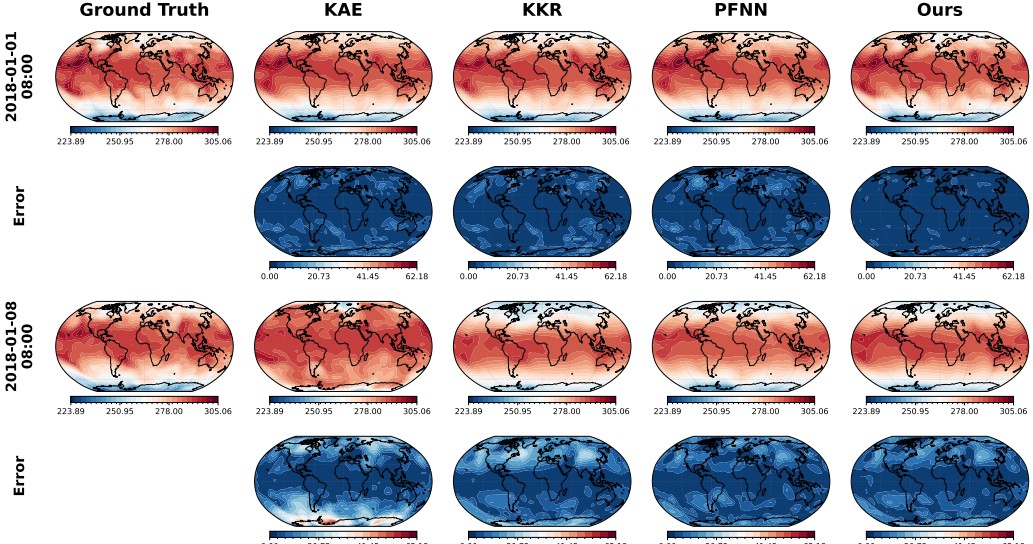

Figure 17: Comparison of continuous predictions for the global temperature starting from $2018 - 01 - 01 - 00 : 00$ to $2018 - 01 - 08 - 08 : 00$. Ground truth are contrasted with predictions from KAE, KKR, PFNN, and our method. Error maps in the lower panels demonstrate that, compared with other models, showing with more stable and accurate results of our model.

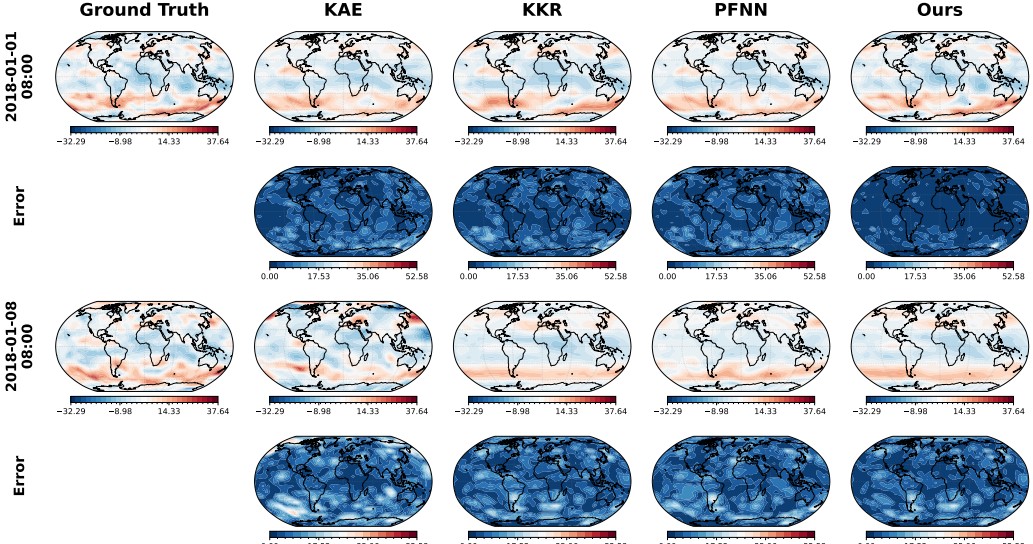

Figure 18: Comparison of continuous predictions for the global $u-$direction wind starting from $2018-01-01-00:00$ to $2018-01-08-08:00$. Ground truth are contrasted with predictions from KAE, KKR, PFNN, and our method. Error maps in the lower panels demonstrate that, compared with other models, showing with more stable and accurate results of our model.

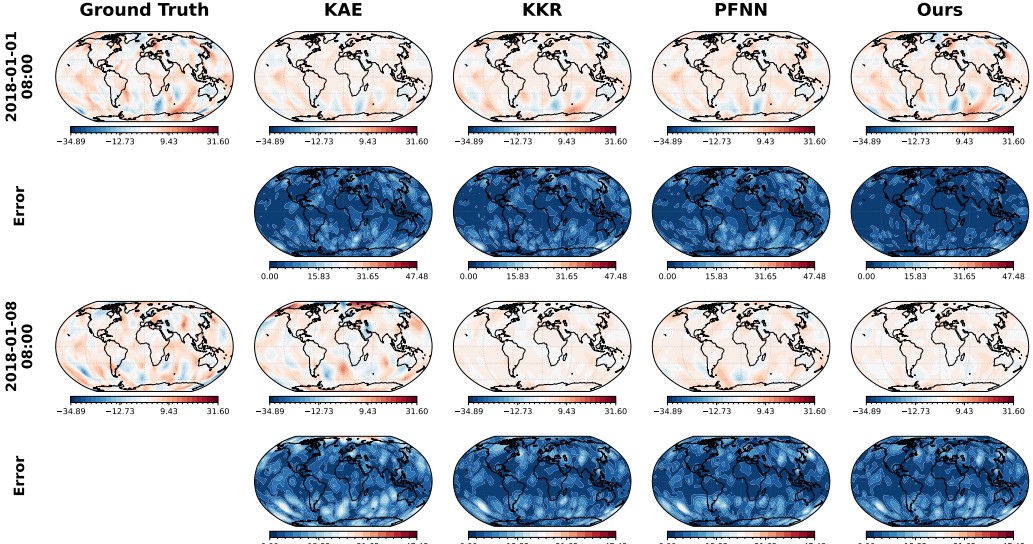

Figure 19: Comparison of continuous predictions for the global $v-$direction wind starting from $2018-01-01-00:00$ to $2018-01-08-08:00$. Ground truth are contrasted with predictions from KAE, KKR, PFNN, and our method. Error maps in the lower panels demonstrate that, compared with other models, showing with more stable and accurate results of our model.

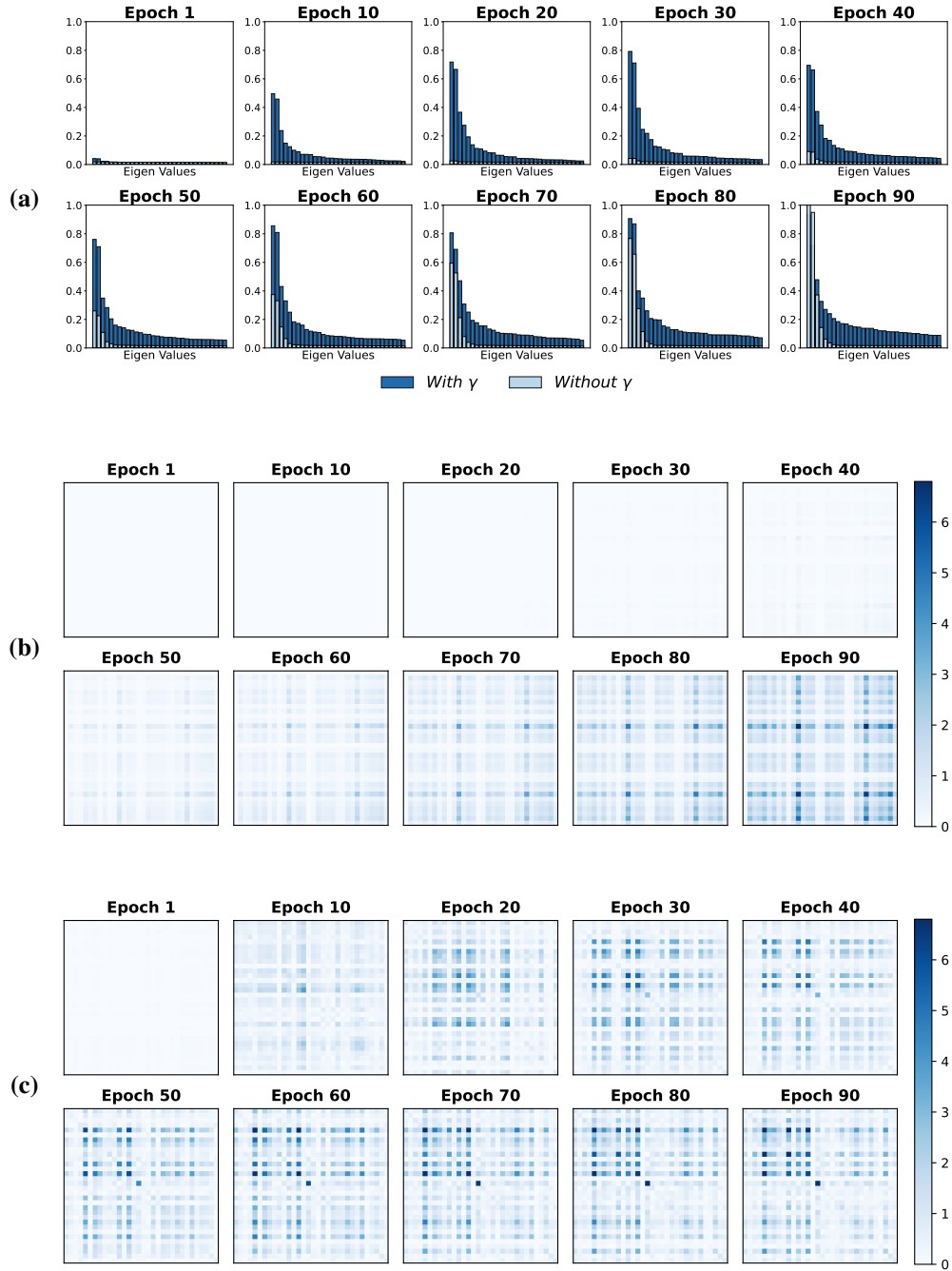

Figure 20: Comparison between $\gamma = 0.0$ and $\gamma = 0.5$ on the physical simulation task with latent dimension 32. **(a)** Evolution of the eigenvalue spectrum over training epochs. **(b)** Heatmap of the latent covariance matrix for $\gamma = 0.0$ across epochs. **(c)** Heatmap of the latent covariance matrix for $\gamma = 0.5$ across epochs. With the addition of the von Neumann entropy regularizer, two clear effects emerge during training. First, the latent covariance matrix no longer collapses to a few dominant modes: the eigenvalue distribution becomes more uniform and remains close to full rank throughout optimization (see **(a)**), indicating that the model learns a richer set of modes rather than compressing them into a low-dimensional subspace. Second, the covariance structure transitions from highly sparse (when $\gamma = 0$) to dense and full-rank under entropy regularization (from **(b)** to **(c)**), verifying our theoretical prediction in Proposition 5.

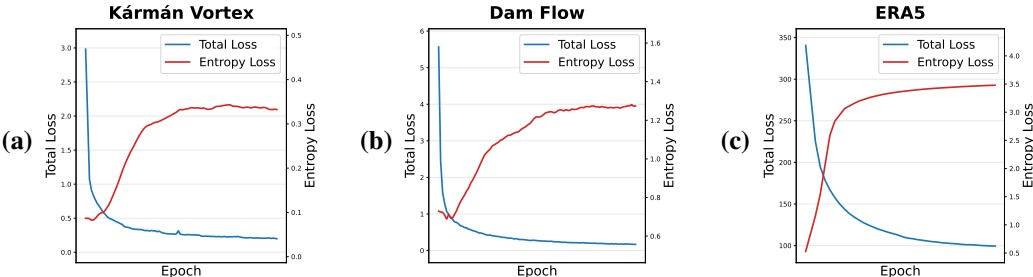

Figure 21: Visualization of the von Neumann entropy regularization loss and the total training loss over epochs for the physical simulation tasks. The stable behavior of both the total loss and the von Neumann entropy loss indicates that our training procedure is numerically stable and robust across different systems.

### G.5.2 VISUAL PERCEPTION

Table 8: Percentage to Goal (%) for different algorithms under noisy rollout. A higher value indicates that the system reaches closer to the goal within a fixed number of control steps.

| Domain | E2C | PCC | KAE | Ours |
|---|---|---|---|---|
| Planar ($n = 40 \times 40$) | 6.2 (1.5) | 34.8 (3.6) | 5.1 (1.2) | **39.6 (2.8)** |
| Pendulum ($n = 48 \times 40 \times 2$) | 45.5 (3.9) | 59.8 (3.2) | 26.3 (2.8) | **62.7 (2.9)** |
| Cartpole ($n = 80 \times 80 \times 2$) | 8.1 (1.6) | 53.1 (3.5) | 57.2 (3.8) | **61.9 (3.0)** |
| 3-link ($n = 80 \times 80 \times 2$) | 5.0 (1.0) | **21.3 (1.9)** | 2.1 (0.6) | 19.5 (2.0) |

Table 9: Percentage to Goal (%) for different algorithms under noiseless rollouts.

| Domain | E2C | PCC | KAE | Ours |
|---|---|---|---|---|
| Planar ($n = 40 \times 40$) | 37.8 (3.5) | 71.4 (0.6) | 12.2 (1.4) | **73.8 (0.5)** |
| Pendulum ($n = 48 \times 40 \times 2$) | 88.5 (0.6) | 90.1 (0.5) | 68.2 (1.9) | **91.5 (0.4)** |
| Cartpole ($n = 80 \times 80 \times 2$) | 39.5 (3.3) | 94.1 (1.6) | 98.5 (0.3) | **97.6 (1.2)** |
| 3-link ($n = 80 \times 80 \times 2$) | 21.5 (0.9) | **48.5 (1.6)** | 11.8 (0.8) | 47.9 (1.4) |

### G.5.3 GRAPH-STRUCTURED DYNAMICS

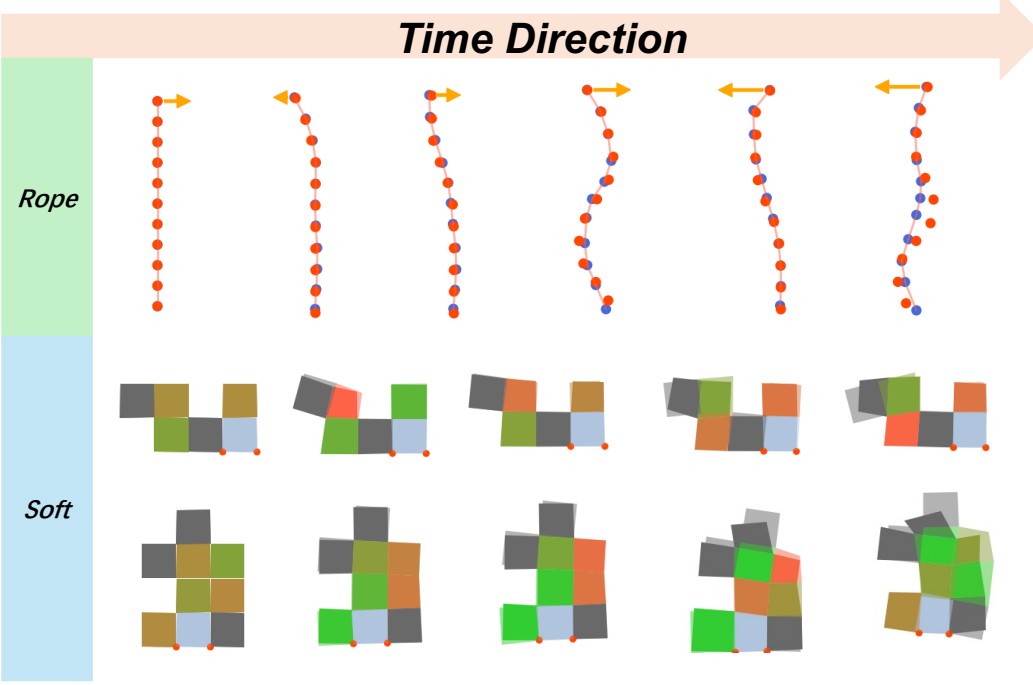

Figure 22: Comparison of ground truth and our predictions over time for rope and soft-body dynamics. Top row (Rope): red dots indicate ground truth positions, while blue dots show our predicted trajectories. Middle and bottom rows (Soft): translucent shapes represent ground truth deformations, and solid colored blocks denote our predictions. The time axis progresses from left to right.

### G.5.4 MODEL ARCHITECTURE

Table 10: Model architecture for Kármán Vortex and Dam Flow task with input dimension $(C, H, W)$, where $C$ denotes the number of velocity components and $H \times W$ is the spatial resolution.

| Components | Layer | Layer number | $C, (H, W)$ | Activation |
|---|---|---|---|---|
| Encoder | Convolution Block | 1 | $C \to 8C, (\frac{H}{2}, \frac{W}{2})$ | ReLU |
| | Convolution Block | 3 | $8C \to 64C, (\frac{H}{16}, \frac{W}{16})$ | ReLU |
| | Convolution2d | 1 | $64C \to 128C, (\frac{H}{16}, \frac{W}{16})$ | ReLU |
| | Flatten | – | $128C, (\frac{H}{16}, \frac{W}{16}) \to \frac{CHW}{2}$ | – |
| | Fully Connected | 1 | $\frac{CHW}{2} \to d_s$ | – |
| Koopman Operator | Linear | 1 | $d_s \to d_s$ | – |
| Decoder | Fully Connected | 1 | $d_s \to \frac{CHW}{2}$ | ReLU |
| | Transpose | – | $\frac{CHW}{2} \to (128C, \frac{H}{16}, \frac{W}{16})$ | – |
| | ConvTranspose Block | 3 | $128C \to 8C, (H, W)$ | ReLU |
| | ConvTranspose2d | 1 | $8C \to C, (H, W)$ | – |
| | Conv2d Refinement | 3 | $C \to C, (H, W)$ | ReLU |

Table 11: Model architecture for ERA5 task with input dimension $(C, H, W)$, where $C$ denotes the number of channels and $H \times W$ is the spatial resolution, using a factorized-attention encoder.

| Components | Layer | Layer number | $C, (H, W)$ | Activation |
|---|---|---|---|---|
| Encoder | Conv2d | 1 | $C \to \frac{64C}{5}, (H, W)$ | – |
| | Conv2d | 1 | $\frac{64C}{5} \to \frac{64C}{5}, (\frac{H}{4}, \frac{W}{4})$ | – |
| | FactorizedBlock | 1 | $\frac{64C}{5}, (\frac{H}{4}, \frac{W}{4})$ | GELU |
| | Flatten | – | $\frac{64C}{5}, (\frac{H}{4}, \frac{W}{4}) \to \frac{4CHW}{5}$ | – |
| | Fully Connected | 1 | $\frac{4CHW}{5} \to d_s$ | – |
| Koopman Operator | Linear | 1 | $d_s \to d_s$ | – |
| Decoder | Fully Connected | 1 | $d_s \to \frac{4CHW}{5}$ | ReLU |
| | Transpose | – | $\frac{4CHW}{5} \to (\frac{64C}{5}, \frac{H}{4}, \frac{W}{4})$ | – |
| | ConvTranspose2d | 2 | $\frac{64C}{5} \to \frac{64C}{5}, (H, W)$ | ReLU |
| | Conv2d | 1 | $\frac{64C}{5} \to C, (H, W)$ | – |
| | Conv2d Refinement | 3 | $C \to C, (H, W)$ | ReLU |

For fair comparison, the architectures for visual input and graph-structured dynamics follow the settings in Levine et al. (2020) and Li et al. (2020).

Table 12: Hyperparameter settings across physical simulations, visual inputs, and graph-structured dynamics

| Parameter | Symbol | Physical Sim. | Visual Inputs | Graph Dyn. |
|---|---|---|---|---|
| Temporal coherence | $\alpha$ | 2.00 | 3.00 | 2.00 |
| Structural consistency | $\beta$ | – | 2.00 | – |
| von Neumann entropy | $\gamma$ | 0.10 | 0.50 | 0.10 |
| InfoNCE neighborhood | $k$ | 3 | 5 | 5 |

### G.6 BASELINE ALGORITHMS

**Physical Simulation Tasks.**

- **VAE** (Kingma et al., 2013): Baseline implemented using a standard variational autoencoder with a nonlinear forward map in latent space. Code available at `https://github.com/bvezilic/Variational-autoencoder`.
- **KAE** (Pan et al., 2023): Koopman learning with an autoencoder architecture. Code available at `https://github.com/dynamicslab/pykoopman`.
- **KKR** (Bevanda et al., 2023): For the low-dimensional Lorenz–63 system, we adopt fixed kernel functions as basis following the implementation in `https://github.com/TUM-ITR/koopcore`. For high-dimensional systems, we use deep kernel features following Yang et al. (2025), with code available at `https://github.com/yyimingucl/TensorVar/blob/main/model/KS_model.py`.
- **PFNN** (Cheng et al., 2025): A state-of-the-art Koopman variant for learning and predicting chaotic dynamics. Code available at `https://github.com/Hy23333/PFNN`.

**Visual Inputs.**

- **E2C** (Banijamali et al., 2019): A latent embedding approach based on the VAE framework. Code available at `https://github.com/ericjang/e2c`.
- **KAE** (Pan et al., 2023): Koopman learning with an autoencoder architecture. Code available at `https://github.com/dynamicslab/pykoopman`.
- **PCC** (Banijamali et al., 2019): A state-of-the-art latent embedding algorithm also based on the VAE framework. Code available at `https://github.com/VinAIResearch/PCC-pytorch/tree/master/sample_results`.

**Graph-Structured Dynamics.**

- **CKO** (Li et al., 2020): A Koopman-based framework for learning and predicting general graph-structured dynamics. Code available at `https://github.com/YunzhuLi/CompositionalKoopmanOperators`.

