# OpenReview forum: "Information Shapes Koopman Representation"
_ICLR.cc/2026/Conference — ICLR 2026 Oral_

### Official Review · Reviewer_hgxb · 2025-10-18

**Soundness:** 2
**Presentation:** 2
**Contribution:** 2
**Rating:** 6
**Confidence:** 2

**Summary:**

This paper presents an information-theoretic framework for learning Koopman representations of nonlinear dynamical systems.In this paper, authors proposed a unified framework that balances simplicity and expressiveness in learning finite-dimensional Koopman representation.

**Strengths:**

(1) The proposed information-theoretic Lagrangian provides a clear mathematical formulation for balancing simplicity and expressiveness of predictive sufficiency;
(2) The information decomposition in Proposition 3 connecting mutual information components to Koopman spectral properties is insightful. This bridges dynamical systems theory with information theory smoothly;
(3) Experiments across three diverse domains demonstrates broad practical application.

**Weaknesses:**

(1) While the paper provides asymptotic and information-theoretic insights, it lacks finite-sample guarantees or non-asymptotic convergence results;
(2) The practical loss in Eq. 9 is a little bit heavy, with many tunable hyperparameters $\alpha, \beta, \gamma$. It is unclear how sensitive performance is to these values;
(3) Missing comparisons or discusstion of the following recent deep Koopman methods:
  (a) https://www.nature.com/articles/s41467-018-07210-0;
  (b) https://openreview.net/pdf?id=Svk7jjhlSu;
  (c) https://pubs.aip.org/aip/cha/article/27/10/103111/151485;
  (d) https://www.sciencedirect.com/science/article/abs/pii/S0021999124004431;
  (e) https://link.springer.com/article/10.1007/s00332-019-09567-y.

**Questions:**

(1) How sensitive are the results to the choice of von Neumann entropy regularization coefficient $\gamma$? What happens if you set $\gamma = 0$ (no entropy regularization) on all tasks quantitatively?
(2) The current formulation involves covariance and entropy computations, then how does the computational cost scale for high-dimensional latent spaces?
(3) How does your approach perform on stochastic dynamical systems, where the Koopman operator is no longer deterministic setting?
(4) The InfoNCE computation treats temporal neighbors as positives. But if the system has $T$-periodic dynamics, shouldn't $z_{T}$ also be a positive? How do you handle periodicity?

---

> ### Author Response · Authors · 2025-11-20
> **Response 1**
>
> We thank the reviewer for the constructive feedback and for recognizing our theoretical contributions. We appreciate the positive remarks on our information-theoretic Lagrangian, the spectral interpretation in Proposition 3, and the broad empirical evaluation.
>
>
> **weakness 1** `While the paper provides asymptotic and information-theoretic insights, it lacks finite-sample guarantees or non-asymptotic convergence results; `
>
> We thank the reviewer for raising this question. Our primary goal in this paper is to provide an information-theoretic perspective for improving Koopman representations in deep architectures, rather than establishing full non-asymptotic convergence theory. Directly proving sample-complexity guarantees for deep architecture seems very chanllenging. A natural future direction is to combine kernel tricks with information theory, as explored in recent works [1–4]. We view extending our framework along this line as a promising avenue for future research.
>
> Empirically, we observe stable convergence across datasets, as evidenced in the revision by the training curves and covariance evolution plots (Figures 20 and 21).
>
> [1] Kostic, Vladimir R., et al. "Learning invariant representations of time-homogeneous stochastic dynamical systems." arXiv preprint arXiv:2307.09912 (2023).
>
> [2] Kostic, Vladimir, et al. "Neural conditional probability for uncertainty quantification." Advances in Neural Information Processing Systems 37 (2024): 60999-61039.
>
> [3] Kostic, Vladimir R., et al. "Consistent long-term forecasting of ergodic dynamical systems." Forty-first International Conference on Machine Learning. 2024.
>
> [4] Bach, Francis. "Information theory with kernel methods." IEEE Transactions on Information Theory 69.2 (2022): 752-775.
>
>
> **weakness 2** `The practical loss in Eq. 9 is a little bit heavy, with many tunable hyperparameters alpha, beta, gamma; It is unclear how sensitive performance is to these values;`
>
> We thank the reviewer for this question. In practice, the algorithm is not highly sensitive to hyperparameter choices. Our experiments show that all coefficients admit a broad stable range, and only extreme values degrade performance. Below we summarize both (i) necessity and (ii) robustness of each term.
>
> **The Necessity of the Lagrangian** The core finding is each Lagrangian term is necessary. Removing any term as $\alpha = 0, \beta = 0, \gamma = 0$ and $k =1$ results in a bad performance (see Table below). This confirms our theoretical claim that all three components are necessary for learning Koopman representation under the information-theoretical view.
>
> **Robustness in a Wide Parameter Range.** the algorithm demonstrates robustness in performance across a wide parameter range around the optimum:
>
> > The optimal performance for $\alpha$ (Temporal Coherence) is sustained across a broad range $[1,5]$,
>
> > $\beta$ (Structural Consistency) shows stable performance in the range $[1, 2.5]$,
>
> > The entropy term $\gamma$ is stable in $[0.2, 1]$,
>
> > The neighborhood size $k$ is effective within $[3, 10]$.
>
> These broad plateaus indicate that the algorithm does not require delicate hyperparameter tuning.  We also use the same set of hyperparameters across all environments, without any per-task tuning.
>
> _Table: The tables below present the performance metric "Percentage to Goal (%)" on the Planar task, showing how control success rate varies when each multiplier is adjusted while others are kept optimal ($\alpha = 3, \beta = 2, \gamma = 0.5, k =5$). The optimal results are highlighted in bold._
>
> | Domain | $\alpha = 0$ | $\alpha = 0.5$ | $\alpha = 1$ | $\alpha = 3$ | $\alpha = 5$ | $\alpha = 7$ |
> |-|-|-|-|-|-|-|
> | Planar (n = 40×40) | 7.0 (2.1) | 18.2 (3.5) | 33.1 (2.6) | **39.6 (2.8)** | 37.8 (2.9) |  10.2 (1.7) |
>
>
> | Domain | $\beta = 0$ | $\beta = 0.5$ | $\beta = 1$ | $\beta = 2$ | $\beta = 2.5$ | $\beta = 3$ |
> |-|-|-|-|-|-|-|
> | Planar (n = 40×40) | 8.3 (2.3) | 25.4 (3.2) | 34.4 (2.7) | **39.6 (2.8)** | 33.7 (3.0) | 19.5 (2.8) |
>
>
> | Domain | $\gamma = 0$ | $\gamma = 0.2$ | $\gamma = 0.5$ | $\gamma = 1$ | $\gamma = 1.5$ | $\gamma = 2$ |
> |-|-|-|-|-|-|-|
> | Planar (n = 40×40) | 9.1 (2.2) | 27.1 (3.0) | **39.6 (2.8)** | 34.7 (2.7) | 23.1 (3.1) | 12.4 (2.4) |
>
> | Domain | $k = 1$ | $k = 3$ | $k = 5$ | $k = 10$ | $k = 15$ | $k = 20$ |
> |-|-|-|-|-|-|-|
> | Planar (n = 40×40) | 14.2 (2.9) | 31.8 (2.7) | **39.6 (2.8)** | 25.9 (2.8) | 16.4 (3.1) | 11.2 (2.5) |

---

> > ### Author Response · Authors · 2025-11-20
> > **Response 2**
> >
> > **weakness 3** `Missing comparisons or discusstion of the following recent deep Koopman methods`
> >
> > We thank the reviewer for providing more Koopman relevant works. We have added a discussion of all five papers in the revised literature review. These methods represent important advances in deep Koopman learning, and we appreciate the opportunity to clarify how our approach differs.
> >
> > [1] Learns nonlinear observables with deep autoencoders to approximate Koopman eigenfunctions.
> >
> > [2] Introduces Residual-based Koopman-learning Network (ResKoopNet) addresses this by explicitly minimizing the spectral residual to compute Koopman eigenpairs.
> >
> > [3] introduce a regularized EDMD approach that learns Koopman-invariant subspaces.
> >
> > [4] leverage neural operator techniques to learn Koopman operator for nonlinear PDE.
> >
> > [5] introduce a variational approch for learn Markovian process.
> >
> > In contrast, our contribution is fundamentally different: we provide the first information-theoretic formulation of Koopman representation learning, yielding (i) an information-theoretic error bound, (ii) a spectral decomposition of retained information, and (iii) a principled mutual information and von Neumann entropy Lagrangian that jointly enforces temporal coherence, structural simplicity, and spectral diversity. Our formulation is architecture-agnostic and complementary to these operator-specific methods.
> >
> >
> > [1] Lusch, Bethany, J. Nathan Kutz, and Steven L. Brunton. "Deep learning for universal linear embeddings of nonlinear dynamics." Nature communications 9.1 (2018): 4950.
> >
> > [2] Xu, Yuanchao, et al. "ResKoopNet: Learning Koopman Representations for Complex Dynamics with Spectral Residuals." arXiv preprint arXiv:2501.00701 (2025).
> >
> > [3] Li, Qianxiao, et al. "Extended dynamic mode decomposition with dictionary learning: A data-driven adaptive spectral decomposition of the Koopman operator." Chaos: An Interdisciplinary Journal of Nonlinear Science 27.10 (2017).
> >
> > [4] Xiong, Wei, et al. "Koopman neural operator as a mesh-free solver of non-linear partial differential equations." Journal of Computational Physics 513 (2024): 113194.
> >
> > [5] Wu, Hao, and Frank Noé. "Variational approach for learning Markov processes from time series data." Journal of Nonlinear Science 30.1 (2020): 23-66.
> >
> > **Question 1** `How sensitive are the results to the choice of von Neumann entropy regularization coefficient gamma`
> >
> > The answer is referred to `weakness 2`.

---

> > > ### Author Response · Authors · 2025-11-20
> > > **Response 3**
> > >
> > > **Question 2 and 3** `The current formulation involves covariance and entropy computations, then how does the computational cost scale for high-dimensional latent spaces? How does your approach perform on stochastic dynamical systems, where the Koopman operator is no longer deterministic setting?`
> > >
> > > We thank the reviewer for Questions 2 and 3. In brief, our method is computationally efficient and scales well to high-dimensional and stochastic dynamical systems. We address each point below.
> > >
> > > **Computational cost in high-dimensional latent spaces**
> > >
> > > Our proposed algorithm is computationally efficient and easy to scale. We ran extensive experiments on high-dimensional systems, and the overhead from the von Neumann entropy term is minimal. Table a (included in the revision) provides direct evidence: the additional entropy component adds only a small constant cost per epoch.
> > >
> > >
> > > _**Table a:** Per-epoch training time for different models across three tasks. For our method, the additional von Neumann entropy term introduces only a small constant overhead, showing that the formulation remains computationally efficient in practice (included as Table 7 in our paper)._
> > >
> > > | Task              | Metric                     | VAE               | KAE               | KKR               | PFNN              | Ours               |
> > > |-------------------|-----------------------------|-------------------|-------------------|-------------------|-------------------|---------------------|
> > > | **Kármán vortex** | Total Epoch time (s)              | 172.94 ± 4.11      | 195.26 ± 3.47      | 186.47 ± 1.86      | 182.52 ± 2.53      | 201.23 ± 1.07       |
> > > |                   | von Neumann entropy (s)     | --                | --                | --                | --                | 0.97 ± 0.13         |
> > > | **Dam Flow**      | Total Epoch time (s)              | 16.21 ± 0.29       | 16.95 ± 0.33       | 17.48 ± 0.29       | --                | 18.92 ± 0.36        |
> > > |                   | von Neumann entropy (s)     | --                | --                | --                | --                | 0.56 ± 0.08         |
> > > | **ERA5**          | Total Epoch time (s)              | --                | 240.20 ± 0.52      | 224.95 ± 0.54      | 242.33 ± 0.71      | 253.24 ± 2.05       |
> > > |                   | von Neumann entropy (s)     | --                | --                | --                | --                | 7.43 ± 0.64         |
> > >
> > >
> > >
> > > **Performance on high-dimensional and stochastic dynamical systems**
> > >
> > > To demonstrate scalability and robustness in noisy, stochastic settings, we added a new experiment on the large-scale ERA5 WeatherBench benchmark. Despite its high dimensionality and strong stochasticity, our method consistently outperforms strong Koopman baselines (KAE, KKR, PFNN) across all channels and forecast horizons (Table b). This shows that our formulation remains effective even when the underlying dynamics is not deterministic.
> > >
> > > _**Table b:** ERA5 forecasting results. WeatherBench is highly stochastic and high-dimensional. Our method achieves consistently lower NRMSE and higher SSIM across channels and forecast horizons. Full results and visualizations are included in the revision (see Table 6 and Figures 15-19)._
> > >
> > > | Channel         | Metric     | KAE              | KKR              | PFNN                 | Ours                          |
> > > |--|---|------|----------|-----------------------|-------------------------------|
> > > | **Geopotential** | 5-NRMSE    | 0.058 (0.020)     | 0.061 (0.027)     | 0.046 (0.012)         | **0.023 (0.005)**            |
> > > |                 | 10-NRMSE   | 0.068 (0.018)     | 0.074 (0.026)     | 0.062 (0.018)         | **0.032 (0.010)**            |
> > > |                 | 50-NRMSE   | 0.157 (0.071)     | 0.082 (0.025)     | 0.082 (0.016)         | **0.075 (0.017)**            |
> > > |                 | 5-SSIM     | 0.860 (0.054)     | 0.852 (0.064)     | 0.882 (0.036)         | **0.964 (0.011)**            |
> > > |                 | 10-SSIM    | 0.836 (0.051)     | 0.820 (0.065)     | 0.848 (0.052)         | **0.943 (0.028)**            |
> > > |                 | 50-SSIM    | 0.665 (0.195)     | 0.790 (0.049)     | 0.765 (0.055)         | **0.806 (0.039)**            |
> > > | **Temperature** | 5-NRMSE    | 0.049 (0.019)     | 0.052 (0.032)     | 0.040 (0.009)         | **0.022 (0.003)**            |
> > > |                 | 10-NRMSE   | 0.056 (0.017)     | 0.064 (0.028)     | 0.051 (0.015)         | **0.026 (0.006)**            |
> > > |                 | 50-NRMSE   | 0.114 (0.054)     | 0.074 (0.030)     | 0.067 (0.018)         | **0.063 (0.019)**            |
> > > |                 | 5-SSIM     | 0.866 (0.046)     | 0.862 (0.058)     | 0.888 (0.026)         | **0.956 (0.008)**            |
> > > |                 | 10-SSIM    | 0.844 (0.044)     | 0.829 (0.063)     | 0.859 (0.042)         | **0.942 (0.019)**            |
> > > |                 | 50-SSIM    | 0.671 (0.216)     | 0.802 (0.059)     | 0.803 (0.051)         | **0.825 (0.042)**            |

---

> > > > ### Author Response · Authors · 2025-11-20
> > > > **Response 4**
> > > >
> > > > **Question 4** `The InfoNCE computation treats temporal neighbors as positives. But if the system has $T$-periodic dynamics, shouldn't also be a positive? How do you handle periodicity?`
> > > >
> > > > We thank the reviewer for this insightful question. Our implementation follows the common InfoNCE setup in which only local temporal neighbors are treated as positives. For systems with known T-periodicity, latent states separated by one period could indeed be added as additional positives, and our formulation supports such an extension. We focus on non-periodic and stochastic systems in our experiments, where such periodic positives do not provide additional benefit.
> > > >
> > > > ----
> > > >
> > > > Many thanks again for your time and consideration, please let us know if we have addressed the concerns and increased your confidence in our work.

---

> > > > ### Comment · Reviewer_hgxb · 2025-11-22
> > > >
> > > > In the new added experimental results, I can tell the results by using your proposed method are much better compared to the baselines. Also, the climate science is also a very challenging but interesting case for Koopman operator application.
> > > >
> > > > By the way, what is the dimension of your data in this new experiment?

---

> > ### Comment · Reviewer_hgxb · 2025-11-22
> >
> > The results in Figure 21 seem very convincing. So, what is the connection between the kernel methods and your proposed information theory framework?

---

> ### Author Response · Authors · 2025-11-22
>
> We thank the reviewer for the positive feedback and for the thoughtful question.
>
> The connection between kernel methods and our information-theoretic framework is that we represent probability distributions via their covariance operators in a reproducing kernel Hilbert space (RKHS), which serve as density operators in the information-theoretic sense [1]. This construction enables us to define von Neumann entropy and KL divergence directly on these operators, thereby extending classical information measures through kernel embeddings.
>
> Building on pioneering work on the Koopman operator and RKHS methods [2–4], the sample complexity of proposed Lagrangian in our framework can be more rigorously analyzed using both kernel techniques and the operator-based information-theoretic tools developed in [1]. Exploring this theoretical connection is an exciting direction that we are actively pursuing.
>
> In addition, our extended weather experiment is from real-world data includes geopotential, temperature, humidity, u-wind, and v-wind, with dimensions **10240**.
>
> [1] Bach, Francis. "Information theory with kernel methods." IEEE Transactions on Information Theory 69.2 (2022): 752-775.
>
> [2] Kostic, Vladimir R., et al. "Learning invariant representations of time-homogeneous stochastic dynamical systems." arXiv preprint arXiv:2307.09912 (2023).
>
> [3] Kostic, Vladimir, et al. "Neural conditional probability for uncertainty quantification." Advances in Neural Information Processing Systems 37 (2024): 60999-61039.
>
> [4] Kostic, Vladimir R., et al. "Consistent long-term forecasting of ergodic dynamical systems." Forty-first International Conference on Machine Learning. 2024.

---

> > ### Comment · Reviewer_hgxb · 2025-11-22
> >
> > Thanks for all above answers. I think that the paper presents an interesting perspective of estimating the Koopman operator via information theory. I feel that the theoretical analysis is solid, and the empirical results from a variety of challenging dynamical systems are also convincing, especially the high dimensional climate science application. Overall, I find the contribution meaningful, and I am comfortable raising my score and confidence.

---

> ### Author Response · Authors · 2025-11-22
> **Thanks comments from Reviewer hgxb**
>
> Dear Reviewer hgxb,
>
> We sincerely appreciate the reviewer’s careful reading and positive assessment of our work. We are glad that the information-theoretic perspective for estimating the Koopman operator was found to be interesting, and we thank the reviewer for recognizing the strength of our theoretical analysis and the empirical results across challenging dynamical systems, including the high-dimensional climate application. We are grateful for the reviewer’s willingness to raise their score and confidence, which is very encouraging for us :).

---

### Official Review · Reviewer_KxuT · 2025-10-31

**Soundness:** 2
**Presentation:** 3
**Contribution:** 3
**Rating:** 6
**Confidence:** 3

**Summary:**

The paper proposes a new representation learning scheme based on the Information Bottleneck framework for learning Koopman operators. It addresses a common flaw in current methods, mode collapse, by introducing a novel information-theoretic Lagrangian that balances latent mutual information (for simplicity) against von Neumann entropy (to ensure expressiveness). The paper experimentally evaluates the proposed algorithm across a range of dynamical systems, demonstrating improved performance against baseline methods.

**Strengths:**

The paper addresses an important challenge in data-driven dynamical systems: learning stable, interpretable, and expressive finite-dimensional representations of the Koopman operator. The authors see the problem through a general, principled, and information-theoretic approach to manage the simplicity-expressiveness trade-off. The latent mutual information ($I(z_{t-n}; z_t)$) promotes temporal coherence but can lead to mode collapse, while von Neumann entropy ($S(\mathcal{C})$) promotes expressiveness and spectral diversity, actively preventing such collapse. This leads to a novel information-theoretic Lagrangian objective for Koopman learning. The paper is well-written and logically organized, making it easy to follow and comprehend. Its clarity is further enhanced by the inclusion of informative diagrams.

**Weaknesses:**

1. The paper's core premise is that standard representation learning (like VAEs) is insufficient for Koopman models because it prioritizes reconstruction over the more restrictive constraint of a linear predictive structure (structural consistency). While this claim is valid, the paper frames this as a relatively novel insight, overlooking a significant body of existing literature that has already identified and addressed this exact problem. For instance, works on VAMPNets (arXiv:1710.06012)  and, more recently, DPNets (	arXiv:2307.09912) are explicitly designed to learn representations that linearize the dynamics, moving far beyond simple reconstruction. By not citing or benchmarking against these more relevant baselines (as well as other related works, such as arxiv:2309.07200), the paper overstates the novelty of its problem formulation, making it difficult to assess how this new solution compares to other state-of-the-art methods targeting the same fundamental challenge.

2. There is a gap between the "pure" information-theoretic Lagrangian (Eq. 8) and the tractable loss function used for optimization (Eq. 9). The practical implementation relies on several approximations:

- The latent mutual information $I(z_{t-n}; z_t)$ is approximated using the InfoNCE contrastive loss, which is a lower bound, not the true MI.
- The von Neumann entropy $S(\frac{\mathcal{C}}{tr(\mathcal{C})})$ is calculated using a stochastic approximation of the covariance matrix $\mathcal{C}$ derived from a single minibatch (Algorithm 1, line 7). This estimate is likely to be very noisy and may introduce high variance into the gradients, potentially affecting training stability and the quality of the final manifold.

3. As the authors state in their appendix (Appendix D), the framework does not yet provide formal guarantees on sample complexity or the non-asymptotic convergence of the representation. It is unclear how many data samples are needed to reliably estimate the information-theoretic quantities (especially the minibatch-based entropy) and converge to a meaningful Koopman subspace.


_Minor_:

- In Eq. 14, line 1003, I believe that it should be $I(x_{t−1}; x_t) − I(z_{t−1}; z_t)$.
- Figure 7 is a bit misleading, in the decoding phase, as it is mentioned in the caption of the figure, “ The latent variables are subsequently decoded back to approximate the original states,” but in the diagram, it goes back to the original state. Therefore, one could wonder why $I(z_{t-1}; x_t | z_t)$ is not zero.

**Questions:**

1. How sensitive the algorithm is with respect to hyperparameter optimization? I haven't found any discussion on hyperparameter selection; is there any intuition or heuristic for choosing (k, alpha, beta, gamma)? Especially since the ablation study in Figure 5 clearly demonstrates that the balance between these terms is critical to success.
2. In the abstract, the Authors noted that KO's infinite-dimensional nature makes finding a good finite-dimensional subspace challenging, especially for deeper architectures. Could you elaborate on why deeper architectures tend to suffer more?
3. At the beginning of section 3.2, the authors said, “The latent mutual information quantifies the magnitude of error, but not the nature of the information lost in the Koopman representation.”, which sounds peculiar to me. Could you elaborate on what you mean by “nature of the information lost”?
4. I don’t understand the meaning of the red and blue arrows in Proposition 3. Does that mean we want to maximize the blue and minimize the red?

---

> ### Author Response · Authors · 2025-11-20
> **Response 1**
>
> We thank reviewer KxuT for the thoughtful review, constructive feedback, and recognition of our work’s strengths.
>
>
> **Weakness 1** `The paper's core premise is that standard representation learning (like VAEs) is insufficient for Koopman models because it prioritizes reconstruction over the more restrictive constraint of a linear predictive structure (structural consistency). While this claim is valid, the paper frames this as a relatively novel insight, overlooking a significant body of existing literature that has already identified and addressed this exact problem. For instance, works on VAMPNets (arXiv:1710.06012) and, more recently, DPNets ( arXiv:2307.09912) are explicitly designed to learn representations that linearize the dynamics, moving far beyond simple reconstruction. By not citing or benchmarking against these more relevant baselines (as well as other related works, such as arxiv:2309.07200), the paper overstates the novelty of its problem formulation, making it difficult to assess how this new solution compares to other state-of-the-art methods targeting the same fundamental challenge.`
>
> We thank the reviewer for this helpful observation. We first clarify the scope of our work: **our contribution is not about representation learning in general VAEs or Markov dynamics, but specifically about Koopman representation learning from an information-theoretic perspective**. All theoretical results, propositions, and the Lagrangian formulation are derived within the Koopman framework rather than the general Markov setting.
>
> Methods such as VAMPnets [1] and DPNets [2] indeed aim to obtain linear embeddings for Markvian or stochastic dynamics, and we have added both papers to the revised related work section. However, **linearization itself is not the novelty we claim**.
>
> Regarding arXiv:2309.07200 [3], this paper focuses on learning latent Markov processes via a Time-lagged Information Bottleneck objective, with experiments centered on reproducing _invariant distributions_ of Markov dynamics. It does not target Koopman representations, and _no open-sourced code is available_. We have therefore cited and discussed it as related work, but it is not an appropriate baseline for Koopman operator learning.
>
> **Our contribution lies elsewhere:** we provide the information-theoretic formulation of Koopman representation learning, addressing a different question:
>
> _Is it possible to learn Koopman representations that are both structurally simple and expressive, under the guidance of information-theoretic principles?_
>
> The key contributions of our paper distinct from VAMPnets, DPNets, and other variational Markov modeling approaches are:
>
> > Information-theoretic error characterization (Propositions 1–2): We identify a quantifiable latent information gap as the root cause of multi-step Koopman prediction error, yielding the information-theoretic error bound for Koopman representation.
>
> > Spectral decomposition of information (Proposition 3): We show that retained information aligns exactly with Koopman spectral modes (coherent, dissipating, residual), revealing which modes are preserved or lost, not merely whether the dynamics are linear.
>
> > Principled mutual information and von Neumann entropy tradeoff (Propositions 4–5): We prove that mutual information encourages simplicity (few coherent modes) while von Neumann entropy prevents spectral mode collapse, yielding a general Koopman Lagrangian that applies across architectures.
>
> Empirically, our framework works across diverse dynamical systems (physical systems, graph and visual inputs), including high-dimensional settings like ERA5 WeatherBench (see new Table 6 and Figures 15-19 in our revised version), suggesting that the information-theoretic perspective is broadly applicable in practice.
>
> [1] Mardt, Andreas, et al. "VAMPnets for deep learning of molecular kinetics." Nature communications 9.1 (2018): 5.
>
> [2] Kostic, Vladimir R., et al. "Learning invariant representations of time-homogeneous stochastic dynamical systems." arXiv preprint arXiv:2307.09912 (2023).
>
> [3] Federici, Marco, et al. "Latent representation and simulation of markov processes via time-lagged information bottleneck." arXiv preprint arXiv:2309.07200 (2023).

---

> ### Author Response · Authors · 2025-11-20
> **Response 2**
>
> **Weakness 2** `There is a gap between the "pure" information-theoretic Lagrangian (Eq. 8) and the tractable loss function used for optimization (Eq. 9). The practical implementation relies on several approximations:`
>
> We thank the reviewer for this point. The transition from the “pure” Lagrangian (Eq. 8) to the tractable loss (Eq. 9) is not a loose approximation; it is a standard variational instantiation. We replace only the intractable quantities with their established variational surrogates.
>
> **Mutual Information:** Eq. 8 uses the exact mutual information $I(z_{t-n}; z_t)$ whose closed-form (Eq. 7) is tractable in low-dimensional cases but becomes intractable in high-dimensional nonlinear settings. Eq. 9 adopts InfoNCE, a widely used and theoretically justified variational lower bound on mutual information. This substitution maintains the directionality of the objective, encouraging temporal coherence and information retention, while making optimization feasible at scale. In the revision, we additionally include training-time estimates of this term (see Table 7 in our paper), showing that the InfoNCE-based formulation remains stable and enables efficient scalability to large, high-dimensional systems.
>
> **Von Neumann Entropy:** Empirically, we observe stable entropy estimates with moderate batch size (our setting $B=256$), and we now provide training curves showing stability throughout optimization in Figure 21. Additionally, the covariance evolution plots in the revision demonstrate that this approximation effectively encourages spectral diversity (see Figure 20 in our paper): the covariance matrix maintains full support with our entropy regularization but collapses to rank-deficient structure when the term is removed. This matches exactly the role prescribed by Eq. 8.
>
>
> [1] Oord, Aaron van den, Yazhe Li, and Oriol Vinyals. "Representation learning with contrastive predictive coding." arXiv preprint arXiv:1807.03748 (2018).
>
>
> **Weakness 3** `As the authors state in their appendix (Appendix D), the framework does not yet provide formal guarantees on sample complexity or the non-asymptotic convergence of the representation. It is unclear how many data samples are needed to reliably estimate the information-theoretic quantities (especially the minibatch-based entropy) and converge to a meaningful Koopman subspace.`
>
> We thank the reviewer for raising this question. At present, our framework does not provide formal non-asymptotic guarantees on sample complexity, and we acknowledge this limitation. This is an interesting and important direction for future work.
>
> Our primary goal in this paper is to provide an information-theoretic perspective for improving Koopman representations in deep architectures, rather than establishing full non-asymptotic convergence theory. Directly proving sample-complexity guarantees for deep architecture seems very chanllenging. A natural future direction is to combine kernel tricks with information theory, as explored in recent works ([1–4], cited for our future direction). We view extending our framework along this line as a promising avenue for future research.
>
>
> Empirically, we observe stable convergence across datasets, as evidenced in the revision by the training curves and covariance evolution plots (Figures 20 and 21). In addition, the visualized manifolds constructed by our model is faithful to the true underlying structure (Figures 2, 3 and 5), demonstrating that it converges to a meaningful Koopman subspace.
>
>
>
> [1] Kostic, Vladimir R., et al. "Learning invariant representations of time-homogeneous stochastic dynamical systems." arXiv preprint arXiv:2307.09912 (2023).
>
> [2] Kostic, Vladimir, et al. "Neural conditional probability for uncertainty quantification." Advances in Neural Information Processing Systems 37 (2024): 60999-61039.
>
> [3] Kostic, Vladimir R., et al. "Consistent long-term forecasting of ergodic dynamical systems." Forty-first International Conference on Machine Learning. 2024.
>
> [4] Bach, Francis. "Information theory with kernel methods." IEEE Transactions on Information Theory 69.2 (2022): 752-775.
>
> **Minor 1**   `In Eq. 14, line 1003`.
>
> Thanks for indicating the typo. We have corrected it in the revision.
>
> **Minor 2** `Figure 7 is a bit misleading, in the decoding phase, as it is mentioned in the caption of the figure, “ The latent variables are subsequently decoded back to approximate the original states,” but in the diagram, it goes back to the original state. `
>
> Thanks for the valuable comment. We have indicated the dash arrow is the approximated result.

---

> ### Author Response · Authors · 2025-11-20
> **Response 3**
>
> **Question 1** `How sensitive the algorithm is with respect to hyperparameter optimization? I haven't found any discussion on hyperparameter selection; is there any intuition or heuristic for choosing (k, alpha, beta, gamma)? Especially since the ablation study in Figure 5 clearly demonstrates that the balance between these terms is critical to success.`
>
> We thank the reviewer for this question. In practice, the algorithm is not highly sensitive to hyperparameter choices. Our experiments show that all coefficients admit a broad stable range, and only extreme values degrade performance. Below we summarize both (i) necessity and (ii) robustness of each term.
>
> **The Necessity of the Lagrangian** The core finding is each Lagrangian term is necessary. Removing any term as $\alpha = 0, \beta = 0, \gamma = 0$ and $k =1$ results in a bad performance (see Table a). This confirms our theoretical claim that all three components are necessary for learning Koopman representation under the information-theoretical view.
>
> **Robustness in a Wide Parameter Range.** the algorithm demonstrates robustness in performance across a wide parameter range around the optimum:
>
> > The optimal performance for $\alpha$ (Temporal Coherence) is sustained across a broad range $[1,5]$,
>
> > $\beta$ (Structural Consistency) shows stable performance in the range $[1,2.5]$,
>
> > The entropy term $\gamma$ is stable in $[0.2, 1]$,
>
> > The neighborhood size $k$ is effective within $[3, 10]$.
>
> These broad plateaus indicate that the algorithm does not require delicate hyperparameter tuning.
>
> _Table a: The tables below present the performance metric "Percentage to Goal (%)" on the Planar task, showing how control success rate varies when each multiplier is adjusted while others are kept optimal ($\alpha = 3, \beta = 2, \gamma = 0.5, k =5$). The optimal results are highlighted in bold._
>
> | Domain | $\alpha = 0$ | $\alpha = 0.5$ | $\alpha = 1$ | $\alpha = 3$ | $\alpha = 5$ | $\alpha = 7$ |
> |-|-|-|-|-|-|-|
> | Planar (n = 40×40) | 7.0 (2.1) | 18.2 (3.5) | 33.1 (2.6) | **39.6 (2.8)** | 37.8 (2.9) |  10.2 (1.7) |
>
>
> | Domain | $\beta = 0$ | $\beta = 0.5$ | $\beta = 1$ | $\beta = 2$ | $\beta = 2.5$ | $\beta = 3$ |
> |-|-|-|-|-|-|-|
> | Planar (n = 40×40) | 8.3 (2.3) | 25.4 (3.2) | 34.4 (2.7) | **39.6 (2.8)** | 33.7 (3.0) | 19.5 (2.8) |
>
>
> | Domain | $\gamma = 0$ | $\gamma = 0.2$ | $\gamma = 0.5$ | $\gamma = 1$ | $\gamma = 1.5$ | $\gamma = 2$ |
> |-|-|-|-|-|-|-|
> | Planar (n = 40×40) | 9.1 (2.2) | 27.1 (3.0) | **39.6 (2.8)** | 34.7 (2.7) | 23.1 (3.1) | 12.4 (2.4) |
>
> | Domain | $k = 1$ | $k = 3$ | $k = 5$ | $k = 10$ | $k = 15$ | $k = 20$ |
> |-|-|-|-|-|-|-|
> | Planar (n = 40×40) | 14.2 (2.9) | 31.8 (2.7) | **39.6 (2.8)** | 25.9 (2.8) | 16.4 (3.1) | 11.2 (2.5) |
>
>
> **Question 2** `In the abstract, the Authors noted that KO's infinite-dimensional nature makes finding a good finite-dimensional subspace challenging, especially for deeper architectures. Could you elaborate on why deeper architectures tend to suffer more?`
>
> We thank the reviewer for the question. Deeper architectures tend to struggle in Koopman learning for two reasons. First, self-supervised deep encoders are optimized primarily for reconstruction and lack physical inductive bias, making deeper models more prone to learning latent spaces that capture short-term behavior but fail to preserve long-term dynamical structure. As shown by our vanilla Koopman autoencoder (KAE) results (see Figures 2, 3), this often leads to distorted or collapsed manifolds relative to the ground truth.
>
> Second, deeper AE/VAE architectures exhibit low effective latent dimension even when the nominal dimension is high. In the revised version, we include covariance matrix visualizations (Figure 20) showing that, without von Neumann entropy regularization, the covariance becomes sparse which indicating mode collapse. This prevents the latent space from representing a diverse set of Koopman modes.
>
> These effects together explain why deeper architectures are more prone to degraded Koopman representations without additional guidance, and why our framework is beneficial: the latent mutual information term preserves long-term coherent structure (see Figures 2,3 and 5), while the von Neumann entropy term encourages diverse modes and prevents collapse see illustration in Figure 20.

---

> > ### Author Response · Authors · 2025-11-20
> > **Response 4**
> >
> > **Question 3** `At the beginning of section 3.2, the authors said, “The latent mutual information quantifies the magnitude of error, but not the nature of the information lost in the Koopman representation.”, which sounds peculiar to me. Could you elaborate on what you mean by “nature of the information lost”?`
> >
> > Thanks for the question. By “the nature of the information lost,” we mean that Propositions 1–2 quantify the magnitude of information loss, but do not decompose this loss across Koopman spectral components. We have revised the sentence to:
> >
> > “The latent mutual information quantifies the magnitude of error, but does not uncover how this loss relates to Koopman spectral properties.”
> >
> > **Question 4** `I don’t understand the meaning of the red and blue arrows in Proposition 3. Does that mean we want to maximize the blue and minimize the red?`
> >
> > We thank the reviewer for this question. Yes, the arrows indicate our objective:
> >
> > 1. Blue arrows denote temporal-coherent components, which we aim to maximize because they correspond to stable, long-term Koopman modes.
> >
> > 2. Red arrows denote fast-dissipating or residual components, which we aim to minimize as they capture transient, non-predictive behavior.
> >
> >
> > ------
> > Many thanks again for your time and consideration, please let us know if we have addressed the concerns and increased your confidence in our work.

---

> > > ### Comment · Reviewer_KxuT · 2025-11-25
> > > **Comment to the rebuttal**
> > >
> > > Dear Authors, first of all, let me apologise for the late reply.
> > > Thank you for the detailed clarifications concerning the issues I've raised in my review. While most of my concerns have been addressed, I am still somewhat confused by your response to weakness 1. In my understanding, the formalism of the Koopman operator can be applied to _any_ non-autonomous Markovian system, where, by Markovian, I mean that future states depend only on the current state, and not on the past. This definition includes both deterministic and stochastic systems.  So when you write:
> > >
> > > > [...] the Lagrangian formulation are derived within the Koopman framework rather than the general Markov setting.
> > >
> > > I don't really see the difference between the Koopman framework and the "general Markov setting". I believe that clarifying this point further will help position this work more effectively within the broader representation learning literature for dynamical systems, as also all of [1, 2, 3] are derived using operatorial methods strongly linked to the Koopman operator itself.
> > >
> > > That being said, I think the overall work is solid and is an interesting representation learning method within the universe of Koopman-based methods. For this reason, I will keep a positive outlook, with increased confidence.
> > >
> > > [1] Mardt, Andreas, et al. "VAMPnets for deep learning of molecular kinetics." Nature communications 9.1 (2018): 5.
> > >
> > > [2] Kostic, Vladimir R., et al. "Learning invariant representations of time-homogeneous stochastic dynamical systems." arXiv preprint arXiv:2307.09912 (2023).
> > >
> > > [3] Federici, Marco, et al. "Latent representation and simulation of markov processes via time-lagged information bottleneck." arXiv preprint arXiv:2309.07200 (2023).

---

> ### Author Response · Authors · 2025-11-26
>
> We appreciate the reviewer's excellent follow-up question, which helps us articulate the precise theoretical positioning of our work.
>
> The key distinction is that while the underlying state dynamics is Markovian ($x_t = T(x_{t-1})$), the Koopman framework imposes a stricter structural constraint on the space where the latent states (a.k.a. observables) evolve. **While all Koopman representations satisfy the Markovian property, the reverse is not true: most Markovian models based on variational methods do not admit a latent space in which the dynamics evolve linearly in a function space. Our Lagrangian explicitly relies on this restriction, which is a structural assumption unique to the Koopman framework and not shared by general variational methods for Markovian dynamics.**  We elaborate below.
>
> **Koopman Operator Imposes Structure on the Observable Space.** The Koopman operator $\mathcal{K}$ lifts the nonlinear dynamics $T$ into a linear operator that acts on the space of observables $\mathcal{H}$ (typically Hilbert or general Banach spaces). This mathematical setting imposes restrictions:
>
> > **Space Constraint:** The dynamics are constrained to evolve linearly within this specific function space $\mathcal{H}$,
>
> > **Spectral Property:** Consequently, the analysis of the Koopman operator concerns its spectral properties (eigenfunctions and eigenvalues). This spectral structure has deep connections to information theory in Hilbert space, beyond merely checking the Markovian property of the underlying state.
>
> The requirement that the evolution is linear on the observables in a suitable function space (and thus governed by its spectral properties) is what makes the Koopman framework more structurally **restrictive** than the general Markovian setting. In this sense, research [1,2] is classified within this line of work, due to the linear representation in the latent function space.
>
>
> **Connection Between Information Theory and Koopman Operator.** Our information-theoretic approach capitalizes on these specific properties under Koopman representation:
>
> > **Latent Mutual Information** governs which spectral weights are allocated to temporally coherent modes (see Propositions 3-4),
>
> > **von Neumann entropy** determines the effective dimension by managing the spectral diversity (see Proposition 5).
>
> This connection between information and spectrum forms a fundamental theoretical connection that is specific to the Koopman formulation. Thus, our work is under the Koopman setting.
>
>
> **Contrast with General Variational Methods (in Markovian setting).** In contrast, methods that rely on general variational formulations [3] typically do not assume the latent space evolution to be linear, nor do they impose spectral constraints or function-space structure. In general Markovian settings, latent dynamics are often modeled as nonlinear maps, without requiring linear evolution in the space of observables. The structural restriction of Koopman linearity is therefore absent.
>
>
> ----
>
> Thank you again for your careful evaluation. We look forward to continued constructive exchange with you.

---

### Official Review · Reviewer_Ga5Z · 2025-10-31

**Soundness:** 2
**Presentation:** 3
**Contribution:** 2
**Rating:** 6
**Confidence:** 3

**Summary:**

The authors explore a new, information-theoretic approach to Koopman operator learning, which is used to model dynamical systems. They argue that existing methods struggle because latent variables often fail to balance expressivity (capturing rich dynamics) and simplicity (remaining compact and interpretable). Drawing parallels to the information bottleneck principle, they introduce a Lagrangian formulation that explicitly trades off these two goals. In this setup, latent mutual information promotes simplicity, while von Neumann entropy encourages diversity and avoids mode collapse. They propose a corresponding algorithm that stabilizes learning, improves interpretability, and yields better empirical performance on various dynamical system benchmarks.

**Strengths:**

S1. The paper presents a fresh conceptual perspective by linking Koopman learning with the information bottleneck framework, which deepens theoretical understanding and connects two previously distinct research areas.

S2. The proposed Lagrangian and algorithmic formulation is both principled and practical, demonstrating consistent empirical improvements and more interpretable latent structures.

S3. The paper provides a well-structured and thorough analysis of the information components underlying the Koopman operator, offering a clear theoretical justification for the proposed Lagrangian formulation.

S4. The visualizations of the learned manifolds are insightful and effectively illustrate how the proposed method improves representation quality and interpretability.

S5. The quantitative results show that the proposed approach consistently outperforms existing methods, supporting its practical advantages and robustness.

**Weaknesses:**

W1. The framework, while elegant, is conceptually dense and abstract, making it difficult to assess how easily it can be implemented or scaled to complex, high-dimensional real-world systems.

W2. The empirical validation, though broad, seems to focus mainly on illustrative examples—more rigorous or comparative testing on large-scale benchmarks would strengthen the claims of generality and robustness.

W3. While the information-theoretic framing is creative, the use of mutual information and entropy as tradeoff terms resembles existing regularization and variational methods, making the contribution partly incremental rather than entirely groundbreaking.

**Questions:**

Q0. Does the proposed approach still model a true Koopman operator, or has it effectively become an information-theoretic Lagrangian framework—based on mutual information and entropy—that only mimics Koopman behavior?

Q1. How sensitive is the proposed Lagrangian formulation to the relative weighting between mutual information and von Neumann entropy? In other words, does performance degrade noticeably if this balance is not carefully tuned?

Q2. Can the authors demonstrate the scalability of their method by applying it to higher-dimensional or real-world dynamical systems to validate its robustness beyond synthetic benchmarks?

Q3. Are there other approximate Koopman operator methods that could be compared against the proposed approach? It would be helpful to see how this method performs relative to alternatives that may more closely capture the underlying manifold, even if the proposed one achieves better overall results.

---

> ### Author Response · Authors · 2025-11-20
> **Response 1**
>
> We thank the reviewer for the thoughtful and constructive evaluation.
> We appreciate the positive remarks on the novelty of our information-theoretic perspective (S1), the principled nature and practical effectiveness of our Lagrangian formulation (S2–S3), as well as the clarity of our manifold visualizations and the consistent empirical gains (S4–S5). We are glad that the reviewer found our theoretical motivation and empirical results convincing, and we thank them for highlighting the interpretability benefits of the learned Koopman representations.
>
> **Weakness 1 and 2, Question 2** `The framework, while elegant, is conceptually dense and abstract, making it difficult to assess how easily it can be implemented or scaled to complex, high-dimensional real-world systems. The empirical validation, though broad, seems to focus mainly on illustrative examples—more rigorous or comparative testing on large-scale benchmarks would strengthen the claims of generality and robustness.`
>
>
> Thank you for the question. While the framework is theoretically motivated, its practical form is **simple and lightweight**. The full method reduces to adding three closed-form regularization terms to a AE/VAE backbone, with no architectural constraints or specialized training tricks. All components are modular, and their computation is explicitly summarized in Algorithms 1 and 2.
>
> To directly address scalability and rigor, we added a new experiment on the **large-scale and stochastic ERA5 WeatherBench benchmark**. Across all channels and forecast horizons (see results in Table a, or Table 6 and Figures 15-19) in our paper), our method consistently outperforms strong Koopman baselines (KAE, KKR, PFNN). This demonstrates that the framework is not only conceptually grounded but also **easy to implement, robust in practice, and scalable to realistic high-dimensional dynamical systems**.
>
> _**Table a:** Per-channel ERA5 forecasting results. WeatherBench is a highly stochastic and high-dimensional real-world system. Our method achieves consistently lower NRMSE and higher SSIM across channels and forecast horizons, demonstrating that the proposed formulation scales effectively to complex real-world dynamics. Full results and visualizations are provided in the revised version (Table 6 and Figures 15-19)._
>
> | Channel         | Metric     | KAE              | KKR              | PFNN                 | Ours                          |
> |-----------------|------------|------------------|------------------|-----------------------|-------------------------------|
> | **Geopotential** | 5-NRMSE    | 0.058 (0.020)     | 0.061 (0.027)     | 0.046 (0.012)         | **0.023 (0.005)**            |
> |                 | 10-NRMSE   | 0.068 (0.018)     | 0.074 (0.026)     | 0.062 (0.018)         | **0.032 (0.010)**            |
> |                 | 50-NRMSE   | 0.157 (0.071)     | 0.082 (0.025)     | 0.082 (0.016)         | **0.075 (0.017)**            |
> |                 | 5-SSIM     | 0.860 (0.054)     | 0.852 (0.064)     | 0.882 (0.036)         | **0.964 (0.011)**            |
> |                 | 10-SSIM    | 0.836 (0.051)     | 0.820 (0.065)     | 0.848 (0.052)         | **0.943 (0.028)**            |
> |                 | 50-SSIM    | 0.665 (0.195)     | 0.790 (0.049)     | 0.765 (0.055)         | **0.806 (0.039)**            |
> | **Temperature** | 5-NRMSE    | 0.049 (0.019)     | 0.052 (0.032)     | 0.040 (0.009)         | **0.022 (0.003)**            |
> |                 | 10-NRMSE   | 0.056 (0.017)     | 0.064 (0.028)     | 0.051 (0.015)         | **0.026 (0.006)**            |
> |                 | 50-NRMSE   | 0.114 (0.054)     | 0.074 (0.030)     | 0.067 (0.018)         | **0.063 (0.019)**            |
> |                 | 5-SSIM     | 0.866 (0.046)     | 0.862 (0.058)     | 0.888 (0.026)         | **0.956 (0.008)**            |
> |                 | 10-SSIM    | 0.844 (0.044)     | 0.829 (0.063)     | 0.859 (0.042)         | **0.942 (0.019)**            |
> |                 | 50-SSIM    | 0.671 (0.216)     | 0.802 (0.059)     | 0.803 (0.051)         | **0.825 (0.042)**            |

---

> > ### Author Response · Authors · 2025-11-20
> > **Response 2**
> >
> > **Weakness 3** `While the information-theoretic framing is creative, the use of mutual information and entropy as tradeoff terms resembles existing regularization and variational methods, making the contribution partly incremental rather than entirely groundbreaking.`
> >
> > We thank the reviewer for the comment. We respectfully clarify that our use of mutual information and entropy is not borrowed from generic variational regularization. Instead, these quantities arise from a new theoretical analysis tailored specifically to Koopman learning.
> >
> > **Novel Theoretical Contribution (Propositions 1–5):** This is, to our knowledge, the first work that rigorously connects information theory with Koopman representation learning. Our analysis provides three new insights:
> >
> > > Information loss as the root cause of Koopman errors (Proposition 1-2): We prove that multi-step prediction error is upper-bounded by a quantifiable latent information gap, establishing the first information-theoretic error bound for Koopman representations.
> >
> > > Spectral disentanglement of information (Proposition 3): We show that retained information decomposes exactly along Koopman spectral modes (coherent, dissipating, residual), forming a new bridge between information flow and Koopman spectra.
> >
> > > Principled tradeoff (Proposition 4-5): We prove that mutual information enforces simplicity by concentrating capacity on coherent modes, whereas von Neumann entropy prevents mode collapse by preserving spectral diversity.
> >
> > **General Algorithmic Framework:** These insights lead to a novel Lagrangian (Eq. 8) that explicitly balances temporal coherence, structural consistency, and predictive sufficiency. Importantly, the formulation is architecture-agnostic: it applies across physical simulation, weather forecasting, visual input and graph dynamics consistently outperforming ad-hoc regularization baselines.
> >
> >
> > **Question 0** `Does the proposed approach still model a true Koopman operator, or has it effectively become an information-theoretic Lagrangian framework—based on mutual information and entropy—that only mimics Koopman behavior?`
> >
> > We appreciate the reviewer’s question. Our method does not modify the true Koopman operator. The latent dynamics remain strictly linear and governed by a single Koopman operator $\mathcal{K}$, exactly as in standard Koopman learning (see Algorithms 1–2). The information-theoretic terms do not replace or approximate the operator; they only shape the representation in which $\mathcal{K}$ is learned.
> >
> > In other words, the operator remains a true Koopman operator, while mutual information encourages a compact representation by capturing the coherent information, and von Neumann entropy prevents spectral collapse.

---

> ### Author Response · Authors · 2025-11-20
> **Response 3**
>
> **Question 1**. `How sensitive is the proposed Lagrangian formulation to the relative weighting between mutual information and von Neumann entropy? In other words, does performance degrade noticeably if this balance is not carefully tuned?`
>
>
> We thank the reviewer for this question. In practice, the algorithm is not highly sensitive to hyperparameter choices. Our experiments show that all coefficients admit a broad stable range, and only extreme values degrade performance. Below we summarize both (i) necessity and (ii) robustness of each term.
>
> **The Necessity of the Lagrangian** The core finding is each Lagrangian term is necessary. Removing any term as $\alpha = 0, \beta = 0, \gamma = 0$ and $k =1$ results in a bad performance (see Table b). This confirms our theoretical claim that all three components are necessary for learning Koopman representation under the information-theoretical view.
>
> **Robustness in a Wide Parameter Range.** the algorithm demonstrates robustness in performance across a wide parameter range around the optimum:
>
> > The optimal performance for $\alpha$ (Temporal Coherence) is sustained across a broad range $[1,5]$,
>
> > $\beta$ (Structural Consistency) shows stable performance in the range $[1,2.5]$,
>
> > The entropy term $\gamma$ is stable in $[0.2, 1]$,
>
> > The neighborhood size $k$ is effective within $[3, 10]$.
>
> These broad plateaus indicate that the algorithm does not require delicate hyperparameter tuning. We also use the same set of hyperparameters across all environments, without any per-task tuning.
>
> _**Table b**: The tables below present the performance metric "Percentage to Goal (%)" on the Planar task, showing how control success rate varies when each multiplier is adjusted while others are kept optimal ($\alpha = 3, \beta = 2, \gamma = 0.5, k =5$). The optimal results are highlighted in bold._
>
> | Domain | $\alpha = 0$ | $\alpha = 0.5$ | $\alpha = 1$ | $\alpha = 3$ | $\alpha = 5$ | $\alpha = 7$ |
> |-|-|-|-|-|-|-|
> | Planar (n = 40×40) | 7.0 (2.1) | 18.2 (3.5) | 33.1 (2.6) | **39.6 (2.8)** | 37.8 (2.9) |  10.2 (1.7) |
>
>
> | Domain | $\beta = 0$ | $\beta = 0.5$ | $\beta = 1$ | $\beta = 2$ | $\beta = 2.5$ | $\beta = 3$ |
> |-|-|-|-|-|-|-|
> | Planar (n = 40×40) | 8.3 (2.3) | 25.4 (3.2) | 34.4 (2.7) | **39.6 (2.8)** | 33.7 (3.0) | 19.5 (2.8) |
>
>
> | Domain | $\gamma = 0$ | $\gamma = 0.2$ | $\gamma = 0.5$ | $\gamma = 1$ | $\gamma = 1.5$ | $\gamma = 2$ |
> |-|-|-|-|-|-|-|
> | Planar (n = 40×40) | 9.1 (2.2) | 27.1 (3.0) | **39.6 (2.8)** | 34.7 (2.7) | 23.1 (3.1) | 12.4 (2.4) |
>
> | Domain | $k = 1$ | $k = 3$ | $k = 5$ | $k = 10$ | $k = 15$ | $k = 20$ |
> |-|-|-|-|-|-|-|
> | Planar (n = 40×40) | 14.2 (2.9) | 31.8 (2.7) | **39.6 (2.8)** | 25.9 (2.8) | 16.4 (3.1) | 11.2 (2.5) |
>
>
> **Question 2**. `Can the authors demonstrate the scalability of their method by applying it to higher-dimensional or real-world dynamical systems to validate its robustness beyond synthetic benchmarks?`
>
> See answer for `Weakness 1 and 2`.
>
>
> **Question 3** `Are there other approximate Koopman operator methods that could be compared against the proposed approach? It would be helpful to see how this method performs relative to alternatives that may more closely capture the underlying manifold, even if the proposed one achieves better overall results.`
>
> We thank the reviewer for the question. While there exist some approaches for approximating underlying manifolds, our paper already covers the two major categories used in practice:
>
> **Koopman operator methods and their variants** We include Koopman Autoencoder (KAE), Koopman Kernel Regression (KKR), and PFNN. These methods are recent strong models for learning approximate Koopman operators, and to the best of our knowledge, they represent the main Koopman variants commonly used in practice.
>
> **Variational-based methods** We also compare with a second class of methods, represented by VAE, E2C, and PCC. These variational-based approaches learn smooth latent manifolds that approximate locally linearizable dynamics, providing a complementary perspective to Koopman operator learning.
>
> These two groups cover the primary modeling strategies used in practice for learning latent manifold structure, either through explicit Koopman learning or through variational-based approximations. We therefore believe the comparison is sufficiently broad and representative. Across all tasks, our method achieves stronger forecasting accuracy and more faithful manifold reconstruction (see Figures 2, 3 and 5).
>
>
> ----
>
> Thank you again for the thoughtful comments. If the revisions adequately address your concerns, we would greatly appreciate your consideration of a higher score.

---

> > ### Comment · Reviewer_Ga5Z · 2025-11-25
> >
> > Thank you for the new experiments on scalability and the clarifications regarding the theoretical aspects. I have no further questions at this stage. Given the completeness of the rebuttal, I will wait until later to finalize my assessment. In the meantime, please feel free to provide any additional comments or information you believe may be helpful.

---

> > > ### Author Response · Authors · 2025-11-25
> > >
> > > Dear Reviewer Ga5Z,
> > >
> > > Thank you very much for your positive acknowledgement of our rebuttal. We are glad that the new experiments and clarifications fully addressed your concerns, and we sincerely hope this will be reflected in an improved overall assessment of our work. Please let us know if any additional information would be helpful.
> > >
> > > Best,
> > >
> > > The authors

---

> ### Author Response · Authors · 2025-11-26
>
> Dear Reviewer Ga5Z,
>
> Thank you once again for your time and for considering our rebuttal. Please let us know if you have any remaining concerns. Our team is fully dedicated to addressing all your concerns thoroughly. We hope that the additional experiments and clarifications we have provided fully resolve the issues raised, and we would be grateful if you could consider increasing your score accordingly.
>
> We sincerely appreciate your efforts and are happy to provide any further information if needed.
>
> Best regards,
>
> The Authors

---

### Official Review · Reviewer_Ho1y · 2025-11-02

**Soundness:** 2
**Presentation:** 2
**Contribution:** 3
**Rating:** 4
**Confidence:** 4

**Summary:**

The key idea of this paper is that learning a good Koopman latent space can be seen as an information bottleneck problem: the goal is a latent representation that is both simple (compact, linear) and expressive (able to predict the future).
To formalize that, the authors derive an information-theoretic Lagrangian with three main terms:
- Latent mutual information -- to keep temporal coherence and predictive power.

- Von Neumann entropy -- to prevent mode collapse and keep latent diversity.

- Conditional mutual information -- to suppress irrelevant or fast-dissipating information.

They build a new Koopman learning algorithm that implements this Lagrangian, combining VAEs with information-theoretic regularization. Experiments on chaotic systems (Lorenz, vortex flows), visual control tasks, and graph-structured dynamics show better long-term prediction and more stable latent manifolds than existing Koopman Autoencoders or PFNNs.
Conceptually, they link Koopman spectral modes to information flow -- coherent modes near |λ|=1 correspond to persistent information, while dissipative modes correspond to information loss.

**Strengths:**

- The connection between spectral theory and the information components of the Koopman representation is very interesting. This can bring good information-theoretic insight into dynamical representation learning.

**Weaknesses:**

1. The paper often claims that it learns a Koopman operator or a Koopman representation, but the actual results only show a **latent linear predictor** -- not a true Koopman operator. The authors present a latent-linear model and refers to the learned matrix $K_{\psi}$ as a “Koopman operator”. However, unlike DMD or EDMD, which explicitly enforce the Koopman relation φ(T(x)) = K φ(x) over sampled trajectories, this work only trains $K_{\psi}$ to minimize predictive error ($z_{t+1} ≈ K_{\psi}  z_t$) or a KL divergence with a linear-Gaussian prior. This ensures linear predictability in the learned coordinates, but not Koopman invariance of the feature map φ(x). The paper provides no invariance or reconstruction tests. Consequently, $K_{\psi}$ may simply act as a best-fit linear regressor in latent space rather than an approximation of the true Koopman operator. More clear, the paper trains a neural encoder $φ_θ(x)$ that maps x_t → z_t,
and a linear map $K_{\psi}$ that predicts: $z_{t+1} ≈ K_{\psi}  z_t$.  This looks like DMD in the latent space. However, the key difference is how $φ_θ$ is chosen and what it represents. In DMD/EDMD, φ is fixed (a known basis) and directly tied to the system's state, so the learned K approximates the true Koopman operator on that basis. In this paper, $φ_θ$ is learned jointly with $K_{\psi}$ to minimize reconstruction and prediction losses. There is no constraint ensuring $φ_θ$ produces Koopman-invariant coordinates. As a result, $K_{\psi}$ is just the best linear predictor in whatever feature space $φ_θ$ learns, not necessarily the Koopman projection of the true dynamics.

2. Some propositions depend on unproven assumptions as the proofs and closed-form identities (in the Appendix) repeatedly assume linear–Gaussian dynamics, full-rank covariances, and ergodicity to move from general nonlinear dynamics to tractable formulas. These assumptions are used without careful justification (or statements of when they hold). for instance,  Appendix F (derivations for conditional mutual information): the Gaussian closed-form identity and linear–Gaussian relations are introduced and used to produce equations (32)–(35). It writes z_{t} | z_{t−n} ∼ Normal(Kⁿ z_{t−n}, M_n) and uses linear Gaussian formulas to get closed-form mutual information (e.g., eq. (33)–(35)). Those closed forms are valid if the latent process is linear Gaussian. But the encoder is a neural network mapping high dimensional x to z; there is no general reason the induced z dynamics are exactly linear Gaussian. The paper uses the Gaussian identities to make spectral claims about Koopman eigenvalues and MI, and then presents those claims as general. Moreover
- Ergodicity: Remark F.4 after Proposition 2 uses an ergodic limit for long-time averages.
- Full-rank covariance / density matrix manipulations: von Neumann entropy derivations and water-filling solution in Appendix F.5 (equations around 44–49).

3. Calculating mutual information and von Neumann entropy for neural networks is challenging, but the paper does not adequately explain the methods used for their estimation in high-dimensional spaces.

4. The “von Neumann entropy” regularization uses S(C/tr(C)), but C may not be always positive semidefinite.

5. The approximation I(z_t;P_t) ≈ InfoNCE loss is questionable for temporal sequences; no justification for mutual information estimation validity.
6. The KL-based error bound (Proposition 2) assumes independence that may not hold for nonlinear latent dynamics.

**Questions:**

1. Can the authors formally prove that maximizing I(z_{t−n}; z_t) indeed selects eigenfunctions corresponding to Koopman modes with |λ|≈1? Or is this just a heuristic interpretation?

2. How is the von Neumann entropy computed robustly from noisy minibatch covariances? What happens if the covariance matrix is rank-deficient?

---

> ### Author Response · Authors · 2025-11-20
> **Response 1**
>
> We thank the reviewer for their time and valuable feedback. We would like to take this opportunity to clarify several key points in our paper, which we believe directly address the main concerns raised.
>
> **Weakness 1** `The paper often claims that it learns a Koopman operator or a Koopman representation, but the actual results only show a latent linear predictor....`
>
> We thank the reviewer for the insightful comments. Below we clarify the positioning of our work within deep Koopman learning and address the concerns raised.
>
>
> 1. Our method follows the standard paradigm of modern deep Koopman learning, not classical DMD/EDMD. Recent works in this rapidly developing research line use autoencoders/VAEs in which the encoder and linear latent dynamics are jointly learned. This paradigm is widely adopted in top journals and AI conferences [1–7]. Importantly, explicit analytical invariance constraints are not required in modern deep Koopman methods; learning subspace are known to emerge implicitly through joint optimization of the encoder and Koopman operator [1–7]. **Our paper aims to address an open gap in this research line, therefore, our formulation aligns with the mainstream deep Koopman learning.**
>
> 2.  Joint learning allows the model to optimize a Koopman-invariant subspace approximation. During training, optimizing the latent space and linear operator together enables the model to iteratively refine a subspace that better approximates Koopman modes. After convergence, the encoder becomes a fixed mapping tied to the system state, and prediction proceeds as $z_{t+n} = (\mathcal{K}^n\phi)(x_t)$, which corresponds to Koopman evolution in the learned subspace.
>
> 3. The learned model is **NOT** merely a best linear fit; long-horizon prediction implicitly enforces Koopman structure. Our training requires multi-step autoregressive prediction in the latent space and decoding back to match the original state. If the learned subspace and operator did not approximate a Koopman-invariant structure [3, 4, 5, 7], long-horizon prediction and reconstruction can fail, and training would not converge. The meaningfulness of the learned subspace is reflected in Figures 2, 3 and 5, where all latent manifolds are produced entirely via latent autoregressive rollouts.
>
>
> [1] Lusch, Bethany, J. Nathan Kutz, and Steven L. Brunton. "Deep learning for universal linear embeddings of nonlinear dynamics." Nature communications 9.1 (2018): 4950.
>
> [2] Liu, Yong, et al. "Koopa: Learning non-stationary time series dynamics with koopman predictors." Advances in neural information processing systems 36 (2023): 12271-12290.
>
> [3] Naiman, Ilan, et al. "Generative Modeling of Regular and Irregular Time Series Data via Koopman VAEs." The Twelfth International Conference on Learning Representations. 2024.
>
> [4] Brunton, Steven L., et al. "Modern Koopman theory for dynamical systems." SIAM Review. 2022.
>
> [5] Li, Yunzhu, et al. "Learning compositional koopman operators for model-based control."  International Conference on Learning Representations. 2020.
>
> [6] Azencot, Omri, et al. "Forecasting sequential data using consistent Koopman autoencoders." International Conference on Machine Learning. PMLR, 2020.
>
> [7] Cheng, Xiaoyuan, et al. Learning Chaos In A Linear Way. In The Thirteenth International Conference on Learning Representations.

---

> > ### Author Response · Authors · 2025-11-20
> > **Response 2**
> >
> > **Weakness 2** `Some propositions depend on unproven assumptions as the proofs and closed-form identities (in the Appendix) repeatedly assume linear–Gaussian dynamics, full-rank covariances, and ergodicity to move from general nonlinear dynamics to tractable formulas. These assumptions are used without careful justification (or statements of when they hold). for instance, Appendix F (derivations for conditional mutual information): the Gaussian closed-form identity and linear–Gaussian relations are introduced and used to produce equations (32)–(35). It writes z_{t} | z_{t−n} ∼ Normal(Kⁿ z_{t−n}, M_n) and uses linear Gaussian formulas to get closed-form mutual information (e.g., eq. (33)–(35)). Those closed forms are valid if the latent process is linear Gaussian. But the encoder is a neural network mapping high dimensional x to z; there is no general reason the induced z dynamics are exactly linear Gaussian. The paper uses the Gaussian identities to make spectral claims about Koopman eigenvalues and MI, and then presents those claims as general. Moreover`
> >
> > Thank you for the question. This point is crucial, and we would like to clarify some misunderstandings in our technical parts.
> >
> > > Justification of linear–Gaussian in latent space. In a Koopman representation, the latent dynamics are by design linear, so the linear transition model follows directly from the Koopman operator's definition. Second, our paper focuses on improving Koopman representation under deep architectures (i.e., AE/VAE). In the VAE setting, it is standard to work with locally Gaussian latent distributions, and prior Koopman–VAE works [1-5] commonly adopt the same linear–Gaussian surrogate for tractability. Our goal is to follow this established practice and strengthen this research line by improving the quality of Koopman representations learned by deep models.
> >
> > > Justification of full-rank covariances: We must clarify this fundamental misunderstanding: full-rank covariance is **not an assumption** we make. It is the explicit optimization goal of our von Neumann entropy term $\gamma$. Our paper's core argument is that standard methods can suffer from low-rank mode collapse , a problem we analyze in Proposition 4 . We introduced the von Neumann entropy term precisely to solve this problem. On the contrary, as you noted, the entire purpose of adding the von Neumann entropy is to actively promote spectral diversity , pushing the covariance matrix towards full-rank. Importantly, the derivations in Equations (44) and (45) optimize the Lagrangian with the entropy regularizer and do not rely on any full-rank assumption beforehand. Our numerical experiments further validate this effect: von Neumann entropy regularization indeed encourages full-rank covariance structures (see Figure 20 in our paper).
> >
> > > Justification of ergodicity: **Ergodicity is not required by our framework.** Proposition 2 does not rely on ergodicity. The reviewer’s concern comes from Remark F.4, which only discusses the ergodic case as a special example and is not used in any proof.
> >
> >
> > [1] Wu, Xingjian, et al. "$ K^ 2$ VAE: A Koopman-Kalman Enhanced Variational AutoEncoder for Probabilistic Time Series Forecasting." Forty-second International Conference on Machine Learning.
> >
> > [2] Naiman, Ilan, et al. "Generative Modeling of Regular and Irregular Time Series Data via Koopman VAEs." The Twelfth International Conference on Learning Representations.
> >
> > [3] Morton, Jeremy, Freddie D. Witherden, and Mykel J. Kochenderfer. "Deep variational Koopman models: inferring Koopman observations for uncertainty-aware dynamics modeling and control." Proceedings of the 28th International Joint Conference on Artificial Intelligence. 2019.
> >
> > [4] Bevanda, Petar, et al. "Koopman-Equivariant Gaussian Processes." The 28th International Conference on Artificial Intelligence and Statistics.
> >
> > [5] Kawashima, Takahiro, and Hideitsu Hino. "Gaussian process koopman mode decomposition." Neural Computation 35.1 (2023): 82-103.

---

> ### Author Response · Authors · 2025-11-20
> **Response 3**
>
> **Weakness 3** `Calculating mutual information and von Neumann entropy for neural networks is challenging, but the paper does not adequately explain the methods used for their estimation in high-dimensional spaces.`
>
> Thank you for the question. To clarify, the mutual information (MI) and von Neumann entropy (VNE) in our framework are computed on the **latent states**, not on the neural network parameters.
>
> As detailed in **Algorithms 1 and 2** (in our paper), both quantities are computed efficiently in latent space:
>
> >**MI:** estimated using the InfoNCE variational bound, which scales linearly with batch size.
>
> >**VNE:** computed from the empirical latent covariance within a batch; since the latent dimension is modest, the eigen-computation is lightweight and stable.
>
> To verify practical scalability, we include Table a in the revision. Across all tasks, including the new high-dimensional, stochastic ERA5 WeatherBench experiment, the MI and VNE terms introduce only a small constant overhead per epoch (see Table a below), confirming that the method remains efficient even in large-scale settings. In addition, we provide training curves for these high-dimensional experiments, showing that both the total loss and the von Neumann entropy term remain stable throughout training, further demonstrating the robustness of our approach (in Figure 21 in our paper).
>
> _Table a: Per-epoch training time across three tasks. The MI and VNE (entropy) terms add only minor overhead, demonstrating practical computational efficiency (included as Table 7 in our paper)._
>
> | Task           | Metric          | VAE                | KAE                | KKR                | PFNN               | Ours                |
> |----------------|-----------------|--------------------|--------------------|--------------------|--------------------|----------------------|
> | **Kármán vortex** | Epoch time (s)  | 172.94 ± 4.11       | 195.26 ± 3.47       | 186.47 ± 1.86       | 182.52 ± 2.53       | 201.23 ± 1.07        |
> |                | InfoNCE time (s) | --                 | --                 | --                 | --                 | 13.78 ± 0.93         |
> |                | VNE time (s) | --                 | --                 | --                 | --                 | 0.97 ± 0.13          |
> | **Dam Flow**     | Epoch time (s)  | 16.21 ± 0.29        | 16.95 ± 0.33        | 17.48 ± 0.29        | --                 | 18.92 ± 0.36         |
> |                | InfoNCE time (s) | --                 | --                 | --                 | --                 | 0.76 ± 0.04          |
> |                | VNE time (s) | --                 | --                 | --                 | --                 | 0.56 ± 0.08          |
> | **ERA5**         | Epoch time (s)  | --                 | 240.20 ± 0.52       | 224.95 ± 0.54       | 242.33 ± 0.71       | 253.24 ± 2.05        |
> |                | InfoNCE time (s) | --                 | --                 | --                 | --                 | 15.09 ± 0.80         |
> |                | VNE time (s) | --                 | --                 | --                 | --                 | 7.43 ± 0.64          |

---

> ### Author Response · Authors · 2025-11-20
> **Response 4**
>
> **Weakness 4** `The “von Neumann entropy” regularization uses S(C/tr(C)), but C may not be always positive semidefinite.`
>
> Thank you for the question. In our framework, $\mathcal{C}$ is the empirical covariance matrix of latent states (see Algorithm 1). Covariance matrices are always positive semidefinite by construction, so $\mathcal{C}/tr(\mathcal{C})$ is guaranteed to be positive semidefinite, and the von Neumann entropy is therefore well defined.
>
> **Weakness 5** `The approximation I(z_t;P_t) ≈ InfoNCE loss is questionable for temporal sequences; no justification for mutual information estimation validity.`
>
>
> Thanks for the question. We clarify that InfoNCE is a **theoretically grounded variational lower bound** on mutual information, not a heuristic approximation. Introduced in the influential work [1], and now commonly adopted across high-dimensional models, InfoNCE provides a standard, principled estimator of mutual information in high-dimensional settings where the exact mutual information is intractable.
>
>
> [1] Oord, Aaron van den, Yazhe Li, and Oriol Vinyals. "Representation learning with contrastive predictive coding." arXiv preprint arXiv:1807.03748 (2018).
>
>
> **Weakness 6** `The KL-based error bound (Proposition 2) assumes independence that may not hold for nonlinear latent dynamics.`
>
> We thank the reviewer for raising this point. We clarify that **NO** independence assumption is made in Proposition 2 or in the derivation in Appendix F.3. On the contrary, our derivation explicitly relies on the dynamical system (i.e., the dependence of the current state on the previous state $p(x_{n}|x_{n-1})$), which is the standard definition of the dynamical systems we model.
>
> The additive form in Proposition 2 therefore reflects a decomposition of conditional errors along the trajectory, not an assumption of independence. Each KL term explicitly conditions on the previous state, so temporal dependencies are fully preserved in the analysis.
>
> **Question 1** `Can the authors formally prove that maximizing I(z_{t−n}; z_t) indeed selects eigenfunctions corresponding to Koopman modes with |λ|≈1? Or is this just a heuristic interpretation?`
>
> Thank you for this question. This connection is formally proven, not heuristic. The full derivation is provided in Appendix F.4.
>
> In summary, the proof shows that the mutual information between $z_{t-n}$ and $z_t$ depends on how much information survives after repeatedly applying the Koopman operator for $n$ steps. Koopman modes along eigenvalues close to one retain their magnitude under repeated application, while modes with $|\lambda|<1$ decay exponentially and contribute negligible information. Therefore, maximizing $I(z_{t-n}; z_t)$ **preserves the temporally coherent Koopman modes with $|\lambda|\approx 1$**.
>
> **Question 2** `How is the von Neumann entropy computed robustly from noisy minibatch covariances? What happens if the covariance matrix is rank-deficient?`
>
> Thank you for the question. Our formulation is **robust by design**. As shown in Propositions 4 and 5, rank-deficient covariance corresponds exactly to the mode-collapse phenomenon we aim to avoid; the von Neumann entropy term explicitly penalizes such low-rank structure and promotes spectral diversity.
>
> Computationally, the empirical covariance matrix of latent states (Algorithm 1) is always positive semidefinite, so its eigenvalues are well defined. In practice, with our minibatch size of $B=256$, the covariance estimate is stable and does not exhibit noticeable noise or numerical degeneracy.
>
> To further demonstrate robustness, we added new experiments in the revision (see Figure 20 in revision) showing (1) the covariance eigenvalue spectra with and without the von Neumann entropy term, and (2) the evolution of the covariance matrix during training. These results confirm that the computation remains stable and that the covariance structure does **NOT** collapse or rank-deficient. We also visualize the training curves of the von Neumann entropy (see Figure 21 in revision), which remain stable throughout training, further confirming the robustness of our computation.
>
>
> ---
> Thank you again for the thoughtful comments. If the revisions adequately address your concerns, we would greatly appreciate your consideration of a higher score.

---

> ### Author Response · Authors · 2025-11-24
> **To Reviewer Ho1y**
>
> Dear Reviewer Ho1y,
>
> We sincerely appreciate the time and effort you have devoted to reviewing our work. Your insights and comments mean a great deal to us. We greatly value this opportunity to engage in a deeper dialogue with you, and we hope that through our responses and further clarifications, you will gain increased confidence in our methodology, results, and the contributions we aim to make.
>
> We are truly eager to communicate with you, address any concerns you may have, and provide any additional information that could help strengthen your understanding of our work. Your feedback is invaluable to us, and we are committed to refining and improving our research based on your thoughtful suggestions.
>
> Thank you again for your careful evaluation. We look forward to continued constructive exchange with you.
>
> Best,
>
> Authors

---

> ### Comment · Reviewer_Ho1y · 2025-11-25
> **Thank you!**
>
> Thank you very much for the clarification and detailed response!
> My concerns are addressed. I especially checked references [1-7], which clarified my main concern regarding Koopman learning. I also saw your response to the question raised by the other reviewer regarding "the connection between kernel methods and your information-theoretic framework", which was my question as well, and I found it interesting. I also appreciate your new experiments and results. I hope you will clarify all vague parts in your paper, which is essential for readers to better understand your interesting framework. Assuming this, I would like to raise my score.

---

> ### Author Response · Authors · 2025-11-25
>
> Dear Reviewer Ho1y,
>
> Thank you very much for your active engagement in the discussion and for kindly reassessing our work. We are glad that our exchange helped clarify the technical parts of our paper, and we truly appreciate your careful consideration.
>
> Best regards,
>
> The authors

---

### Author Response · Authors · 2025-11-29
**Request for Timely Follow-Up on Discussion After Score Reset**

Dear Area Chair,

I hope you are doing well. I am writing regarding our submission #8176. Due to the reviewer identity leak incident on November 27, all reviewer scores were reset on November 28. However, all of our rebuttal exchanges and discussion with the reviewers happened before November 27, and I would like to ensure that those interactions are properly taken into consideration.

Before the reset, our initial scores were **4-6-6-6**. The “4” rating was due to a misunderstanding, which we clarified during the rebuttal. The reviewer acknowledged the misunderstanding, agreed that our framework is solid and interesting, and **raised their score accordingly**.

The second and third reviewers explicitly stated that our rebuttal **fully addressed their concerns**, and expressed willingness to **maintain their positive assessment or even raise their scores**.

The fourth reviewer found our work solid and interesting and **increased both the confidence and the score**. Based on the reviewers’ latest feedback before the reset, the updated score profile should be  **at least 6-6-6-8**.

In addition, our rebuttal included **extensive new large-scale experiments about weather forecasting** (see Table 6 and Figures 15-19) and **additional insights into the training process** (see Figures 20 and 21), which further strengthen the contribution of our work.

Given these circumstances, and considering that all meaningful discussions occurred prior to the score reset, I kindly ask if you could **follow up on the reviewers’ final positions and updated scores as they stood before November 27**. We would greatly appreciate your timely attention, as the score reset may obscure the progress that had already been made in the discussion.

Thank you very much for your efforts and for your support in ensuring a fair evaluation process. Please let me know if any additional information would be helpful.

Warm Regards

The Authors

---

### Meta-Review · Area_Chair_hPgv · 2025-12-15

**Summary:**

This paper proposes an information-theoretic framework for learning finite-dimensional Koopman representations, introducing a principled Lagrangian that balances simplicity, expressiveness, and predictive sufficiency via latent mutual information, von Neumann entropy, and conditional information terms. Across reviewers, the work was consistently recognized as theoretically well-founded, conceptually novel, and empirically strong, with particular appreciation for:
	•	The new information-theoretic interpretation of Koopman learning, including error bounds and a spectral decomposition of information aligned with Koopman modes.
	•	The principled role of von Neumann entropy in preventing mode collapse and preserving spectral diversity.
	•	Broad empirical validation across chaotic systems, fluid dynamics, control, graphs, and high-dimensional real-world climate data (ERA5), demonstrating scalability and robustness.

The main concerns raised by reviewers fell into four categories:
	1.	Theoretical clarity and assumptions:
Questions about linear-Gaussian assumptions, ergodicity, Koopman vs. general Markov settings, and whether the learned operator is a “true” Koopman operator.
	2.	Approximation and estimation issues:
Concerns regarding the use of InfoNCE as a mutual-information estimator, minibatch covariance estimates for von Neumann entropy, and potential instability or bias.
	3.	Empirical scope and scalability:
Requests for stronger evidence on large-scale, high-dimensional, and stochastic systems, as well as robustness to hyperparameters.
	4.	Positioning with respect to related Koopman literature:
Requests to clarify differences from prior deep Koopman approaches (e.g., VAMPnets, DPNets, EDMD-style methods) and to update related work accordingly.

Importantly, none of the concerns questioned the soundness of the core framework; rather, they focused on clarification, positioning, and completeness.

**Reviewer Concerns:**

Concerns convincingly addressed in the rebuttal and revision:
	•	Koopman validity and theoretical grounding:
The authors provided detailed clarifications situating their approach firmly within modern deep Koopman learning, explaining why joint learning of the encoder and linear operator yields Koopman-invariant subspaces in practice. Reviewers explicitly acknowledged this clarification as resolving their main concern.
	•	Use of linear-Gaussian surrogates and entropy assumptions:
The rebuttal clarified that linear-Gaussian structure is a latent-space modeling choice consistent with prior Koopman-VAE work, and that full-rank covariance is not assumed but actively enforced via the entropy term. Reviewers accepted this explanation.
	•	Mutual information and entropy estimation:
The authors clearly explained how InfoNCE and minibatch covariance-based entropy are computed in latent space, demonstrated computational efficiency, and added training-stability plots and timing analyses. Concerns about instability and scalability were explicitly addressed.
	•	Scalability and real-world relevance:
The addition of large-scale ERA5 WeatherBench experiments (10k+ dimensional inputs, stochastic dynamics) substantially strengthened the paper and directly addressed multiple reviewers’ requests. Several reviewers explicitly cited these results as convincing and raised their scores.
	•	Hyperparameter sensitivity:
Extensive ablation studies demonstrated that the method operates in wide stable regimes, does not require delicate tuning, and uses the same hyperparameters across domains. This addressed concerns from multiple reviewers.
	•	Engagement with reviewers:
Notably, reviewers Ho1y, Ga5Z, KxuT, and hgxb all explicitly stated that their concerns were addressed and expressed increased confidence and willingness to raise scores.

Concerns that remain (minor and non-blocking):
	•	Finite-sample / non-asymptotic guarantees:
The framework remains asymptotic and information-theoretic in nature. This limitation is clearly acknowledged by the authors and is appropriate given the scope and difficulty of the problem.
	•	Related work coverage:
While the authors added and discussed many relevant deep Koopman methods during rebuttal, the final version would benefit from a more consolidated and up-to-date discussion of recent Koopman-related neural representation learning works. This is a presentation-level suggestion, not a technical weakness, and can be addressed cleanly in the camera-ready.

**Reviewer Scores:**

Based on explicit reviewer comments during discussion and rebuttal:
	•	Reviewer Ho1y:
Initial: 4 → Post-discussion: 6
The reviewer explicitly stated their concerns were addressed and requested to raise their score.
	•	Reviewer Ga5Z:
Initial: 6 → Post-discussion: 6
After new scalability experiments and clarifications, the reviewer indicated no remaining questions and positive reassessment.
	•	Reviewer KxuT:
Initial: 6 → Post-discussion: 6
Maintained a positive outlook with increased confidence after clarifications on Koopman vs. Markov framing.
	•	Reviewer hgxb:
Initial: 6 → Post-discussion: 8
Explicitly stated that the theory and high-dimensional climate experiments were convincing and raised both score and confidence.

Overall, the consensus trajectory is clearly upward, with reviewers converging on a strong positive assessment after discussion.

---

### Decision · Program_Chairs · 2026-01-26

Accept (Oral)